# Characterization of a *Salmonella enterica* serovar Typhimurium lineage with rough colony morphology and multidrug resistance

Ying Xiang[1,10], Kunpeng Zhu[1,2,10], Kaiyuan Min[3,10], Yaowen Zhang[1,4,10], Jiangfeng Liu[3,10], Kangkang Liu[1], Yiran Han[1], Xinge Li[1], Xinying Du[1], Xin Wang[1], Ying Huang[1], Xinping Li[1], Yuqian Peng[1], Chaojie Yang[1], Hongbo Liu[1], HONGBO Liu[1], Xiaoying Li[1], Hui Wang[1], Chao Wang[1], Qi Wang[1], Huiqun Jia[1], Mingjuan Yang[1], Ligui Wang[1], Yarong Wu[5], Yujun Cui[5], Fei Chen[6], Haiyan Yang[7], Stephen Baker[8], Xuebin Xu[9]✉, Juntao Yang[3]✉, Hongbin Song[1]✉ & Shaofu Qiu[1]✉

*Salmonella enterica* serovar Typhimurium (*S*. Typhimurium) is a major cause of salmonellosis, and the emergence of multidrug-resistant pathovariants has become a growing concern. Here, we investigate a distinct rough colony variant exhibiting a strong biofilm-forming ability isolated in China. Whole-genome sequencing on 2,212 Chinese isolates and 1,739 publicly available genomes reveals the population structure and evolutionary history of the rough colony variants. Characterized by macro, red, dry, and rough (mrdar) colonies, these variants demonstrate enhanced biofilm formation at 28 °C and 37 °C compared to typical rdar colonies. The mrdar variants exhibit extensive multidrug resistance, with significantly higher resistance to at least five classes of antimicrobial agents compared to non-mrdar variants. This resistance is primarily conferred by an IncHI2 plasmid harboring 19 antimicrobial resistance genes. Phylogenomic analysis divides the global collections into six lineages. The majority of mrdar variants belong to sublineage L6.5, which originated from Chinese smooth colony strains and possibly emerged circa 1977. Among the mrdar variants, upregulation of the *csgDEFG* operons is observed, probably due to a distinct point mutation (−44G > T) in the *csgD* gene promoter. Pangenome and genome-wide association analyses identify 87 specific accessory genes and 72 distinct single nucleotide polymorphisms associated with the mrdar morphotype.

*Salmonella* are among the most common enteric pathogens in humans and animals worldwide and cause typhoidal or nontyphoidal *Salmonella* (NTS) infections. NTS serovars, which can cause salmonellosis, are responsible for an estimated 153 million illnesses and 56,969 deaths among humans worldwide each year[1], representing a significant global public health burden. *Salmonella enterica* serovar Typhimurium (*S*. Typhimurium), one of the most common NTSs, is a leading cause of human salmonellosis worldwide[2,3]. In the United States, *S*. Typhimurium

is the most common *Salmonella* serovar associated with sporadic infections and foodborne outbreaks[4]. In Europe, *S.* Typhimurium is the second most common serovar isolated from humans[5]. In Asia, especially in China, *S.* Typhimurium is also among the most frequently encountered serovars[5–7].

*S.* Typhimurium has shown remarkable genetic diversity and the ability to sense and respond to environmental stress. Mutation and evolution are primary drivers of its adaptation to environments[8]. Whole-genome sequencing technology has revolutionized *Salmonella* epidemiological research, enabling the identification of sequence types, the prediction of key genes, the tracking of transmission, the analysis of mutation evolution, and the examination of pathogen evolutionary patterns[9,10]. In recent decades, several dominant *S.* Typhimurium variants, including definitive phage types DT9, DT204, DT29, DT104, and DT193, the monophasic variant 1,4,[5],12:i:-, and invasive NTS (iNTS), have been associated with successive human salmonellosis epidemics[2,3,11–14]. Moreover, studies have investigated *S.* Typhimurium genomic epidemiology at global and regional levels and identified several phylogenetic lineages characterized by multilocus sequence types (STs), such as ST19, ST34, ST36, and ST313[2–4,11–16]. Recently, *Salmonella* enterica serovar 1,4,[5],12:i:-, a monophasic variant of *S.* Typhimurium belonging to ST34, has emerged as a major global cause of NTS disease in animals and humans. Whole-genome evolutionary analysis has shown that this variant evolved into an independent clone and acquired a new genomic island SGI-4, which conferred resistance to multiple antibiotics and the heavy metal copper. Additionally, gene insertion and deletion events have played important roles in the microevolution of this clone[3]. Furthermore, whole-genome analysis of ST313 African isolates has indicated the emergence of two different lineages from independent origins. Lineage I prevailed in the southwest region of Africa, while Lineage II prevailed in Malawi[2]. Later research identified a sub-lineage, II.1, in Congo, which carried IncHI1 plasmids that conferred resistance to azithromycin and extended-spectrum beta-lactamases (ESBLs)[17]. Point mutations in the promoter region of the virulence gene *pgtE* have been found to upregulate its expression in ST313, contributing to increased virulence and thus distinguishing it from ST19 in phenotype[18]. These dominant pathovariants and epidemic lineages exhibit distinct host ranges, niche adaptations, levels of pathogenicity, and risks to food safety. Their infections are associated with a higher probability of hospitalization and treatment failure, leading to prolonged infection and potential onward transmission. Therefore, they pose serious public health threats[11,13,14,17,19]. The emergence of new pathovariants is an ongoing process as new hosts or environmental niches arise because of natural processes or human intervention[20]. It has been reported that epidemic lineages dominate for 4–15 years before being replaced by a new dominant variant[3,21].

In this work, we identify a rough colony variant that has become increasingly prevalent in China in recent years. These rough colony variants exhibit an enhanced biofilm-forming ability and extensive multidrug resistance (MDR), defined as resistance to at least five classes of antimicrobial agents. This indicates a potentially improved capacity for environmental persistence. To understand the evolutionary trajectory and genomic events mediating the microevolution of this variant, phylogenomic analyses are performed by whole-genome sequencing (WGS) of 2212 Chinese isolates contextualized with 1,739 publicly available isolates from multiple geographical regions. We determine a novel phylogenetic lineage characterized by the widespread presence of these rough colony variants. This lineage primarily consists of strains isolated in China and likely originating from Chinese smooth colony strains around 1977. We find that 87 specific accessory genes and 72 distinct single nucleotide polymorphisms (SNPs) are associated with the rough colony variants through pangenome and genome-wide association studies. Phenotypic and molecular investigations are also performed to determine the key factors contributing to the emergence and establishment of this variant. We find that these rough variants show increased expression of *csgDEFG* operons, due to a mutation in the *csgD* gene promoter. To avoid potential confusion, in our study, the term "rough colony variants" refers to strains that exhibit a distinct wrinkled morphology when inoculated and cultured on the modified Swarmagar medium. These strains demonstrate a strong biofilm-forming ability as observed through Congo red assays. The term "novel lineage" refers to the L6.5 lineage identified in the phylogenetic analysis.

## Results

### Emergence and prevalence of a variant of *S.* Typhimurium with a rough colony morphology

We collected 2212 *S.* Typhimurium isolates from 20 cities or provinces spanning different geographic regions of China (Northeast, Northwest, Southeast, Southwest, North, South, and Central China) from 2006 to 2018; these isolates were recovered from diverse sources, including humans, food and the environment (Fig. 1A–C, Table 1). We identified a variant of *S.* Typhimurium with a rough colony morphology. Overall, this variant accounted for 29.6% of all Chinese *S.* Typhimurium isolates and has demonstrated a persistent prevalence in recent years (Fig. 1B, Table 1). This variant was isolated from 19 of the 20 monitored regions of China and was mainly distributed in the East, South, Southwest, and Central regions of China (Fig. 1A, Table 1). A total of 654 strains exhibiting the rough colony variant were successfully identified and isolated from various samples, including humans (419, 64.1%), food (178, 27.2%), and environment (57, 8.7%). Furthermore, it was observed that the prevalence of the rough colony variant significantly varied across different isolation sources, with environmental sources (40.7%) exhibiting a higher proportion compared to human sources (29.4%) and food sources (27.6%) ($P < 0.05$) (Fig. 1C, Table 1).

When the strains were inoculated onto a modified Swarmagar medium for cultivation, the rough colonies exhibited a film-like and wrinkled morphology, while the control *S.* Typhimurium (ATCC 14028) did not display such morphology characteristics. When incubated at 28 °C on Luria-Bertani (LB) agar containing Congo red and Coomassie blue without salt, the control *S.* Typhimurium (ATCC 14028) formed a typical red, dry, and rough (rdar) colony with the formation of concentric rings and a wrinkled appearance after 3 days. In comparison, the rough variants identified in this study (represented by SH15SF175, hereafter abbreviated as SF175) appeared as macro-rdar colonies (designated as mrdar colonies) with more pronounced wrinkles, while the smooth colony strains (represented by 2015114) formed small and white (saw) colonies with no obvious corrugations or wrinkles (Fig. 2A). Of particular note, the rough colony variants remained mrdar-positive with wrinkles at 37 °C, while the control ATCC 14028 and the smooth 2015114 isolates were rdar-negative (Fig. 2B). Furthermore, over extended cultivation periods, the mrdar colony variants developed increasingly larger colonies with more pronounced wrinkles than the control ATCC 14028 strain and the smooth isolates (Fig. 2A, B). Scanning electron microscopy (SEM) examinations revealed that the SF175 mrdar colony formed thick layers of cells that were densely embedded in extracellular polymeric substances (EPSs) with the more extensive formation of the cross-linked fibril matrix (Fig. 2C, D). The EPS and fibrillar network formed a highly resilient structure that appeared to bind the cells together. Notably, at 28 °C, these structures appeared to be thicker and denser compared to those at 37 °C. The control strain and the smooth 2015114 isolate formed monolayers of cells under the same conditions, and these cells were separated from each other.

### Biofilm formation

Crystal violet biofilm assays and pellicle formation assays showed that the mrdar variants examined in this study had greater biofilm-forming ability than the smooth variants; for example, the mrdar colony from

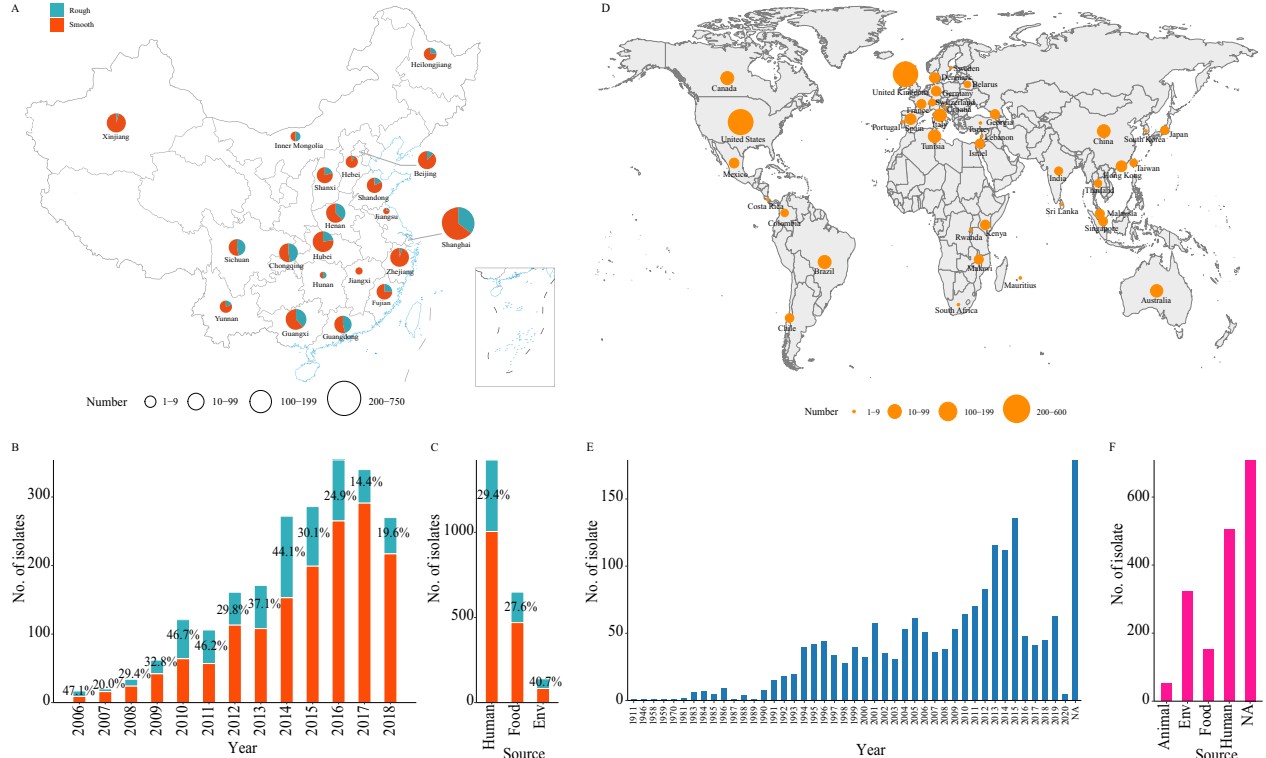

**Fig. 1 | Distribution of the global *S.* Typhimurium isolates used in this study.**
**A** Regional distribution of the 2,212 Chinese *S.* Typhimurium isolates identified in this study. The size of each circle represents the number of strains. The base map was generated using publicly accessible data from the Tianditu Administrative Division Service Center (https://cloudcenter.tianditu.gov.cn/administrativeDivision/). **B** Temporal distribution of the Chinese *S.* Typhimurium isolates. **C** Sources of the Chinese *S.* Typhimurium isolates. The red and green colors represent the smooth and rough colony strains, respectively. **D** Geographic distribution of the 1739 *S.* Typhimurium isolates from the NCBI. The world map data were sourced from the maps package in the R programming language (https://rdocumentation.org/packages/maps). **E** Temporal distribution of the NCBI *S.* Typhimurium isolates. **F** Sources of the NCBI *S.* Typhimurium isolates. Source data are provided as a Source Data file. Env environment, NA not available.

SF175 generated significantly more biofilm than the control ATCC strain and strain 2015114 at both 28 and 37 °C (Fig. S1, S2, S3). Moreover, confocal laser scanning microscopy (CLSM) observations showed that the biofilm of the mrdar colony morphotype had more EPSs (green color) than the smooth morphotype and ATCC 14028. Overall, the mrdar variants exhibited greater biofilm-forming ability at 28 °C than at 37 °C (Fig. S1).

### Antimicrobial resistance (AMR) analysis of the mrdar variants of *S.* Typhimurium

The antimicrobial susceptibility testing revealed that the Chinese *S.* Typhimurium isolates exhibited extensive AMR profiles (Table 2). The majority of these isolates exhibited MDR, with 79.5% demonstrating resistance to three or more Clinical and Laboratory Standards Institute (CLSI) classes, including MDR phenotypes such as ASSuT (45.5%, resistance to at least ampicillin, streptomycin, sulfisoxazole, and tetracycline) and ACSSuT (22.0%, resistance to at least ampicillin, chloramphenicol, streptomycin, sulfisoxazole, and tetracycline) (Table 2). Notably, the mrdar variants exhibited higher rates of resistance to multiple antibiotics than the smooth colony isolates, particularly for older antibiotics, including tetracycline (90.5% vs. 71.1%), ampicillin (89.4% vs. 70.7%), sulfisoxazole (93.7% vs. 68.5%), nalidixic acid (95.3% vs. 34.3%), chloramphenicol (71.6% vs. 36.4%), trimethoprim/sulfamethoxazole (58.3% vs. 31.3%), and gentamicin (57.3% vs. 17.0%). A clear difference in MDR was also observed among the rough and smooth colony isolates, including those with resistance to ≥3 classes (95.4% vs. 72.8%), ≥4 classes (92.5% vs. 65.7%), ≥5 classes (83.9% vs.

28.9%) and ≥6 classes (58.1% vs. 14.8%) and the ACSSuT phenotype (34.6% vs. 16.8%) (Table 2).

### Population structure of global *S.* Typhimurium, including the Chinese isolates

The maximum likelihood (ML) phylogenetic tree was constructed based on 85,508 core SNPs identified from a dataset of 3951 genomes. This dataset comprised 2212 strains that we sequenced and analyzed ourselves, and 1739 strains whose sequencing data were downloaded from the NCBI *Salmonella* Typhimurium Genome database (Supplementary Data 1). We downloaded all available WGS data of *Salmonella* Typhimurium from the NCBI database in August 2021, and after excluding records with incomplete strain background information, such as location and collection dates, we were left with data for 1739 strains. The cgMLST analysis showed that the rough colony strains formed distinct clusters separate from other branches, suggesting considerable genetic divergence (Fig. S4).

The global *S.* Typhimurium populations, rooted with *S.* Paratyphi A270 (accession ERR326600), could be divided into six distinct lineages (L1 to L6) in hierBAPS analysis. Among these lineages, L6 was the largest lineage, and L4 was the smallest (Fig. 3A). The multilocus sequence typing (MLST) analysis showed that the global *S.* Typhimurium populations contained two predominant STs, ST19 (52.3%) and ST34 (42.3%) (Fig. 3A, Supplementary Data 2). ST19 was mainly distributed in Lineages L1, L2, L3, and L5, ST34 was mainly distributed in Lineage L6, and the isolates in Lineage L4 mainly belonged to ST36.

**Table 1 | Characteristics of the Chinese S. Typhimurium strains in this study**

| Characteristic | Total (n = 2212) | Rough (n = 654) | Smooth (n = 1558) |
|---|---|---|---|
| Year, n (%) | | | |
| 2006 | 17 (0.8) | 8 (1.2) | 9 (0.6) |
| 2007 | 20 (0.9) | 4 (0.6) | 16 (1.0) |
| 2008 | 34 (1.5) | 10 (1.5) | 24 (1.5) |
| 2009 | 61 (2.8) | 20 (3.1) | 41 (2.6) |
| 2010 | 120 (5.4) | 56 (8.6) | 64 (4.1) |
| 2011 | 106 (4.8) | 49 (7.5) | 57 (3.7) |
| 2012 | 161 (7.3) | 48 (7.3) | 113 (7.3) |
| 2013 | 170 (7.7) | 63 (9.6) | 107 (6.9) |
| 2014 | 270 (12.2) | 119 (18.2) | 151 (9.7) |
| 2015 | 286 (12.9) | 86 (13.1) | 200 (12.8) |
| 2016 | 357 (16.1) | 89 (13.6) | 268 (17.2) |
| 2017 | 340 (15.4) | 49 (7.5) | 291 (18.7) |
| 2018 | 270 (12.2) | 53 (8.1) | 217 (13.9) |
| Region, n (%) | | | |
| Central China | 276 (12.5) | 86 (13.1) | 190 (12.2) |
| Eastern China | 1028 (46.5) | 289 (44.2) | 739 (47.4) |
| Northeast China | 39 (1.8) | 9 (1.4) | 30 (1.9) |
| Northern China | 214 (9.7) | 32 (4.9) | 182 (11.7) |
| Northwest China | 117 (5.3) | 5 (0.8) | 112 (7.2) |
| Southern China | 312 (14.1) | 133 (20.3) | 179 (11.5) |
| Southwest China | 226 (10.2) | 100 (15.3) | 126 (8.1) |
| Source, n (%) | | | |
| Human | 1426 (64.5) | 419 (64.1) | 1007 (64.6) |
| Food | 646 (29.2) | 178 (27.2) | 468 (30.1) |
| Environment | 140 (6.3) | 57 (8.7) | 83 (5.3) |
| Sex, n (%) | 1426 | 419 | 1007 |
| Female | 551 (38.6) | 163 (38.9) | 388 (38.5) |
| Male | 763 (53.5) | 244 (58.2) | 519 (51.5) |
| NA | 112 (7.9) | 12 (2.9) | 100 (9.9) |
| Age in years, n (%) | 1426 | 419 | 1007 |
| 0–6 | 819 (57.4) | 303 (72.3) | 516 (51.2) |
| 7–17 | 47 (3.3) | 17 (4.1) | 30 (3.0) |
| 18–40 | 195 (13.7) | 36 (8.6) | 159 (15.8) |
| 41–65 | 167 (11.7) | 38 (9.1) | 129 (12.8) |
| ≥66 | 82 (5.8) | 13 (3.1) | 69 (6.9) |
| NA | 116 (8.1) | 12 (2.9) | 104 (10.3) |

*NA* not available.

ST19 (73.4%) was the main ST among the NCBI sequences, but ST34 (59.3%) was the main ST among the Chinese isolates in this study (Supplementary Data 2).

All lineages appeared to have a global distribution (Fig. 3A, Fig. S5). Organisms from North America were widely distributed across all six lineages, indicating their basal position within the current *S.* Typhimurium populations. Although the Chinese isolates in this study were also present in all six lineages, they were primarily observed in Lineage L6 (59.0%). The Chinese organisms in each lineage were clustered together and formed locally established groups that likely originated from abroad, predominantly from North America (Fig. 3A, B, Fig. S5).

From the rooted phylogenetic tree and BEAST analysis, we estimated that the current *S.* Typhimurium population descended from a most recent common ancestor (MRCA) that existed circa 1696 (95% confidence interval [CI]: 1686–1706). It then diversified into six distinct lineages during the 1800s (L1), 1820s (L2), 1790s (L3), 1860s (L4), 1910s (L5), and 1880s (L6) (Fig. 3A, Fig. S6). These findings suggest that the current *S.* Typhimurium population descended from a common ancestor and has successfully dispersed and evolved into several phylogenetic lineages that are now circulating globally. The locally established Chinese groups can probably be traced back to ancestors from abroad, especially from North America.

**Phylogenetic analysis of the mrdar variants of S. Typhimurium**
To determine the phylogeny of the mrdar variants, a separate ML tree was constructed based on 21,168 core SNPs identified from 1754 isolates of Lineage L6 (Fig. 3B). The L6 lineage, a recently derived clade that developed circa 1884 (95% CI: 1874–1894), comprised 44.4% of all isolates in this study. Among the 654 mrdar colony variants, the majority were typed as ST34 (621, 95.0%) and located in Lineage L6 (625, 95.6%), particularly sublineage L6.5 (614, 93.9%). It was estimated that the mrdar variants of L6.5 could be traced back to 1977 (95% CI: 1966–1987) and likely originated from the smooth phenotype of *S.* Typhimurium in China (Fig. 3B). Based on the regional distribution data, the sublineage L6.5 was primarily composed of strains from China, including strains from various regions within the country. Additionally, this sublineage included strains from other parts of the world, such as Europe (23 isolates), North America (11 isolates), Oceania (four isolates), and South America (one isolate) (Fig. 3C, Supplementary Data 1).

**Distribution of the AMR determinants among the global S. Typhimurium lineages**
We examined the presence of genetic determinants of AMR, including the acquired resistance gene content and the presence of QRDR mutations[22]. A total of 152 different AMR determinants were identified, and each of the isolates contained numerous AMR determinants (ranging from 20 to 50; Fig. S7A–C, Supplementary Data 1). The six lineages contained different contents of AMR determinants, and the isolates in Lineage L6 contained more AMR determinants than those in other lineages (Fig. S7A, Fig. S8). Moreover, the mrdar variants carried more AMR determinants than the smooth colony isolates (Fig. S7B), and the mrdar variants in Lineage L6 carried more AMR determinants than the smooth colony strains in L6 (Fig. S7C). In particular, the rough colony variants in sublineage L6.5 carried more AMR determinants, including 12 aminoglycoside resistance genes (*AAC(3)-IV, AAC(6')-Ib7, AAC(6')-Ib-cr6, aadA1, aadA2, aadA3, aadA12, aadA25, strA, strB, APH(3')-Ia,* and *APH(4)-Ia*), three beta-lactam resistance genes ($bla_{OXA-1}$, $bla_{TEM-1}$, and $bla_{TEM-60}$), two efflux pump-mediated genes with decreased susceptibility to disinfectants (*qacEdelta1* and *qacL*), three phenicol resistance genes (*catB3, floR,* and *cmlA1*), one rifamycin resistance gene (*arr-3*), three sulfonamide resistance genes (*sul1, sul2,* and *sul3*), two tetracycline resistance genes (*tetB* and *tetR*), one trimethoprim resistance gene (*dfrA12*), and two quinolone resistance genes (*oqxA* and *oqxB*) (Fig. S8). The differences in resistance phenotypes between mrdar variants and smooth strains were largely due to the presence of specific resistance genes. For example, aminoglycoside resistance genes can confer resistance to gentamicin, and beta-lactam resistance genes can confer resistance to ampicillin. Moreover, the identification of resistance genes such as *floR* and *arr3* suggests that there may be differences in resistance to other antibiotics that have not been tested yet, such as fluphenicol and rifampicin. Resistance to fluoroquinolones is commonly associated with point mutations in quinolone resistance-determining regions (QRDRs) of the *gyrA, gyrB, parC,* and *parE* genes[23]. The QRDR mutations were mainly observed in Chinese strains, particularly within sublineages L6.5 and L1.3. Within sublineage L6.5, the mrdar variants mostly harbored the D87N or D87Y mutation in the *gyrA* gene, while sublineage L1.3 strains, characterized by their smooth morphology, carried mutations not only in the *gyrA* gene (S83L and/or D87N) but also in the *parC* gene (S80R) (Fig. S8). The presence of double or triple mutations in the QRDR is correlated with a high level

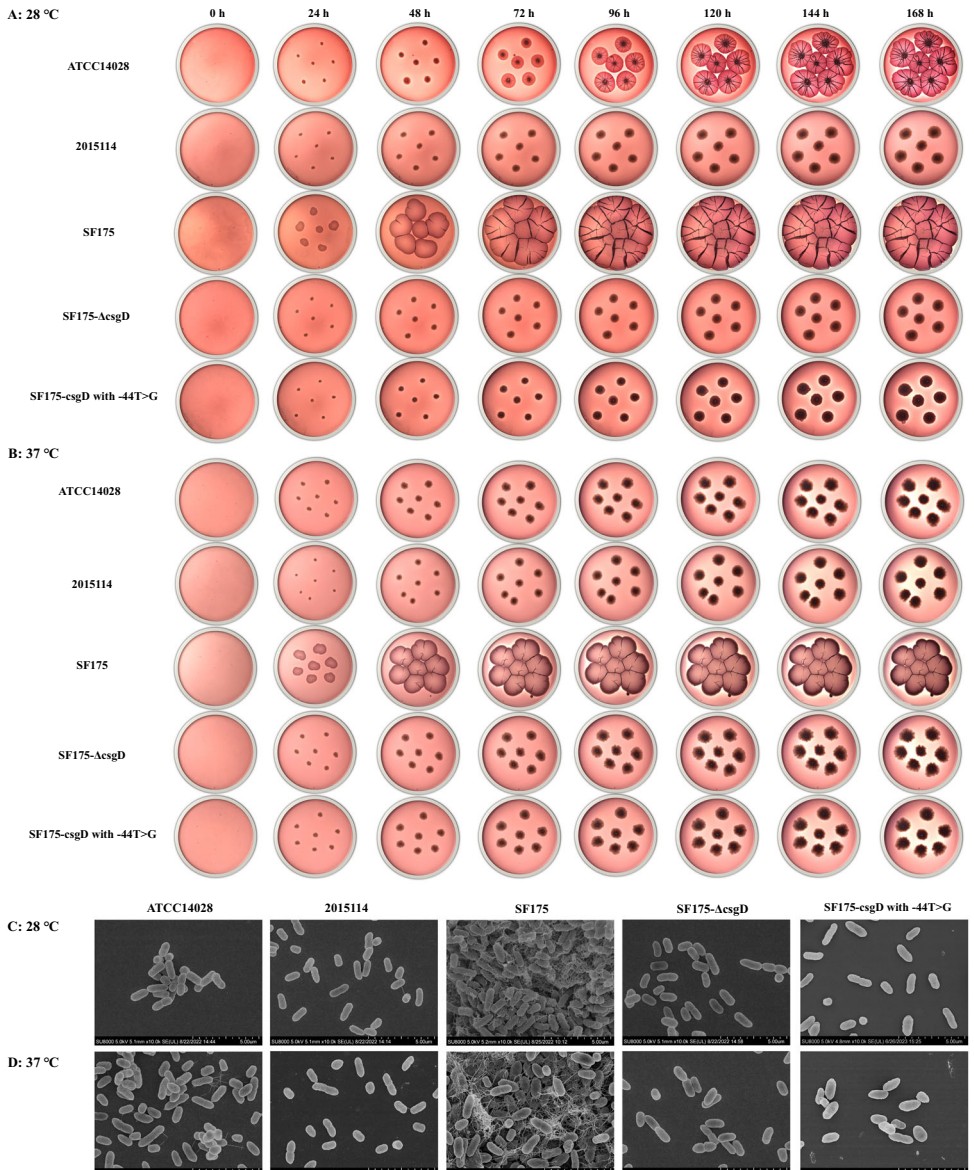

**Fig. 2 | Morphology of the rough and smooth colonies of the representative *S*. Typhimurium isolates in this study.** In this study, the rough colony strain SF175, smooth colony strain 2015114, control strain ATCC 14028, and mutants SF175-Δ*csgD* and SF175-*csgD* with -44 T > G were selected as representatives to observe the colony morphology. **A**, **B** Colony morphology of the representative *S*. Typhimurium isolates grown for seven consecutive days on Luria-Bertani agar containing Congo red and Coomassie blue without salt at 28 °C and 37 °C. **C**, **D** Scanning electron micrographs of the representative *S*. Typhimurium isolates cultured for 24 h at 28 °C and 37 °C. Representative scanning electron micrograph for one replicate. Scale bar, 5 mm. Source data are provided as a Source Data file.

of resistance to ciprofloxacin, and single mutations are associated with decreased sensitivity to this antibiotic[24]. Consequently, compared to smooth colony strains, the mrdar variants exhibited a lower resistance rate (3.98% vs. 7.18%) and a higher intermediate rate (10.70% vs. 2.50%) to ciprofloxacin.

### Distribution of virulence factors (VFs) among global *S*. Typhimurium lineages

A total of 265 different virulence genes were identified from the VF database (VFDB), and each of the isolates contained numerous VFs (ranging from 61 to 264; Supplementary Data 1). The six lineages contained different contents of virulence genes, with the isolates in Lineage L2 harboring the most, followed by L3, L1, L5, L6, and L4 (Fig. S7D, Fig. S9). Furthermore, we found that some VFs were associated with the *S*. Typhimurium lineages; for example, the plasmid-encoded fimbriae gene *pefABCD*, the type III secretion system (T3SS)-secreted effector gene *spvBCD* (*Salmonella* pathogenicity island 2 [SPI-2] encoding), and

genes *mig-5* and *rck* were present in the strains in Lineages L2 and L3 and some of the strains in Lineage L1 but absent in the strains in Lineages L4, L5 and L6 (Fig. S9). Overall, mrdar variant strains had a lower presence of virulence genes compared to non-variant strains (Fig. S7E). Nevertheless, within the L6 lineage, mrdar variants showed a higher abundance of virulence genes compared to smooth isolates (Fig. S7F). Additionally, the mrdar variants in sublineage L6.5 primarily carried the *fljAB* genes (Fig. S9).

### Identification of AMR- and virulence-associated plasmids

The plasmid replicon prediction results showed that a total of 47 plasmid replicons were predicted. The IncFII(S) replicon was present in lineages L1, L2, and L3, and the IncHI2 replicon was detected in sublineage L6.5, where the rough colony strains were mainly distributed (Fig. 4A). The combination of Illumina and Nanopore sequencing data led to the generation of two complete plasmids, specifically pSH15G1765-2 and pYY-2016-057-2 (Fig. 4B, C). The plasmid

**Table 2 | Antimicrobial resistance of the Chinese S. Typhimurium strains in this study**

| Antimicrobial | Antimicrobial resistance (%) | | | P value* |
|---|---|---|---|---|
| | Total (n = 2212) | Rough (n = 654) | Smooth (n = 1558) | |
| Ceftriaxone | 15.6 | 17.7 | 14.7 | 0.087 |
| Tetracycline | 76.7 | 90.5 | 71.1 | 9.384E-23 |
| Ceftiofur | 16.0 | 17.7 | 15.2 | 0.163 |
| Cefoxitin | 3.7 | 2.4 | 4.2 | 0.063 |
| Gentamicin | 28.9 | 57.3 | 17.0 | 8.329E-81 |
| Ampicillin | 76.2 | 89.4 | 70.7 | 3.725E-21 |
| Chloramphenicol | 46.8 | 71.6 | 36.4 | 2.197E-51 |
| Ciprofloxacin | 6.2 | 4.0 | 7.2 | 0.006 |
| Trimethoprim/ sulfamethoxazole | 39.3 | 58.3 | 31.3 | 4.450E-32 |
| Sulfisoxazole | 75.9 | 93.7 | 68.5 | 1.573E-36 |
| Nalidixic acid | 52.3 | 95.3 | 34.3 | 4.454E-150 |
| Streptomycin | 54.2 | 56.0 | 53.4 | 0.272 |
| Azithromycin | 9.7 | 9.2 | 9.9 | 0.651 |
| Amoxicillin/ clavulanic | 2.9 | 3.2 | 2.8 | 0.730 |
| ≥3 CLSI classes | 79.5 | 95.4 | 72.8 | 4.240E-21 |
| ≥4 CLSI classes | 73.6 | 92.5 | 65.7 | 0.009 |
| ≥5 CLSI classes | 45.2 | 83.9 | 28.9 | 1.706E-65 |
| ≥6 CLSI classes | 27.6 | 58.1 | 14.8 | 2.223E-95 |
| ASSuT | 45.5 | 49.7 | 43.8 | 0.011 |
| ACSSuT | 22.0 | 34.6 | 16.8 | 7.111E-20 |
| ACSSuTCRO | 5.8 | 6.1 | 5.6 | 0.935 |
| ACSSuTCIP | 4.1 | 2.0 | 5.0 | 0.002 |
| ACSSuTAZI | 3.7 | 4.1 | 3.5 | 0.533 |

Two-sided chi-square tests were performed without adjusting for multiple comparisons.
*CLSI* Clinical and Laboratory Standard Institute, *ASSuT* resistance to ampicillin, streptomycin, sulfamethoxazole/sulfisoxazole, and tetracycline, *ACSSuT* resistance to ampicillin, chloramphenicol, streptomycin, sulfamethoxazole/sulfisoxazole, and tetracycline, *CRO* ceftriaxone, *CIP* ciprofloxacin, *AZI* azithromycin.
*$P < 0.05$ denotes that there is a clear significant difference in resistance to the corresponding antimicrobials.

pSH15G1765-2, which belonged to the IncFII(S) replicon and measured 114,289 bp in size, was prevalent in lineages L1, L2, and L3. This plasmid was highly similar to plasmid pST90-2 (accession CP050736) identified from a Chinese S. Typhimurium in 2012 and the virulence plasmid pSLT (accession AE006471)[25]. Similar to pSLT and pST90-2, the plasmid pSH15G1765-2 carried the virulence genes *pefABCD*, *spvBCD*, and *rck* (Fig. 4B); compared with these two plasmids, pSH15G1765-2 acquired *ISStma11* and the mercury resistance (mer) operon *merTPCA*, which is associated with mercury resistance[26]. The plasmid pYY-2016-057-2, which was frequently found in the L6.5 sublineage, measured 183,430 bp and belonged to the IncHI2 replicon. Its structure was similar to that of plasmid p2016089-170 (accession CP090536), which was previously identified from a *Salmonella* 4,[5],12:i:- strain in China. The pYY-2016-057-2 plasmid carried 19 AMR genes, *sul1*, *qacEdelta1*, *arr-3*, *catB3*, *bla*OXA-1, *aac(6')-Ib-cr*, *aac(3)-IV*, *aph(4)-Ia*, *sul2*, *floR*, *sul3*, *qacL*, *ANT(3")-IIa*, *cmlA1*, *aadA2*, *AAC(3)-Ib*, *dfrA12*, *oqxA*, and *oqxB* (Fig. 4C). Compared with p2016089-170, pYY-2016-057-2 acquired *sul3*, *qacL*, *ANT(3")-IIa*, *cmlA1*, *aadA2*, *AAC(3)-Ib*, and *dfrA12*.

**Pangenome analysis**
Pangenome analysis based on the 3951 isolates identified a total of 41,024 genes, including 1821 core genes, 1453 soft-core genes, 2094 shell genes, and 35,636 cloud genes (Fig. S10, Fig. S11). Among the

accessory genes, 89 specific genes that are significantly associated with the S. Typhimurium lineages with P values below 1e-300 were selected, of which 87 were distributed in the rough isolates in Lineage L6.5 (including *addA*, *alx_2*, *dam_4*, *hha_2*, *hipA*, *hns_2*, *pcoC*, *repB_3*, *smc*, *stiP*, *yceD_2*, *yceD_3*, *yceD_5*) and 74 were hypothetical protein genes, while the remaining two genes, *dsbD_2* and *dcuA_2*, were distributed in the smooth isolates in other lineages (Fig. S11A, Supplementary Data 3). Notably, 78 of the significantly associated accessory genes such as *alx_2*, *hha_2*, *hipA*, *hns_2*, *repB_3*, *smc*, *stiP*, *yceD_2*, *yceD_3*, *yceD_5* were located in the pYY-2016-057-2 plasmid (Fig. 4C). Clusters of Orthologous Groups (COG) functional classification showed that 29 genes mapped to 32 COG clusters, which are mainly involved in cell cycle control, cell division and chromosome partitioning, signal transduction mechanisms, coenzyme transport and metabolism, and replication, recombination, and repair (Fig. S11B).

**Genome-wide association study (GWAS)**
GWAS analysis identified a total of 72 SNPs significantly ($-\log_{10}(P) > 200$) associated with the rough colony variants, of which 65 were located within protein-coding regions and seven within intergenic regions. Among the coding region SNPs, 31 (43.1%) resulted in missense mutations, one (1.4%) was a nonsense mutation, and 33 (55.6%) were synonymous variants (Fig. S12A, Supplementary Data 4). COG analysis showed that the nonsynonymous SNPs affected 31 genes in 35 COG function classes, and these were mainly associated with amino acid transport and metabolism, transcription, cell wall/membrane/envelope biogenesis, inorganic ion transport and metabolism, and intracellular trafficking, secretion, and vesicular transport (Fig. S12B, Supplementary Data 4).

**Multi-omics analysis of the rough and smooth colony strains**
To explore the potential genetic variations mediating morphotype conversion, multi-omics, including genomic and proteomic sequencing, was performed based on the 20 rough and 20 smooth selected colony organisms (Supplementary Data 5). WGS analysis showed that the 20 rough colony isolates clustered together with some differential core SNPs compared to the 20 smooth colony organisms (Fig. 5A). No obvious recombination events were identified among the 20 rough and 20 smooth colony isolates. Data-independent acquisition mass spectrometry (DIA-MS) proteomic analysis showed that a total of 2998 proteins were shared by the rough and smooth colony organisms, and 108 and 116 proteins were identified to be unique to the rough and smooth colony isolates, respectively (Fig. S13A, Supplementary Data 6). A total of 424 differential proteins were identified in the rough colony isolates versus the smooth colony isolates, among which 326 showed increased abundance and 98 showed decreased abundance (Fig. S13B, Supplementary Data 7). Heatmap analysis exhibited clear hierarchical clustering of these proteins based on the abundance correlated with rough or smooth colony morphotypes (Fig. S13C). Gene ontology (GO) and Kyoto Encyclopedia of Genes and Genomes (KEGG) pathway analyses indicated that ATP-binding cassette (ABC) transporters, flagellar assembly, bacterial chemotaxis, biofilm formation, Glyoxylate and dicarboxylate metabolism, and Pyruvate metabolism were among the top 20 KEGG pathways, and these are involved in the functions of signal transduction, membrane transport, cell motility, and cellular community (Fig. S13D, Supplementary Data 8). In the biofilm formation pathway, the abundance of CsgD, CsgC, FliA, and FlgM were upregulated, and those of CsrA and RcsB were downregulated (Fig. S13E). In the bacterial chemotaxis pathway, the abundance of methyl-accepting chemotaxis proteins (MCPs), including Tsr, Tar, Trg, and MglB, and other chemotaxis proteins, including FliGMN, were upregulated (Fig. S13F). In the flagellar assembly pathway, the abundance of FliACDGKLMN and FlgBCDEGIKLM was upregulated (Fig. S13G).

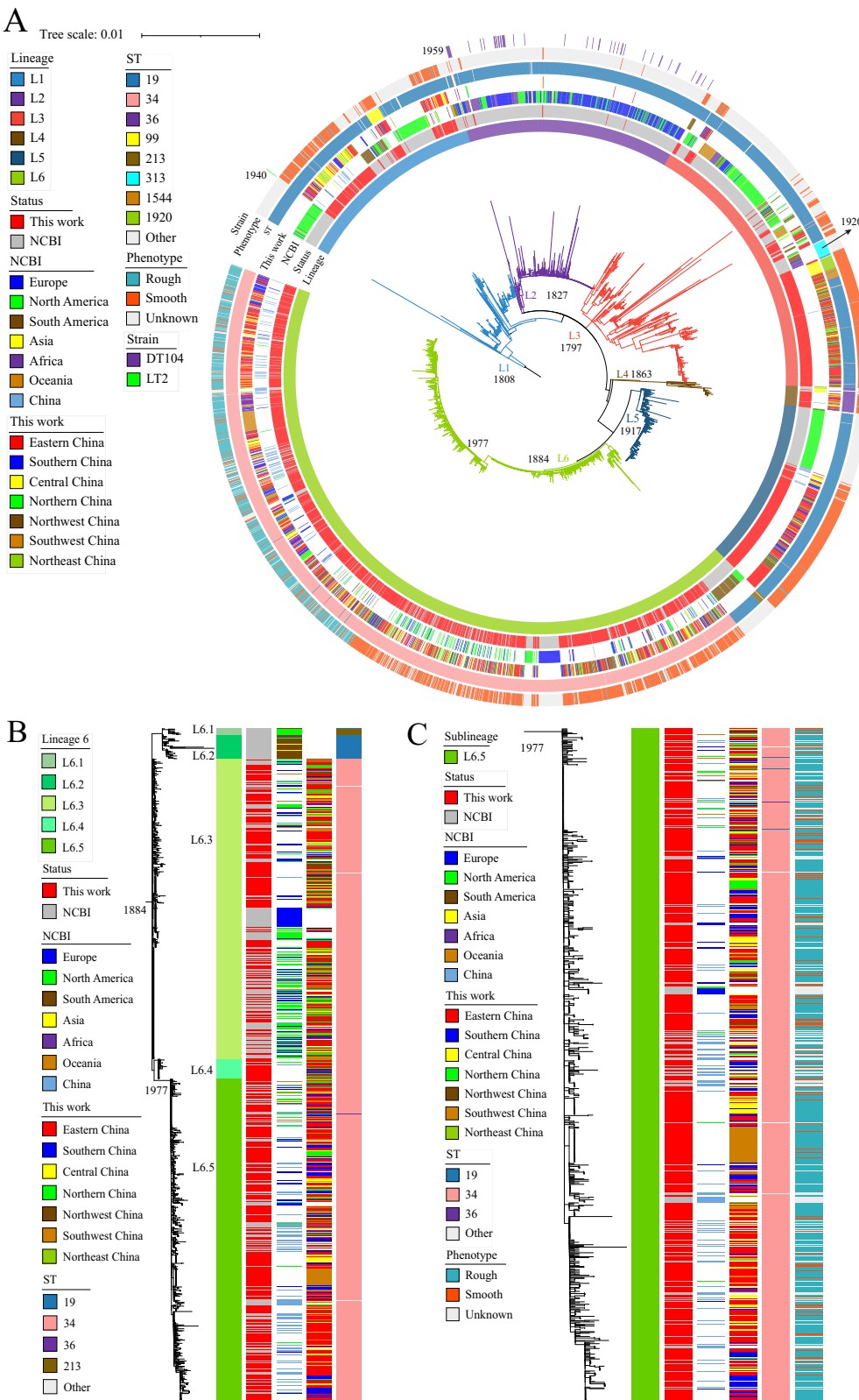

**Fig. 3 | Population structure of the 3951 *S.* Typhimurium genomes analyzed in this study. A** Maximum likelihood phylogeny was performed on 3,951 global *S.* Typhimurium genomes, including the 2212 Chinese strains in this study (marked in red) and 1739 public genomes from the NCBI (marked in gray). The population structure was estimated using hierBAPS, and lineages or sublineages are labeled with different colors as LX or LX.Y, where X and Y are the lineage and sublineage

numbers, respectively. The rings, from inner to outer, labeled with different colors indicate the lineages, sublineages, geographical origin, isolation regions, sequence types, and colony phenotypes. **B** Maximum likelihood tree of *S.* Typhimurium Lineage L6. **C** Maximum likelihood tree of *S.* Typhimurium sublineage L6.5. Divergence dates are given on the major nodes in the tree. Source data are provided as a Source Data file. ST sequence type.

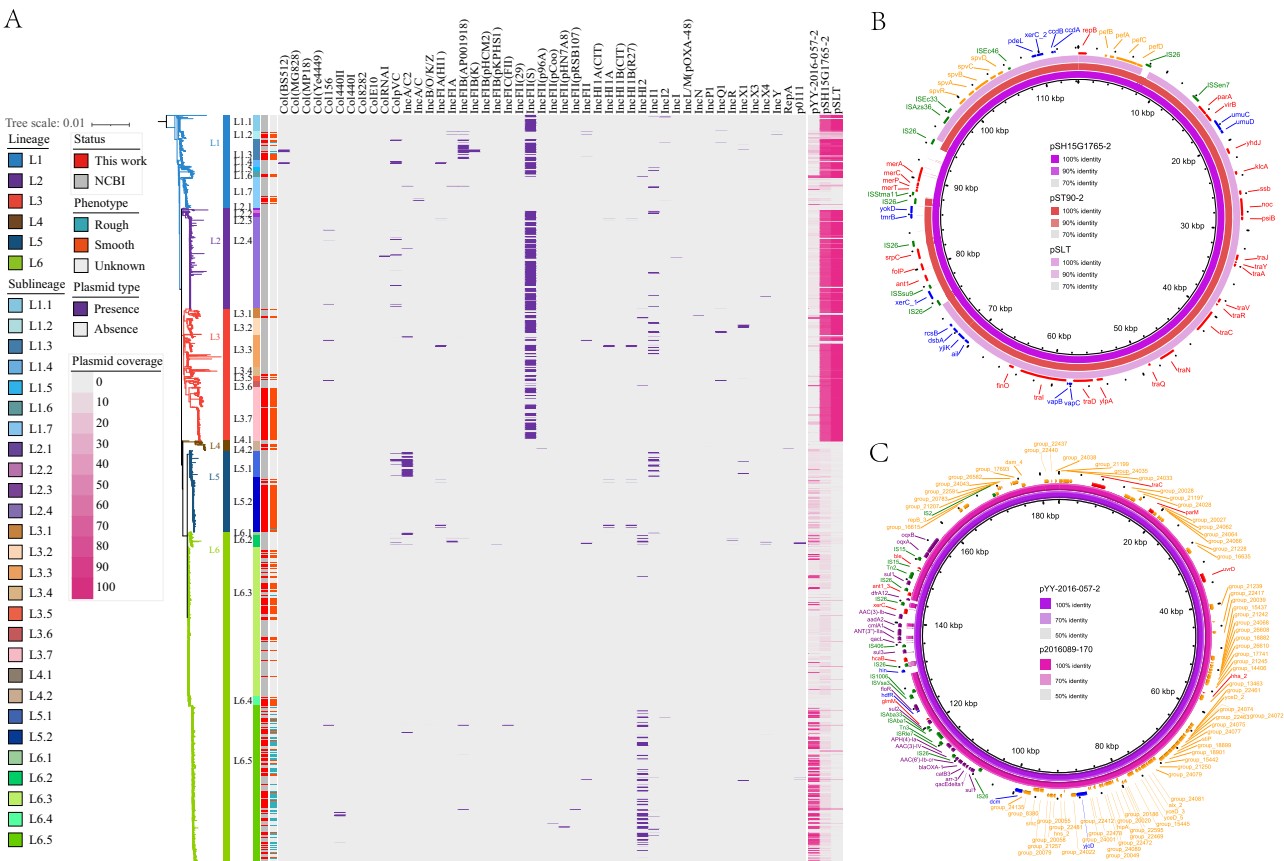

**Fig. 4 | Profiles and comparisons of plasmids associated with antimicrobial resistance and virulence factor genes. A** Plasmid profiles among the global *S.* Typhimurium isolates. The bands to the left of the heatmap indicate the lineage, sublineage, source, and colony morphology of the strains. Purple stripes represent the presence of different plasmid replicons, and red stripes represent the presence of the complete plasmids identified in this study. A total of 47 plasmid replicons were predicted, and two distinct plasmids harboring antimicrobial resistance or virulence factor genes were identified by a combination of Nanopore long-read sequencing and Illumina short-read sequencing. **B**, **C** Pairwise comparisons of two

distinct, complete plasmids identified in this study. The plasmid pSH15G1765-2 belonging to the IncFII(S) replicon (114 kb in size) is highly similar to the plasmid pST90-2 (accession CP050736) identified from a Chinese *S.* Typhimurium strain in 2012 and the virulence plasmid pSLT (accession AE006471) from *S.* Typhimurium strain LT2. pYY-2016-057-2, belonging to the IncHI2 replicon (183 kb in size), is similar to a previously reported plasmid p2016089-170 (accession CP090536) identified from a *Salmonella* 4,[5],12:i:-strain in China. Source data are provided as a Source Data file.

To verify the reliability of the proteomic data, the mRNA abundance of the *csgBAC-csgDEFG* operons was detected, and the results showed that the rough colony isolate SF175 had notably increased mRNA levels of the *csgDEFG* operons compared with the control strain ATCC 14028 and the smooth colony isolate 2015114, which is consistent with the proteomic data (Fig. S13H, Supplementary Data 9). According to the genomic and proteomic analyses, it can be inferred that biofilm formation may be associated with the mrdar morphotype.

### The *csgD* knockout mutant is unable to form the mrdar morphotype

To further determine the molecular mechanism conferring mrdar conversion, we investigated whether the majority of the global *S.* Typhimurium populations, irrespective of having the rough or smooth colony morphotype, contained biofilm formation-associated genes, including the *csgBAC-csgDEFG* operons, the *bcsABZC-bcsEFG* operons, and flagellar genes, and shared identical gene sequences (Fig. 5B). However, when comparing the intergenic region between the *csgBAC-csgDEFG* operons of the rough and smooth colony isolates, a distinct point mutation (−44G > T) was identified in the promoter of the *csgD* gene. This mutation was widely distributed in the rough colony isolates, with a prevalence rate of 95.3% (623/654) (Fig. 5A, B). Notably, among the rough colony strains of the L6 lineage, 98.8% (618/625)

carried this point mutation. To further determine the function of the *csgD* gene and its promoter mutation in conferring rdar conversion, we constructed a *csgD* knockout mutant (SF175-Δ*csgD*) and a single point mutant with −44 T > G substitution in the *csgD* promoter (SF175-*csgD* with −44 T > G) (Supplementary Data 9). We observed that these two mutants were unable to form rdar colonies, showing substantially reduced biofilm formation at 28 °C and 37 °C, in contrast to the wild-type SF175 isolate (Fig. 2, Fig. S1). These results indicated the important regulatory role of the *csgD* gene and its promoter, and a single point mutation in the promoter of *csgD* can influence the CsgD-regulated rough colony and biofilm formation.

## Discussion

The major finding of this study is the identification of a macro-rdar colony morphology of *S.* Typhimurium that emerged and became prevalent in China. We provided a comprehensive description of the genomic and proteomic characteristics of these mrdar variants and observed that most of them clustered in a novel sublineage on the phylogenetic tree. The typical mrdar morphotype was first described by Römling et al. during their investigation of the multicellular behavior of two unrelated *S.* Typhimurium strains[27]. They observed that these two strains spontaneously developed this phenotypic variation during laboratory passage and identified mutations in the *csgD* promoter as the underlying cause. Specifically, one strain had a single T

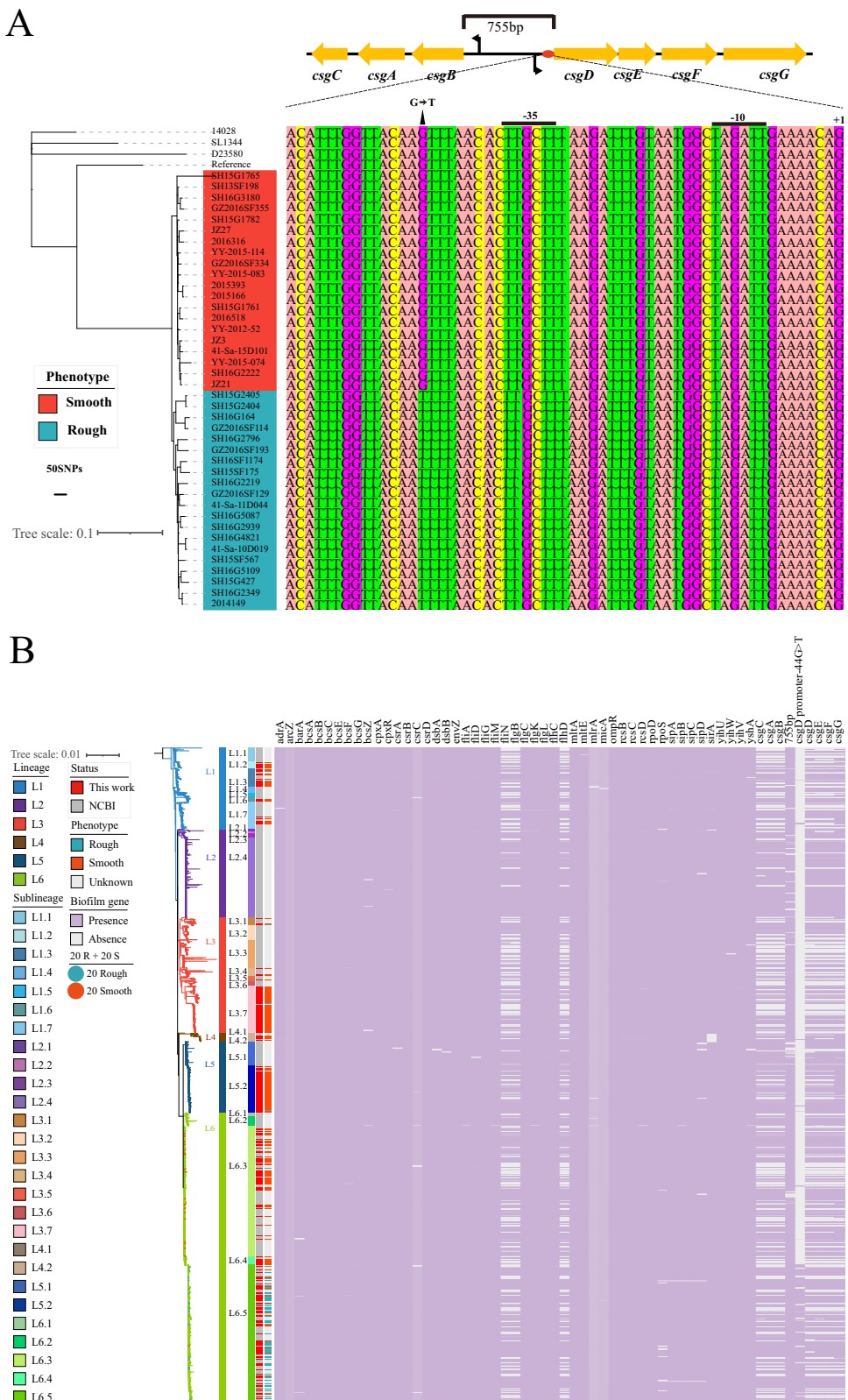

**Fig. 5 | Maximum likelihood phylogeny of the global *S.* Typhimurium isolates with a heatmap showing the distribution of biofilm formation-associated genes or mutations. A** Maximum likelihood phylogeny of the 20 rough and 20 smooth *S.* Typhimurium isolates with a heatmap showing the nucleotide mutations of *csgBAC-csgDEFG* operons. The reference genomes *S.* Typhimurium ATCC 14028 (GenBank accession CP043907.1), LT2 (GenBank accession NC_003197), SL1344 (GenBank accession FQ312003), and D23580 (GenBank accession FN424405) are also included. **B** Maximum likelihood phylogeny of the 3,951 global *S.* Typhimurium genomes with a heatmap showing the distribution of biofilm formation-associated genes or mutations. The presence of a biofilm gene or mutation is indicated in purple and the absence of a gene is indicated in gray. The selected 20 rough and 20 smooth *S.* Typhimurium isolates are indicated in blue and red, respectively. Source data are provided as a Source Data file. SNP single nucleotide polymorphism.

insertion after position -17, while the other had a −44 T > G mutation in the *csgD* promoter. In our study, we report a novel lineage of *S*. Typhimurium carrying the −44 T > G mutation in the *csgD* promoter within a wild *S*. Typhimurium population. Notably, this lineage appeared to be spreading not only within China but also across multiple regions of the world.

Our present study demonstrated that wild-type ATCC 14028 formed a typical rdar morphotype at 28 °C but not at 37 °C; thus, the typical rdar morphotype seems to be temperature-restricted[28,29]. However, the mrdar variants identified in this study seem to be more tolerant to temperature variations than ATCC 14028 with the typical rdar morphology and the smooth colony variants, as these display stronger biofilm-forming ability at 28 °C and 37 °C. Furthermore, the mrdar variants were more commonly MDR than the non-mrdar variants, and, of concern, a greater proportion of mrdar variants were isolated from the environment than the proportions of these variants isolated from humans and food. It has been demonstrated that *Salmonella* rdar biofilm formation can protect the attached bacterial cells from desiccation, disinfection, antimicrobials, host immune responses, and other environmental stresses and facilitate their transmission, colonization, and persistence on diverse biotic and abiotic surfaces in the environment and inside hosts[30,31]. These findings indicate that this mrdar variant represents an evolutionary transition to a distinct form of colony behavior by *S*. Typhimurium, which is not influenced by temperature and may possess an augmented ability to adapt to the environment.

This study provides a snapshot of the large phylogeny of Chinese *S*. Typhimurium, especially the mrdar variants within the global phylogeny. Using a global collection of 3,951 genomes, we found that the global collections were divided into six distinct lineages, all of which contained both the Chinese isolates and those from other regions of the world. Within each lineage, the organisms isolated abroad were basal to the Chinese isolate. These results suggest that the contemporary Chinese *S*. Typhimurium isolates were probably sourced from several ancestors circulating worldwide and introduced into China possibly through multiple transmission events. Notably, the majority of the mrdar variants were found in sublineage L6.5, which probably evolved from smooth Chinese isolates in L6. The L6.5 sublineage consisted of 840 strains, including 708 strains tested in this study and 132 strains downloaded from NCBI. Among them, 777 strains carried the −44G > T substitution in the *csgD* gene promoter, accounting for a high proportion of 92.50%. The substitution was also observed in strains collected outside of China, suggesting the potential global distribution of mrdar variants of *S*. Typhimurium carrying the −44G > T substitution in the *csgD* gene promoter. The dates of lineage divergence were provided by Bayesian phylogenetic inference. In this study, reference *S*. Typhimurium LT2 was predicted to date to 1940 (95% CI: 1929–1950) (Fig. 3, Fig. S6) and was isolated in the 1940s[25]. Moreover, the MRCA of *S*. Typhimurium DT104 was predicted to date to 1959 (95% CI: 1948–1969) (Fig. 3, Fig. S6), consistent with the first reports of DT104 strains since the 1960s in England and Wales[14,32]. It is estimated that the current global *S*. Typhimurium population shares an MRCA that existed circa 1696 (95% CI: 1686–1706) and then spread and localized into several circulating lineages. The ancestor of the L6.5 strains was traced back to circa 1977 (95% CI: 1966–1987), showing that the clonal expansion of mrdar variants occurred in recent evolutionary history. The cgMLST analysis showed that the rough colony strains formed distinct clusters separate from other branches, suggesting a considerable genetic uniqueness of the rough colony strains. These findings indicated that the mrdar variants had evolved into a more recently emergent clone and underwent a contemporary localized clonal expansion and global dispersal, thus with potentially great public health significance.

It has been reported that both gene acquisition and genome degradation are linked with the emergence and niche adaptation of several important pathovariants with great public health significance[11,19,20]. For example, the iNTS variants of ST313 *S*. Typhimurium presented with the loss of multi-cellular stress resistance associated with the rdar-negative phenotype, which was caused by the inactivation of BcsG cellulose biosynthetic enzyme and two nucleotide changes (−80A > C, −189A > G) in the *csgD* gene promoter region[19,28,29]. Consequently, the iNTS variants of ST313 *S*. Typhimurium have become more host-adapted to humans and become a frequent cause of bloodstream infections in sub-Saharan Africa. Here, we identified a mrdar colonial phenotype with a stronger ability of biofilm formation, which means better tolerance and resistance[33]. The rdar colonies display a distinct form of colony behavior that allows biofilm formation and enhances *Salmonella* stress resistance in the environment[29]. Acquisition of the rdar biofilm morphotype is a complex process influenced by structural compositions, the bacterial genome, environmental signals, and stress factors. The typical *Salmonella* biofilm matrix is mainly composed of curli fimbriae, cellulose, biofilm-associated protein BapA, and O-antigen capsule[30,31]. Genes for curli production are found in the *csg* operon[29,34]. Divergent *csg* operons, comprising curli biosynthesis machinery (*csgBAC*), transcriptional regulation (*csgD*), and curli assembly machinery (*csgEFG*), are central to curli biosynthesis. CsgD is involved in activating the transcription of both operons and acts as the primary transcriptional activator of the rdar biofilm morphotype[27,28]. In this study we investigated whether the mrdar phenotype strains contained *csg* operons with sequences identical to the smooth isolates, but with a distinct and dominant point mutation (−44G > T) in the promoter region of the *csgD* gene in the mrdar phenotype strains. Data from the proteomics and transcriptomics analysis confirmed that the mrdar variants had a higher abundance of *csgDEFG* operons than the smooth colony strains. In addition, we constructed a *csgD* knockout mutant (SF175-Δ*csgD*) and a single point mutant (SF175-*csgD* with −44 T > G), and the mutants lost the rough colony morphotype. These findings define the key role of the *csgD* gene and the −44G > T mutation in its promoter region in rdar biofilm formation. It is worth noting that previous studies have indicated a strong correlation between *rpoS* and the ability to form biofilms[35,36]. However, in our analysis of proteomic data from 40 strains, we found no statistically significant difference in the abundance of RpoS between rough and smooth colony strains. Out of the 654 mrdar variants confirmed through experiments, only 27 had nonsynonymous mutations in the *rpoS* gene. Furthermore, these 27 strains also had mutations in the *csgD* promoter, suggesting that RpoS was not the cause of the variants in our study. Furthermore, we constructed an *rpoS* knockout strain, which showed a decrease in biofilm formation ability, but still exhibited the characteristics of a rough colony on the modified Swarmagar medium (Fig. S14).

The known biofilm formation pathways include curli expression, cellulose synthesis, BapA secretion, O-antigen capsule, and others[30,31]. In addition to the notable upregulation of the *csg* operon in the mrdar variants, no obvious changes in the abundance of the *bcsABZC-bcsEFG* operons responsible for cellulose synthesis were observed among the rough and smooth colony strains. Furthermore, the *bapA* gene responsible for BapA secretion was downregulated among the mrdar variants. Therefore, the mrdar biofilm formation observed in the present study was seemingly not fully dependent on the typical known biofilm pathways, and there may be other regulatory pathways responsible for rdar biofilm formation. The biofilm formation process includes several stages: surface attachment, growth and maturation of matrix-embedded communities, and finally, biofilm dispersion[37,38]. It has been demonstrated that flagellar motility plays an important role in *Salmonella* biofilm initiation and maturation[39]. Bacteria sense the chemical stimuli and swim toward the nutrient, attaching to the substrate through flagellar motility. Flagella can serve as adhesins, directly promoting surface attachment. In the mature biofilm, motility is repressed, but the flagella could be repurposed as important structural

elements of the biofilm matrix[39,40]. A previous study showed that hyperexpression of the T3SS encoded by SPI-1 facilitates the biofilm phenotype, which is not dependent on established biofilm mechanisms such as curli and cellulose but dependent on the SPI-1 T3SS secreted proteins SipA, SipB, SipC, SopB, SopE, and SptP, which are enriched and present in the biofilms[41]. In this study, the proteomic analysis showed that there were differential proteins among the rough and smooth colony strains, which are mainly involved in biofilm formation, flagellar assembly, the two-component system, ABC transporters, bacterial chemotaxis, and quorum sensing pathways. We observed clear upregulation of the abundance of flagellar-associated proteins, including FliACDGKLMN and FlgBCDEGIKLM; MCPs, including Tsr, Tar, Trg, and MglB; and secreted proteins, including SipABCD, sopB, SopE, SopE2, SptP, and InvG (Supplementary Data 7). Moreover, Hfq, an RNA chaperone, was also upregulated, which is one of the major global regulators that influences gene expression in response to environmental stimuli and plays a diverse role in bacterial physiology, including biofilm formation, motility, and the T3SS[42,43]. Hfq has been identified as a global activator of biofilm formation in *S*. Typhimurium and positively affects *csgD* expression and biofilm formation[31]. According to the above analyses, mrdar biofilm formation may be a complex and rigorous regulatory process involving multiple pathways, thus mediating better bacterial adaptation among mrdar variants.

In addition to gene acquisition and genome degradation, other genetic factors, such as accessory genome elements and SNPs, are considered to be associated with the evolutionary success of the variants of *S*. Typhimurium. Accessory genome elements and loss-of-function SNPs have been demonstrated to be involved in the host adaptation of the iNTS variants of *S*. Typhimurium ST313[19,44]. Here, we performed pangenome and GWAS analyses to identify the accessory genes and SNPs significantly associated with the evolutionary success of the mrdar variants. We found 87 accessory genes and 72 SNPs that are significantly associated with the mrdar variants in sublineage L6.5, and COG functional analysis showed that the accessory genes and nonsynonymous SNP-affected genes mapped to the COG clusters are mainly involved in replication, recombination and repair, signal transduction mechanisms, inorganic ion transport and metabolism, amino acid transport and metabolism, intracellular trafficking, secretion and vesicular transport, and cell wall/membrane/envelope biogenesis (Fig. S11, Fig. S12). Among the affected genes with COG functional clusters, the *hipBA* operon was unique to the mrdar variants (Supplementary Data 3 and 6). HipA, encoding a serine/threonine kinase, is the only toxin that has been validated as a biofilm tolerance factor and is a critical persistence factor that induces bacterial persistence, and the *hipBA* operon, categorized as a toxin-antitoxin module, has been linked to the persistent phenotype[45–47]. The above analyses provide strong evidence to support the adaptive evolution of mrdar variants by the accumulation of genomic signatures, which may be associated with the local establishment and niche adaptation of these variants.

Antibiotic use might have provided the selection pressure that drove the emergence and microevolution of the MDR dominant pathovariants, and the acquisition of AMR is conferred largely by either horizontal gene transfer or chromosomal gene mutations of AMR determinants[2,11,17,19]. The acquisition of AMR determinants may act as a major driving force for the adaptive evolution of MDR *S*. Typhimurium lineages. Here, among the mrdar variants, we observed AMR and a higher rate of MDR, such as the ASSuT and ACSSuT resistance patterns, compared to what was observed among the smooth colony strains. Moreover, several ESBL genes, including *bla*CTX-M, *bla*OXA, and *bla*TEM, macrolide resistance genes, including *mphA*, mef(B), *ermX*, and *ereA2*, and the plasmid-mediated colistin resistance (*mcr*) gene were also detected in the mrdar variants[48], and point mutations (D87N or D87Y) in the *gyrA* gene occurred in most of the mrdar variants. These AMR determinants conferred additional resistance to first-line antibiotics for the treatment of *Salmonella* infections, including third-generation cephalosporins, azithromycin, and ciprofloxacin, and also resistance to colistin, the last treatment option for multidrug-resistant gram-negative bacteria. Although certain resistance genes, such as *mcr-9*, do not necessarily confer polymyxin-resistant phenotypes in *Salmonella*, the majority of these genes are located on plasmids and transposable elements[49]. This facilitates the transfer of these genes to distantly related bacteria through mechanisms such as conjugation, transduction, or transformation, ultimately leading to the dissemination of antibiotic resistance[50,51]. One plausible explanation for the severe MDR of this specific variant is the easy availability and enrichment of AMR determinants, particularly the distinct plasmid pYY-2016-057-2 belonging to the IncHI2 replicon, which carries 19 AMR genes that confer resistance to aminoglycosides, sulfonamides, trimethoprim, phenicol, cephalosporins, quinolones, rifamycin, and disinfecting agents. Of concern, antibiotic usage in China is complex and widespread and has presented a new and serious antibiotic crisis, as severe antibiotic pollution in the environment has been found[52,53]. Therefore, the severe antibiotic pollution in the environment may be another driving force for the emergence of severe MDR, as this variant was present in a higher proportion of environmental strains and showed stronger environmental adaptability.

This study has several limitations. While the global genome dataset was partially obtained from the NCBI database, the colonial phenotypes (rough or smooth) of the isolates from abroad were not determined, and thus, we can only make a preliminary judgment by analyzing the point mutation of the *csgD* promoter and provide some preliminary insights into the global distribution of mrdar variants. Further international epidemiological studies are required to understand the spread of the mrdar variant, assess the selective benefits and drawbacks of the −44 T > G mutation, and elucidate the role of other SNPs identified in the L6.5 lineage. It has been demonstrated that the *csgD* gene and mutations in the *csgD* promoter region are essential for rdar biofilm formation, but the regulation mechanism of the −44G > T point mutation in the promoter region of the *csgD* gene on the phenotype conversion and biofilm formation requires further study.

In conclusion, this study reports a distinct variant of the rdar colony morphology of *S*. Typhimurium that has emerged and become prevalent in China. This variant exhibited a potentially enhanced capacity for environmental adaptation, including a high proportion of MDR and enhanced biofilm-forming ability. Furthermore, the variant strains clustered together to form a novel lineage on the phylogenetic tree. This study also highlights the potential national and international spread of this variant through a large-scale genomic epidemiology study. The acquisition of distinct phenotypic characteristics and genomic changes probably resulted in the emergence of this new lineage and prevalence of this variant, contributing to its competitive advantages and genome adaptation.

## Methods

### Bacterial isolates

A collection of 2212 Chinese *S*. Typhimurium isolates with different spatial and temporal contexts were included in this study (Fig. 1A–C, Table 1, Supplementary Data 1). The *S*. Typhimurium isolates were identified with API 20E biochemical tests (bioMerieux, Marcy l'Etoile, France) and serotyped with a commercial antiserum kit (SSI, Copenhagen, Denmark) following the Kauffmann-White scheme. In cases where autoagglutinating strains could not be serotyped by the Kauffmann-White scheme, retrospective serotype confirmation was performed through WGS. An additional collection of 1739 globally distributed isolates was included, which were obtained from the NCBI database and span six continents (Asia, Africa, North America, South America, Europe, and Oceania) (Fig. 1D–F, Supplementary Data 2). This study was approved by the institutional review board of

the Chinese PLA Center for Disease Control and Prevention, Beijing, China.

### Characterization of the rough and smooth colony morphology of *S.* Typhimurium

For colony morphology characterization, modified Swarmagar medium culturing and Congo red assays were performed (Supplementary Methods). The *S.* Typhimurium isolate ATCC 14028 was selected as a control strain due to its typical rdar colony morphology[28,29]. The rough and smooth morphotypes were also observed through SEM examinations (Supplementary Methods). One rough colony strain (SF175), one smooth colony strain (2015114), and the control strain (ATCC 14028) were selected as representatives in this study.

### Biofilm formation

To determine the biofilm formation ability of the rough colony variants, we selected 20 representative rough colony strains and 20 representative smooth colony strains (Supplementary Data 5) for comparison, and the *S.* Typhimurium isolate ATCC 14028 was used as a control. These 40 representative strains were randomly selected from the L6 strains based on the results of phylogenetic analysis. Biofilm formation was examined by crystal violet assays, pellicle formation assays, and CLSM imaging (Supplementary Methods).

### Antimicrobial susceptibility testing

Antimicrobial susceptibility testing was performed by broth microdilution using a 96-well microtiter plate (Sensititre CMV3AGNF, Trek Diagnostic Systems; Thermo Fisher Scientific, Inc., West Sussex, United Kingdom) according to the recommendations of the CLSI guidelines[54]. Fourteen different antibiotics were included in this experiment: ceftriaxone (CRO), tetracycline (TET), ceftiofur (XNL), cefoxitin (FOX), gentamicin (GEN), ampicillin (AMP), chloramphenicol (CHL), ciprofloxacin (CIP), trimethoprim/sulfamethoxazole (SXT), sulfisoxazole (FIS), nalidixic acid (NAL), streptomycin (STR), azithromycin (AZI), and amoxicillin/clavulanic acid in a 2:1 ratio (AUG2). *Escherichia coli* ATCC 25922 was used for quality control.

### Whole-genome sequencing and genome assembly

In this study, the genomes of 2212 Chinese *S.* Typhimurium isolates were sequenced (Supplementary Methods). The 20 rough and 20 smooth colony strains selected were also submitted for long-read sequencing to obtain their complete genomes (Supplementary Methods). Genomic comparisons were performed on the complete genomes from the 20 rough and 20 smooth colony strains to analyze their genetic differences in chromosomal regions such as SNPs, indels, and rearrangements with BLAST, Snippy (v4.6.0) (https://github.com/tseemann/snippy), and Gubbins (v 2.4.1)[55].

### SNP calling and phylogenetic analysis

A genome dataset that included the 2,212 Chinese strains sequenced in this study and 1739 public genomes from NCBI was constructed (Supplementary Data 1). Among these genomes, the core genome SNPs were identified using the Snippy (v4.6.0) pipeline, and the population structure was estimated using hierBAPS (hierarchical Bayesian analysis of population structure)[56] (Supplementary Methods).

### MLST analysis

MLST analysis was performed by scanning the sequences of seven housekeeping genes (*aroC*, *dnaN*, *hemD*, *hisD*, *purE*, *sucA*, and *thrA*) against PubMLST typing schemes using MLST.

### cgMLST analysis

We used the CHEWBBACA software to construct a local cgMLST core gene database of *Salmonella enterica*, which contained 3000 core

allelic genes[57]. We then ran the AlleleCall function of chewBBACA.py to align the bacterial genomes with the core gene loci and obtain the allelic profiles. We applied the ExtractCgMLST function of chewBBACA.py to filter out the accessory genes based on a presence threshold of 0.99 and obtained a final matrix of 2661 core genes. We used the phyloviz software[58] and the goeBURST Full MST algorithm to construct a minimum spanning tree based on the cgMLST results and show the genetic relationships and the clonal complexes of the *Salmonella enterica* strains[59].

### AMR gene and VF detection

AMR genes were detected using ABRicate (https://github.com/tseemann/abricate) against the Comprehensive Antibiotic Resistance Database (CARD)[60], and thresholds of 80% were selected for minimum coverage and minimum identity. Point mutations in the QRDRs of the *gyrA*, *gyrB*, *parC*, and *parE* genes were investigated using Snippy. The virulence genes were aligned to the VFDB[61] using BLAST (v 2.5.0)[62] with thresholds of 80% for minimum coverage and minimum identity.

### Plasmid detection

Complete plasmid sequences were generated from the 20 rough and 20 smooth colony strains sequenced by Nanopore long-read sequencing in combination with Illumina short-read sequencing (Supplementary Methods). Known plasmid replicons were screened using ABRicate (https://github.com/tseemann/abricate), and the genomic assemblies from Illumina short-read sequencing were mapped against the selected plasmids using BLAST to determine whether the strains carried corresponding plasmids with minimum identity and minimum coverage thresholds of 95%.

### Pangenome analysis

The assembled genomes were annotated with Prokka[63]. Based on the annotated assemblies, a pangenome was constructed for all 3951 isolates using Roary[64]. The Chi-square test selected *P* values below 1e-300 from the accessory genes, and significantly related genes were defined as those present in more than 75% of the isolates in sublineage 6.5 and less than 25% in the other lineages or those present in less than 25% of the isolates in sublineage 6.5 and more than 75% in the other lineages. COG classification of the genes identified by pangenome analysis was performed by mapping to the COG function database using BLAST, followed by visualization using R.

### GWAS analysis

A GWAS was also performed on the 2212 Chinese strains to identify significant SNPs associated with rough or smooth colony morphology (Supplementary Methods).

### Distribution of the biofilm-associated genes and mutations

The typical rdar morphotype is the best-studied *Salmonella* biofilm phenotype[27,30,31]. In this study, 58 biofilm-associated genes, such as curli fimbriae-encoding genes (*csgBAC-csgDEFG* operons) and cellulose synthesis genes (*bcsABZC-bcsEFG* operons), were included to search for their distribution and identify the presence of point mutations by sequence comparisons within the rough and smooth colony strains. In addition, the *csgBAC-csgDEFG* operons and promoters of *csgD* were compared with those of LT2 and ATCC 14028 using the Snippy pipeline (Supplementary Methods).

### Proteomic analysis of the rough and smooth colony strains

To compare the proteome profiles of the rough and smooth colony strains, the selected 20 rough and 20 smooth colony strains were subjected to data-independent acquisition (DIA)-based proteomics sequencing (Supplementary Data 5) (Supplementary Methods). No imputation method was used in this analysis. Differential proteins were considered if the fold change between the rough strains and

smooth controls was >1.5 or <0.67 and the *P* value of the t-test was <0.05. The differential proteins were then analyzed by hierarchical clustering with Cluster 3.0 (http://bonsai.hgc.jp/mdehoon/software/cluster/software.htm) and Java Treeview software (http://jtreeview.sourceforge.net). Subsequently, the differential proteins were subjected to GO analysis with Blast2GO (https://www.blast2go.com/) and matched against the KEGG database (http://www.kegg.jp/) by the KEGG Automatic Annotation Server (KAAS, https://www.genome.jp/tools/kaas/) to obtain their KEGG orthology identifications. *P* < 0.05 by Fisher's exact test was considered to indicate significance.

Based on the proteomic results, differential proteins, including the *csgDEFG* operons, were selected to assess their mRNA abundance using real-time quantitative polymerase chain reaction (RT–qPCR) assays (Supplementary Methods). In addition, recombination analysis was conducted on the 20 smooth and 20 rough colony strains from the proteomic analysis (Supplementary Methods).

### Construction of the mutants with *csgD* gene knockout and a single nucleotide replacement (−44 T > G) in its promoter

To determine the possible role of the *csgD* gene and its promoter in mediating morphotype conversion and biofilm formation of the mrdar variants, a mutant with the knockout of the *csgD* gene and a mutant with a single nucleotide replacement (−44 T > G) in its promoter were constructed for the rough colony strain SF175 (Supplementary Methods).

### Statistical analysis

Between-group comparisons were performed using the chi-square test. The Kruskal–Wallis test was performed between lineages for the carried AMR genes and VFs, while the Wilcoxon test was performed between rough and smooth phenotype isolates. Other descriptive data are summarized using descriptive statistics. *P* values of less than 0.05 were considered to indicate statistical significance. Statistical analysis was performed using R (v 4.0.3), and statistical plots were visualized using ggplot2 (v 3.3.5).

### Reporting summary

Further information on research design is available in the Nature Portfolio Reporting Summary linked to this article.

## Data availability

The publicly available sequences used in this study are available in GenBank (https://www.ncbi.nlm.nih.gov/datasets/genome/), with accession numbers listed in Supplementary Data 1. The sequencing data generated in this study have been deposited in the NCBI Sequence Read Archive under the BioProject number PRJNA981225. The annotated plasmid sequences have been deposited in GenBank under accession numbers PP852730 and PP852731. The mass spectrometry proteomics data have been deposited in the ProteomeXchange Consortium (https://proteomecentral.proteomexchange.org) via the iProX partner repository with the dataset identifier PXD047483. Source data are provided with this paper.

## Code availability

No custom software or code was developed for this study. All analyses relied on standard statistical and bioinformatics methods and were performed using publicly available tools and packages, which have been appropriately cited in the Methods section.

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

## Acknowledgements

This work was supported by the National Nature Science Foundation of China (no. 82173580 to S.F.Q. and no. 82202538 to Y.X.) and the Chinese Academy of Medical Sciences (CAMS) Innovation Fund for Medical Sciences (2021-I2M-1-044 to J.T.Y.). We would like to thank Chunli Li (Institute of Microbiology, Chinese Academy of Sciences) for sample preparation and SEM examination.

## Author contributions

S.F.Q., J.T.Y., X.B.X., and H.B.S. conceived and designed the experiments. Y.X., Y.W.Z., Y.R.H., X.G.L., X.W., X.P.L., Y.Q.P., C.J.Y., H.b.L., X.Y.D., H.B.L., X.Y.L., H.W., C.W., Y.H., Q.W., H.Q.J., and M.J.Y. performed the experiments. S.F.Q., Y.X., K.P.Z., K.Y.M., Y.W.Z., J.F.L., K.K.L., Y.R.W., Y.J.C., and F.C. analyzed the data. Y.X., Y.W.Z., K.P.Z., K.K.L., Y.R.H., X.G.L., X.W., X.P.L., Y.Q.P., Y.R.W., C.J.Y., H.b.L., X.Y.D., H.B.L., H.W., Q.W., C.W., L.G.W., H.Y.Y., S.B., X.B.X., and J.T.Y. contributed reagents, materials, or analysis tools. S.F.Q., Y.X., K.P.Z., Y.W.Z., and K.Y.M. wrote the paper. All authors reviewed the manuscript critically for content and approved the decision to submit it for publication.

## Competing interests

The authors declare no competing interests.

## Additional information

[1]Center for Disease Control and Prevention of Chinese PLA, Beijing, China. [2]Kaifeng Center for Disease Control and Prevention, Kaifeng, China. [3]State Key Laboratory of Common Mechanism Research for Major Diseases, Institute of Basic Medical Sciences Chinese Academy of Medical Sciences, School of Basic Medicine Peking Union Medical College, Beijing, China. [4]Daxing Center for Disease Control and Prevention, Beijing, China. [5]State Key Laboratory of Pathogen and Biosecurity, Beijing Institute of Microbiology and Epidemiology, Beijing, China. [6]CAS Key Laboratory of Genome Sciences & Information, Beijing Institute of Genomics, Chinese Academy of Sciences, China National Center for Bioinformation, Beijing, China. [7]Department of Epidemiology, School of Public Health, Zhengzhou University, Zhengzhou, China. [8]University of Cambridge School of Clinical Medicine, Cambridge Biomedical Campus, Cambridge, United Kingdom. [9]Shanghai Municipal Center for Disease Control and Prevention, Shanghai, China. [10]These authors contributed equally: Ying Xiang, Kunpeng Zhu, Kaiyuan Min, Yaowen Zhang, Jiangfeng Liu. ✉e-mail: xxb72@sina.com; yangjt@pumc.edu.cn; hongbinsong@263.net; qiushf0613@hotmail.com

