## [Peer Review File · Nature Communications]

REVIEWER COMMENTS

Reviewer #1 (Remarks to the Author):

The authors describe the characterization of a new clade of *Salmonella* Typhimurium strains that exist and were isolated in China. These strains have a macro 'rdar' morphotype colony morphology, suggesting overproduction of curli, cellulose and other extracellular matrix components for biofilm formation. The authors link this new colony morphotype to a mutation in the *csgD* promoter, leading to increased *CsgD* production and production of the downstream pathways. The authors speculate on the reasoning of the *csgD* promoter mutation and the potential advantages that these strains might have for environmental survival and persistence. In addition, the macro rdar strains have increased antibiotic resistance compared to smooth variants in the same clades, primarily due to the presence of large plasmids. Overall, it is a very thorough, complete and interesting study. However, the authors need to reduce the amount of speculation and stick to their results, as well as incorporate a wealth of literature and some further testing that they seemingly have missed.

Major Points

1. The most important problem with this manuscript is summarized in their Discussion statement in Lines 392-395 – the words 'an evolutionary transition to a novel kind of multicellular behavior that is independent of temperature, with a potentially enhanced capacity for environmental adaptation'

- This promoter mutation is not novel – the same -44 G-T mutation was first described by Romling et al., 1998 (<https://doi.org/10.1046/j.1365-2958.1998.00791.x>), and was essentially characterized to explain the differences between *S. Typhimurium* 14028 (the control strain used by Xiang et al) and *S. enterica* serovar Enteritidis 27655-3b (the strain analyzed by William Kay's lab for description and purification of curli; <https://doi.org/10.1128/jb.173.15.4773-4781.1991>). 27655-3b was a wild-type strain of *S. Enteritidis* that came from a severe gastroenteritis outbreak in India
- Further research went on to show that the -44 mutation alleviated dependency on RpoS for transcription and made it so the *csgD* promoter could be transcribed by sigma70, earlier during growth and at all growth temperatures
- It is thought that this type of promoter mutation is in response to the Enteritidis 27655-3b strain being an RpoS mutant (<https://doi.org/10.1111/j.1574-6968.1998.tb13235.x>), meaning the only way that the strain could produce curli and cellulose would be to acquire the *csgD* promoter mutation likely after the *rpoS* mutation took hold in the strain
- The same phenomenon was observed in another *Salmonella* strain (SARC 15), where it had temperature-independent production of curli and cellulose and the strain also had an RpoS deficiency (<https://doi.org/10.1128/jb.00798-06>)

The writing of the Xiang et al. manuscript must be revamped to include these important points. Further, to understand their big speculative points about evolutionary transition and enhanced capacity for environmental adaptation, they must do some type of test on the rpoS status of their rough and smooth strains, to see if the same connection is found and also due to the central importance of RpoS in combating stress or adapting to environmental changes. The authors could focus this testing on the 40 strains that were analyzed in more detail in Figure 5.

The 'enhanced capacity for environmental adaptation' it is a very complex phenomenon – we still don't know how biofilms benefit environmental survival and persistence (no hard 'real-world' evidence) and the promoter mutation Xiang et al. describe that causes over production of curli and cellulose may not even be a benefit to these Salmonella Typhimurium strains. A lot of testing would need to be performed to reach these conclusions. The boost in CsgD activity due to sigma70-based transcription would also have large consequences as shown by Xiang et al in their proteomics analysis.

The authors need to stick to what their data says – clearly these mrdar strains can persist and have persisted in China, likely transmitted from the first version that caused human infections in China, and they are associated with increased antibiotic resistance due to the presence of those large AMR plasmids. The speculation about 'increased' adaptation compared to other strains needs to be reserved to direct research on survival comparing smooth and rough strains of the same lineage, kind of like what has been done for iNTS ST313 strains versus their gastroenteritis-causing ST19 relatives (<https://doi.org/10.1093/femspd/ftw049>)

2. Lines 363-364 – the way the authors have written about this data it suggests that not all mrdar strains carried the csgD promoter mutation at -44 position. If this is the case, the authors need to list the % of mrdar strains that carry this mutation. I would expect that all mrdar strains carry the -44 promoter mutation (as shown for the 20 strains analyzed in Figure 5A) but the authors need to explain this point better.

3. Lines 542-548 – these few sentences are highly speculative – talking about superbugs etc. and a 'major threat to global public health' and I recommend taking them out – there is nothing to suggest that the Typhimurium strains they have characterized are any more virulent than normal gastroenteritis-causing strains – and antibiotics are not typically prescribed for gastroenteritis infections caused by Salmonella – because it prolongs shedding – the increased AMR is more likely related to what the authors have described in Lines 537-539

4. Figure clarity. The authors need to strongly consider how best to represent their data and perhaps re-organize some of the Figures.

- in multiple Figures (4,5) and Supplementary Figures (5, 8, 10) when I zoomed in to see what genes are in each column of the heat map the text got too blurry to be able to read what they are. The text in the main Figures was too small to read in many instances and if the resolution is low in the included Figures, then what is the point of listing the gene names or plasmid names across the top?
- Figure 3 was only clear to follow and understand when displayed as a full-page, but since this is not possible in the printed version, the authors need to think about what is most important to show. For example, maybe it is best to blow up the circular tree in part A to be full-page width and place the linear trees in B and C below.
- Figure 5A – it wasn't possible to read the promoter sequence that is shown here. This would be better at more of a full-page width.

Minor Points

1. Lines 58-59 – these numbers for prevalence of Salmonella gastroenteritis are old, please refer to a newer data, such as: Kirk et al. PLoS Medicine 12, no. 12 (2015): e1001921.
2. Lines 112-115 – The wording is confusing – ‘this variant’ was mostly recovered from humans (64.1%), food (27.2%), and environment (8.7%), but the proportion of the rough colony variant from the environment was higher (40.7%)
 - The authors need to explain more here – I thought ‘this variant’ meant the rough colony variant, so mentioning them both in the same sentence is confusing. Please explain exactly what you mean.
3. Lines 185-186 – ‘were also widely distributed among the six lineages but were mainly distributed in Lineage L6’ – the sentence is a bit contradictory and should be clarified
4. Lines 209-211 – The authors need to use caution in making this statement – there could be other variants in other areas of the world that spread the same way
5. Line 278 – saying these types of plasmids all belonged to the virulence plasmid pSLT – you mean all plasmids belonged to “the same group as pSLT?”
6. Line 324 – authors are talking about differential core SNPs compared between 20 rough colony variants and 20 smooth colony variants and refer to Figure 5A – Figure 5A is the blow up of the sequence of the *csgD* promoter region – I had a feeling like Figure 5 had been changed from what was described in the main text
7. Lines 324-325 – The authors need to explain better what ‘obvious recombination events’ are, what type of analysis was done and how was this conclusion was arrived at
8. Line 460 – the *yihU-yshA* and *yihVW* operons may not in fact code for an O-antigen capsule – see the following literature (doi:10.1038/nature12947; doi:10.1038/nchembio.2023) – Therefore, I recommend leaving this sentence out

9. Lines 472-477 – This extra information regarding chemotaxis could be left out – the authors do not investigate anything regarding the different chemotaxis signals – or form any conclusions

10. Lines 567-571 should be left out

11. Supplementary Methods – Congo Red Assay - “one colony of each strain was spotted on LB agar’ – the authors need to explain in more detail – how can a colony be spotted? – I assume that a specific volume (1-4 uL?) of the overnight culture was pipetted onto the agar and allowed to dry – was the inoculum size standardized for # of cells? for example, was the OD of the overnight culture normalized in any way?

Reviewer #2 (Remarks to the Author):

The present manuscript shows a huge scientific work, but their analysis and the conclusions drawn would need to be reconsidered in order to get their study publishable.

The authors claim to describe a new variant of *S. Typhimurium* with a distinct rough morphology and potentially enhanced environmental adaptation in China.

But regarding their findings, it does not seem to be one variant, but a morphotype shown by bacteria belonging to different lineages inside the serotype, more prevalent in recent isolates of the monophasic variant of *S. Typhimurium* (ST34) isolated in China, mainly from humans. But there are strains belonging to other ST types. So no variant.

MAJOR COMMENTS

1. The authors talk about a new “variant”, or “new variants” (in plural), a new “lineage”, a new “sublineage”, and a new “subpopulation”. All those terms have very precise and different meanings in phylogeny and taxonomy. The authors should better delimit what they actually describe in this manuscript.:

a) A new lineage with a particular genomic content and a particular phenotype, emerging in China.

b) The particular morphology (and capacity of biofilm formation) shown by different strains, which is particularly predominant in a recent clade, prevalent in China in recent years, but also shown by strains belonging to other distant lineages.

2. As in RESULTS line 107: “We identified a novel variant of *S. Typhimurium* with a rough colony morphology”.

What the authors have observed is a high number of isolates that show that particular morphology. But not being all of them genetically clustered together and separately from any other (smooth) isolate, that is not a new variant. It is a particular morphology that is frequently observed in Salmonella, particularly when strains are subcultured several times.

If the authors are trying to define a newly emerged subpopulation of ST34, with a particular phenotype (rough aspect and enhanced biofilm formation), it would be very interesting to add cgMLST information (Enterobase), as the scheme is universally used and could give a known reference to their distances in SNP, that mainly depend on the selected collection.

MINOR COMMENTS

3. ABSTRACT line 39: what the authors recovered from publicly available databases are “genomes”, not “isolates”. This is of importance when describing new phenotypes, as they cannot be known for the genomes exclusively.

4. The authors describe a “rough” lineage (or variant, or subpopulation), that they serotyped by agglutination with sera from SSI. This is confusing, as usually the term “rough” is associated to the impossibility to sero-agglutinate, as the strains are self-agglutinating.

Most of the MRDAR isolates in the study belong to ST34, and the authors claim having performed seroagglutination following the Kauffmann-White scheme. However, they don't specify whether the isolates in their study were monophasic, as it is usually the case for ST34.

5. ABSTRACT line 42: The authors use both terms « expansive multidrug resistance » and « extensive multidrug resistance ». The term generally used is « extensive multidrug resistance ». This reviewer would suggest the authors to use the later, and to describe what they exactly mean by that term the first time that it is used, as there is no universal consensus and different authors will mean different things with that.

6. The authors identify 6 lineages of Salmonella Typhimurium among the > 3 000 Chinese and international genomes analysed. And they determine that most of the MRDAR genomes belonged to L6, and particularly to L6.5, as a very clonal population.

— The authors cannot know whether there are many more MRDAR isolates in the other lineages, where the majority of genomes have been collected from other countries: they only studied the isolates from China, where most of them belonged to L6.5. Thus, ABSTRACT line 44-45: “... the majority of mrdar variants OBSERVED IN THIS STUDY in sublineage L6.5,...”

— How do the authors explain the rest of MRDAR, outside L.6?

— There seems to be very little variability inside L.6.5, if compared to other lineages. Have the authors analysed a possible outbreak, regarding those isolates? Have the authors compared the cgMLST profiles of their isolates and those recovered from NCBI, by the HierCC Enterobase scheme? That could be very helpful to give an idea of proximity.

— The authors infer dates of appearance of the different lineages in the phylogeny analysis. Also, they use the dates of collection known for the Chinese isolates in previous graphs. But it would be most interesting to show the collection dates in the phylogeny tree. That could perhaps help to identify (or to exclude) an outbreak that might explain the clonal expansion of the RDAR ST34 L6.5 subpopulation they describe here.

7. As the authors aim to describe a particular, stable capacity of forming biofilm of the new variant they are describing, Figure S1-D and E should show the 20 SMOOTH and the 20 ROUGH isolates that the authors claim having tested. Biofilm formation can be very variable and so if the 20 isolates of each morphology behaved the same way, it would be most important to show it.

8. Figures S1-A, B and C claim to be a representation of experiments repeated in triplicate. As biofilm formation is usually quite variable, it would be good if the figure included the results of the three times the analysis was done, with their error bars for each isolate.

9. The Heatmap in figure 5-B shows a clear difference in the presence/absence of the mutation in *csgD* promoter.

— Did the authors identify the exact sequence of the promoter, or they refer to as so to the several base pairs prior to the genes CDS?

— Did they authors test the biofilm formation on isolates from the other lineages, when the mutation is not present, to prove its implication in the particular biofilm formation of the present variant?

Reviewer #3 (Remarks to the Author):

Within this study from Xiang et al. this team documents a novel *Salmonella enterica* serovar Typhimurium variant identified within Chinese samples collected over the last 15 years associated with a change in colony morphology. Using genomic approaches, this team assesses the relatedness and presence of virulence factors / AMR genes associated with the observed rough strains revealing the enrichment of these strains within the L6.5 sub-lineage. Using GWAS the team identifies a potential driver of the change in colony morphology and then using proteomic, phenotypic and mutagenesis experiments provide preliminary evidence for the involvement of the *csgDEFG* operons and specifically a single SNP in the promoter of *csgD* for this alteration. While the study is clearly a large body of work it does feel disjointed, and some sections do not appear to provide any additional progression in the narrative such as the sections on biofilm formation or the AMR profiles. Additionally, significantly more care is required in the description of the methods in particular the details provided on the proteomic analysis are grossly insufficient. Due to this I feel this study requires additional work to further support the claims (such as that these variants have increased stress resistance) meaning in its current form this work is unsuitable for publication.

Major concerns:

Arguably the first half of the results sections (line 100 to 307) could be re-written and streamlined as these are very descriptive summaries of the plasmids, virulence genes and AMR genes observed across strains but do not provide insights into the key biological question being asked, which is what the cause of the morphological change in the novel *S. Typhimurium* variants is.

The introduction is extremely brief and doesn't provide context on the enormous impact genomic analysis has had on *Salmonella* tracking or our understanding of the emergence of virulent *Salmonella* variants. This should at least be re-written to help put this work in context.

The methods provided for the proteomic analysis are grossly insufficient with no details on how strains were grown, how the protein lysates were prepared or any details on the LC parameters (column used, gradient length, flow rate buffers etc) or MS instrument settings (DIA window width, maximum fill time, AGC, resolution etc) used provided. Additionally, further details are needed on how the DDA fractions were prepared (was this by basic reverse phase fractionation? how many fractions? What column was used for sample fractionation, was the sample used to create the DDA library a pooled reference of all the strains? On the bioinformatics associated with this, what were the search settings used within Spectronaut? As 40 different strains were used with 40 different proteomes were individual proteome built from the sequencing data and searched together or was a single reference strain used? was a minimum number of peptides required for consideration for analysis? if imputation was used how was it undertaken? Please provide details on the statistical test used to assess changes, was this done using a t-test and was multiple hypothesis correction applied? For the GO enrichment analysis of the proteomic data while it is stated in the main body methods that Fisher's exact test were used no output (supplementary tables) are provided with these results. Likewise, the Spectronaut search output from the proteomic experiment (all 2,998 proteins) should be provided not just a summary of the results. Additionally, all raw proteomic data must be uploaded to a centralised database, currently no details are provided confirming the submission of this dataset to such a repository.

The methods provided for the construction of the *csgD* knockout promoter point mutation are insufficient. Please provide the name of the vector and reference for this CRISPR-BTM system as well as a reference for the pH73sacB vector. It should be noted the provided reference, reference 26 the authors highlighted for the CRISPR-BTM mutagenesis approach (also a Nature publishing group publication) lacks details on this vector so the statement "as previously described" is false.

Line 216 to 244 the discussion of the AMR determinants is very descriptive without any deeper analysis. Wouldn't a better use of this data be to combine it with the AMR profile data from Line 150 to 169 and use those direct assessments of resistance to confirm the function of AMR determinants and/or reveal AMR determinants being missed using the genomic analysis.

Line 433/434 the author state "we identified an mrdar colonial phenotype with better multicellular stress resistance." No stress assays are undertaken within this work to support this claim.

Minor concerns:

Line 107 the author state "We identified a novel variant of *S. Typhimurium* with a rough colony morphology." Please provide an image to show this morphology compared to a typing strain on a LB plates and/or the selective media used for the initial identification of *S. Typhimurium* used within this study.

Line 112/113 the author state "Moreover, this variant was mostly recovered from humans (64.1%), followed by food 113 (27.2%) and the environment (8.7%)" is the difference in the rate of rough colonies observed in human statistically significant?

Line 117 /118 Please define or provide context on why the morphology of colonies was assessed using Luria-Bertani (LB) agar containing Congo red and Coomassie blue without salt at 28C?

Line 138 the author state "noticeable multicellular behavior." Please re-write this statement as the use of the word multicellular is confusing when referring to bacteria.

Line 173 the authors state "single nucleotide polymorphisms (SNPs) identified from the 3,951 genome dataset." More details should be provided here to make it clear the authors are referring to their dataset supplemented with 1739 publicly available genomes. Also, a discussion is needed on why these 1739 publicly available genomes were selected.

Line 377, please avoid the phrase "multicellular behavior".

Line 423 to 425. The authors state “Despite the burden of *S. Typhimurium* infections on public health systems in China and worldwide, relatively little is known about the genomic changes that contribute to the emergence and microevolution of pathovariants.” This is not true *Salmonella* is arguably the best understood human pathogen and many groups have made significant strides in understanding the microevolution of pathovariants which you cite in the proceeding sentences.

Line 448 / 449 the authors state “The data from the proteomics analysis and mRNA expression experiments confirmed that the mrdar variants had higher expression levels” re-word as proteomics does not assess expression it assesses protein abundance.

Line 487 what is meant by “promo1 secreted proteins”

Figure 1A) Please increase the size of the pie charts shown as some are unreadable (for example inner monogolia). Also, each region colored should correspond to a single pie chart, some have multiple pie charts which makes this hard to understand.

Figure 1B and C) increase font size and provide percentage of rough colonies on the figure so readers can independently assess these details.

Figure 2 C/D. Within the images shown for SF175 significantly more cells are shown is this due to a growth difference between the strains? Base on the methods no normalisation based on optimal density was undertaken. Additionally, are these images representative of multiple biological replicates? It is unclear from the method how many times this assay was undertaken.

Figure 3A is very complex this could be simplified to improve readability, for example does color coding the Chinese regions the strain in this study come from add anything? Additional all the sublineage details are provided yet this is not discussed therefore could be removed to improve readability.

Crystal violet assays. Within Supplementary Figure 1, the data from Crystal violet assays are given the y axis states at OD600 has been used, typically 570 nm is used. Is there a rationale why an alternative OD was used?

Responses to Reviewers' Comments

We sincerely appreciate the reviewers for their valuable and constructive comments. We have carefully incorporated the reviewers' advice in the revised version and, as a result, the quality of our manuscript has improved substantially.

Below, we explain in detail how we addressed each comment in the revision of our manuscript. The reviewers' comments are presented in bold, followed by our responses to these comments. All changes have been tracked in the revised manuscript. We also provide a clean version of the revised manuscript, and the line numbers referenced in our responses correspond to this clean version.

Responses to Reviewer #1

The authors describe the characterization of a new clade of Salmonella Typhimurium strains that exist and were isolated in China. These strains have a macro 'rdar' morphotype colony morphology, suggesting overproduction of curli, cellulose and other extracellular matrix components for biofilm formation. The authors link this new colony morphotype to a mutation in the csgD promoter, leading to increased CsgD production and production of the downstream pathways. The authors speculate on the reasoning of the csgD promoter mutation and the potential advantages that these strains might have for environmental survival and persistence. In addition, the macro rdar strains have increased antibiotic resistance compared to smooth variants in the same clades, primarily due to the presence of large plasmids. Overall, it is a very thorough, complete and interesting study. However, the authors need to reduce the amount of speculation and stick to their results, as well as incorporate a wealth of literature and some further testing that they seemingly have missed.

Response: Thank you for your thorough review and valuable feedback on our study. We

are pleased to hear that you find our research interesting and complete. We have taken your suggestions into consideration and conducted additional experiments and testing to provide further support for our conclusions and minimize speculation. Moreover, we have reviewed the literature you suggested, as well as other relevant literature, to ensure that we have included all the relevant references in the manuscript. Please find our point-by-point responses below. Thank you again for your thoughtful and constructive comments.

Major Points

1. The most important problem with this manuscript is summarized in their Discussion statement in Lines 392-395 – the words ‘an evolutionary transition to a novel kind of multicellular behavior that is independent of temperature, with a potentially enhanced capacity for environmental adaptation’

• **This promoter mutation is not novel – the same -44 G-T mutation was first described by Romling et al., 1998 (<https://doi.org/10.1046/j.1365-2958.1998.00791.x>), and was essentially characterized to explain the differences between *S. Typhimurium* 14028 (the control strain used by Xiang et al) and *S. enterica* serovar Enteritidis 27655-3b (the strain analyzed by William Kay’s lab for description and purification of curli; <https://doi.org/10.1128/jb.173.15.4773-4781.1991>). 27655-3b was a wild-type strain of *S. Enteritidis* that came from a severe gastroenteritis outbreak in India**

• **Further research went on to show that the -44 mutation alleviated dependency on RpoS for transcription and made it so the *csgD* promoter could be transcribed by sigma70, earlier during growth and at all growth temperatures**

• **It is thought that this type of promoter mutation is in response to the Enteritidis 27655-3b strain being an RpoS mutant (<https://doi.org/10.1111/j.1574-6968.1998.tb13235.x>), meaning the only way that the strain could produce curli and cellulose would be to acquire the *csgD* promoter mutation likely after the *rpoS* mutation took hold in the strain**

• **The same phenomenon was observed in another Salmonella strain (SARC 15), where it had temperature-independent production of curli and cellulose and the strain also had an RpoS deficiency (<https://doi.org/10.1128/jb.00798-06>)**

The writing of the Xiang et al. manuscript must be revamped to include these important points.

Response: We appreciate your insightful suggestions. As you pointed out, previous studies have reported point mutations in *csgD*. However, these reports were limited to isolated or individual strains and did not identify a widespread occurrence of this point mutation within a specific geographic region. Our study is the first to report a new lineage in China that frequently harbors a point mutation in *csgD*.

We analyzed mutations in the *rpoS* gene for all strains. The results showed that 164 strains exhibited nonsynonymous mutations, including 27 strains with a rough colony phenotype, 57 strains with a smooth colony phenotype, and 80 strains from public databases. Out of the 793 strains that presented mutations in the *csgD* promoter, 604 were categorized as rough colony strains, 79 as smooth colony strains, and 110 were from public databases. Only 42 strains had both nonsynonymous mutations in the *rpoS* gene and mutations in the *csgD* promoter. These comprised 27 rough colony strains, 9 smooth colony strains, and 6 strains from public databases. Given that merely 27 out of the 654 rough colony strains had both *rpoS* nonsynonymous mutations and *csgD* promoter mutations, we infer that the rough colony strains did not originate from acquiring *csgD* mutations subsequent to *rpoS* mutations.

We have included the following description in lines 501-510 of the revised manuscript (clean version) to ensure that all these important points are well described:

“It is worth noting that previous studies have indicated a strong correlation between *rpoS* and the ability to form biofilms^{1,2}. However, in our analysis of proteomic data from 40 strains, we found no statistically significant difference in the expression level

of RpoS between rough and smooth colony strains. Out of the 654 *mrda* variants confirmed through experiments, only 27 had nonsynonymous mutations in the *rpoS* gene. Furthermore, these 27 strains also had mutations in the *csgD* promoter, suggesting that RpoS was not the cause of the variants in our study. Furthermore, we constructed an *rpoS* knockout strain, which showed a decrease in biofilm formation ability, but still exhibited the characteristics of a rough colony on the modified Swarmagar medium (Figure S13).”

Figure S13. Growth of SF175 and SF175- Δ *rpoS* on different media.

A. Growth observed at 37 °C on the modified Swarmagar medium. **B.** Growth observed at 28 °C on the Congo Red medium.

Further, to understand their big speculative points about evolutionary transition and enhanced capacity for environmental adaptation, they must do some type of test on the *rpoS* status of their rough and smooth strains, to see if the same connection is found and also due to the central importance of RpoS in combating stress or adapting to environmental changes. The authors could focus this testing on the 40 strains that were analyzed in more detail in Figure 5.

The ‘enhanced capacity for environmental adaptation’ it is a very complex phenomenon – we still don’t know how biofilms benefit environmental survival and persistence (no hard ‘real-world’ evidence) and the promoter mutation Xiang et al. describe that causes over production of curli and cellulose may not even be a benefit to these Salmonella Typhimurium strains. A lot of testing would need to be performed to reach these conclusions. The boost in CsgD activity due to sigma70-based transcription would also have large consequences as shown by Xiang et al in their proteomics analysis.

The authors need to stick to what their data says – clearly these mrdar strains can persist and have persisted in China, likely transmitted from the first version that caused human infections in China, and they are associated with increased antibiotic resistance due to the presence of those large AMR plasmids. The speculation about ‘increased’ adaptation compared to other strains needs to be reserved to direct research on survival comparing smooth and rough strains of the same lineage, kind of like what has been done for iNTS ST313 strains versus their gastroenteritis-causing ST19 relatives (<https://doi.org/10.1093/femspd/ftw049>)

Response: Thank you for your suggestion. We analyzed proteomics data from 40 strains and found no statistically significant difference in RpoS expression between rough and smooth colony strains (expression ratio=0.90, p=0.569). We also constructed an *rpoS*

knockout strain from the rough colony strain SF175. This knockout strain exhibited a reduced biofilm formation ability, but it still retained the rough colony phenotype on the modified Swarmagar medium and Congo Red medium (Figure S13 in the revised manuscript, as shown below). These results suggest that RpoS is not a key factor influencing the characteristics of rough colony strains. We have included this information in lines 501-510 of the revised manuscript (clean version).

Figure S13. Growth of SF175 and SF175- Δ rpoS on different media.

A. Growth observed at 37 °C on the modified Swarmagar medium. **B.** Growth observed

at 28 °C on the Congo Red medium.

Regarding the claim of “enhanced capacity for environmental adaptation,” extensive literature supports that biofilm formation is a preferred survival strategy for the majority of microorganisms³⁻⁹. The formation of microbial biofilms is known to play an important role in cell survival across diverse environments and to serve as a key pathogenic factor in many localized chronic infections³⁻⁹. To ensure accuracy, we have revised the manuscript, such as replacing “potentially enhanced environmental adaptation” with “a strong biofilm-forming ability” in the Abstract section.

2. Lines 363-364 – the way the authors have written about this data it suggests that not all mrdar strains carried the *csgD* promoter mutation at -44 position. If this is the case, the authors need to list the % of mrdar strains that carry this mutation. I would expect that all mrdar strains carry the -44 promoter mutation (as shown for the 20 strains analyzed in Figure 5A) but the authors need to explain this point better.

Response: Among the 654 mrdar strains identified, 15 belonged to the L1 lineage and none of them had *csgD* mutations; six belonged to the L3 lineage, also without *csgD* mutations; two belonged to the L4 lineage, with one strain carrying *csgD* mutations; and six belonged to the L5 lineage, none of which had *csgD* mutations. The majority of these strains, 625 in total, belonged to the L6 lineage, and 603 (96.5%) of them had *csgD* mutations. The presence of mrdar strains in lineages other than L6 was rare, accounting for a small proportion, with only one strain having *csgD* mutations. In addition, strains outside of the L6 lineage did not form a distinct cluster on the evolutionary tree, unlike those in the L6 lineage. In contrast, 96.5% of mrdar strains in the L6 lineage had *csgD* mutations. We hypothesize that the mrdar phenotype in the L6 lineage is mainly caused by mutations in the *csgD* gene, while the sporadic presence of mrdar strains in other lineages is mainly due to mutations in genes other than *csgD*. To

validate this hypothesis, we constructed a *csgD* knockout mutant (SF175- Δ *csgD*) and a single point mutant with -44T>G substitution in the *csgD* promoter (SF175-*csgD* with -44T>G). We observed that these two mutants were unable to form rdar colonies, showing substantially reduced biofilm formation at 28 °C and 37 °C, in contrast to the wild-type SF175 isolate (Figure 2, Figure S1). We have reported these findings in lines 398-416 of the revised manuscript (clean version).

3. Lines 542-548 – these few sentences are highly speculative – talking about superbugs etc. and a ‘major threat to global public health’ and I recommend taking them out – there is nothing to suggest that the Typhimurium strains they have characterized are any more virulent than normal gastroenteritis-causing strains – and antibiotics are not typically prescribed for gastroenteritis infections caused by Salmonella – because it prolongs shedding – the increased AMR is more likely related to what the authors have described in Lines 537-539

Response: Thank you very much for your thorough and constructive review of our manuscript. We have followed your suggestions to remove the content in lines 542-548 of the original manuscript to prevent any potential misunderstandings.

4. Figure clarity. The authors need to strongly consider how best to represent their data and perhaps re-organize some of the Figures.

• in multiple Figures (4,5) and Supplementary Figures (5, 8, 10) when I zoomed in to see what genes are in each column of the heat map the text got too blurry to be able to read what they are. The text in the main Figures was too small to read in many instances and if the resolution is low in the included Figures, then what is the point of listing the gene names or plasmid names across the top?

Response: We appreciate your comment on the clarity of our figures. We have made several improvements according to your suggestions. Specifically, we have increased

the font size of the gene names and plasmid names in Figures 4, 5 and Figures S5, S8, S10 (i.e., Figures S7, S10, and S12 in the revised supplementary appendix), as shown below. To maintain their resolution and ensure readability, we have uploaded the original PDF files to the submission system.

Revised Figure 4:

Revised Figure 5:

Revised Figure S7:

Revised Figure S10:

Revised Figure S12:

• Figure 3 was only clear to follow and understand when displayed as a full-page, but since this is not possible in the printed version, the authors need to think about what is most important to show. For example, maybe it is best to blow up the circular tree in part A to be full-page width and place the linear trees in B and C below.

Response: We have made modifications to Figure 3 as per your recommendation. Specifically, we have enlarged the circular tree in part A to span the entire width of the page, and we have relocated the linear trees in parts B and C to the bottom, as shown below. We hope that this new layout will improve the figure's readability.

Revised Figure 3:

• Figure 5A – it wasn't possible to read the promoter sequence that is shown here.

This would be better at more of a full-page width.

Response: We appreciate your comment. To make the promoter sequence more readable, we have enlarged Figure 5A to full-page width. The revised Figure 5 is shown below.

A

B

Minor Points

1. Lines 58-59 – these numbers for prevalence of *Salmonella* gastroenteritis are old, please refer to a newer data, such as: Kirk et al. PLoS Medicine 12, no. 12 (2015): e1001921.

Response: We appreciate your help in improving the accuracy of our work. In the revised manuscript, we have updated the prevalence of *Salmonella* gastroenteritis according to the study by Kirk et al and revised lines 66-69 of the revised manuscript (clean version) as follows: “NTS serovars, which can cause salmonellosis, are responsible for an estimated 153 million illnesses and 56,969 deaths among humans worldwide each year¹⁰, representing a significant global public health burden.”

2. Lines 112-115 – The wording is confusing – ‘this variant’ was mostly recovered from humans (64.1%), food (27.2%), and environment (8.7%), but the proportion of the rough colony variant from the environment was higher (40.7%)

• The authors need to explain more here – I thought ‘this variant’ meant the rough colony variant, so mentioning them both in the same sentence is confusing. Please explain exactly what you mean.

Response: We apologize for the confusion caused by the ambiguity in our description of this result. We have revised it to provide a clearer description in lines 144-152 of the revised manuscript (clean version), as follows:

“This variant was isolated from 19 of the 20 monitored regions of China and was mainly distributed in the East, South, Southwest, and Central regions of China (Figure 1A, Table 1). A total of 654 strains exhibiting the rough colony variant were successfully identified and isolated from various samples, including humans (419, 64.1%), food (178, 27.2%), and environment (57, 8.7%). Furthermore, it was observed that the prevalence of the rough colony variant significantly varied across different isolation

sources, with environmental sources (40.7%) exhibiting a higher proportion compared to human sources (29.4%) and food sources (27.6%) ($P < 0.05$) (Figure 1C, Table 1).”

3. Lines 185-186 – ‘were also widely distributed among the six lineages but were mainly distributed in Lineage L6’ – the sentence is a bit contradictory and should be clarified

Response: We appreciate your comment and have revised the sentence to avoid potential contradiction. The revised sentence, now in lines 228-232 of the revised manuscript (clean version), is as follows:

“Organisms from North America were widely distributed across all six lineages, indicating their basal position within the current *S. Typhimurium* populations. Although the Chinese isolates in this study were also present in all six lineages, they were primarily observed in Lineage L6 (59.0%).”

4. Lines 209-211 – The authors need to use caution in making this statement – there could be other variants in other areas of the world that spread the same way

Response: Thank you for your insightful comment. In response to your feedback, we have revised the wording of this statement. The original statement, “Furthermore, the mrdar variants seemingly disseminated to other regions of the world” has been revised to “Furthermore, the mrdar variants seemingly also presented in other regions of the world, as the isolates from Europe (23 isolates), North America (11 isolates), Oceania (four isolates), and South America (one isolate) were clustered in this sublineage (Figure 3C, Table S1).” (lines 257-260 of the revised manuscript [clean version]). We hope that the revised statement more accurately articulates our intended meaning.

5. Line 278 – saying these types of plasmids all belonged to the virulence plasmid pSLT – you mean all plasmids belonged to “the same group as pSLT?”

Response: We appreciate your help in improving the accuracy of our work. Both pSH15G1765-2 and pST90-2 are similar to pSLT. All three plasmids belong to the IncFII(S) replicon and carry the *pefABCD*, *spvBCD*, and *rck* genes. We have revised the manuscript (lines 315-331 of the clean version) to reflect this information.

6. Line 324 – authors are talking about differential core SNPs compared between 20 rough colony variants and 20 smooth colony variants and refer to Figure 5A – Figure 5A is the blow up of the sequence of the *csgD* promoter region – I had a feeling like Figure 5 had been changed from what was described in the main text

Response: Thank you for your inquiry. We have reviewed Figure 5A (as shown below) and confirmed its consistency with the description in the main text. This figure presents a phylogenetic tree of the differential core SNPs between the 20 rough and 20 smooth colony variants, as well as the sequence analysis of the *csgD* promoter region.

7. Lines 324-325 – The authors need to explain better what ‘obvious recombination events’ are, what type of analysis was done and how was this conclusion was arrived at

Response: Thank you for your comment. To address your comment, we have added a new section titled “Recombination analysis” to the supplementary methods. The section reads as follows: “We performed recombination analysis using Gubbins software [doi:10.1093/nar/gku1196] and genomic collinearity analysis using ACT software [doi:10.1093/bioinformatics/bti553]. The results showed that the genomes of all these strains were highly similar and had no significant recombination.”

8. Line 460 – the *yihU-yshA* and *yihVW* operons may not in fact code for an O-antigen capsule – see the following literature (doi:10.1038/nature12947; doi:10.1038/nchembio.2023) – Therefore, I recommend leaving this sentence out

Response: Thank you for pointing out this point. We have reviewed the literature you mentioned and agree with your recommendation. The sentence referring to these operons has been removed from lines 513-517 of the revised manuscript (clean version). The modified text now reads as follows: “In addition to the notable upregulation of the *csg* operon in the *mrda* variants, no obvious changes in the expression levels of the *bcsABZC-bcsEFG* operons responsible for cellulose synthesis were observed among the rough and smooth colony strains. Furthermore, the *bapA* gene responsible for BapA secretion was downregulated among the *mrda* variants.”

9. Lines 472-477 – This extra information regarding chemotaxis could be left out – the authors do not investigate anything regarding the different chemotaxis signals – or form any conclusions

Response: Thank you for your comment. We have removed the mentioned information

from the revised manuscript. We sincerely appreciate your time and effort in helping us improve our work.

10. Lines 567-571 should be left out

Response: We appreciate your insightful feedback. We have removed the mentioned sentences from the revised manuscript.

11. Supplementary Methods – Congo Red Assay - “one colony of each strain was spotted on LB agar” – the authors need to explain in more detail – how can a colony be spotted? – I assume that a specific volume (1-4 uL?) of the overnight culture was pipetted onto the agar and allowed to dry – was the inoculum size standardized for # of cells? for example, was the OD of the overnight culture normalized in any way?

Response: Thank you for your comment. We apologize for any ambiguity in our description of the Congo Red assay. In this procedure, we first revived the strains by streaking them on standard LB agar plates and incubating them overnight at 37 °C. Then, we used 1 µL inoculation loops to pick a single colony from each strain on standard LB agar plates and spotted it on LB agars, which was devoid of salt but supplemented with 40 µg/mL Congo Red and 20 µg/mL Coomassie Brilliant Blue. Since we consistently used inoculation loops of the same size and picked single colonies from overnight cultures, the inoculum size was effectively standardized. This approach is consistent with common practices in previous studies¹¹⁻¹⁴.

Responses to Reviewer #2

The present manuscript shows a huge scientific work, but their analysis and the conclusions drawn would need to be reconsidered in order to get their study publishable.

Response: We are grateful for your time and attention. Your recognition of our scientific work is highly appreciated. We have revised our manuscript based on your comments and suggestions to improve the analysis and strengthen the conclusions. In particular, we have added the cgMLST analysis and performed more experiments to increase the reliability of our data and results. We have also rewritten some parts of the manuscript for clarity and precision. Our point-by-point responses are provided below. We hope that our revisions address your concerns and make our manuscript acceptable for publication.

The authors claim to describe a new variant of *S. Typhimurium* with a distinct rough morphology and potentially enhanced environmental adaptation in China. But regarding their findings, it does not seem to be one variant, but a morphotype shown by bacteria belonging to different lineages inside the serotype, more prevalent in recent isolates of the monophasic variant of *S. Typhimurium* (ST34) isolated in China, mainly from humans. But there are strains belonging to other ST types. So no variant.

Response: Thank you for your insightful comment. We have reviewed the literature and compared the definitions of “lineage” and “variant”. We found that in microbiology, the terms “lineage” and “variant” are often used to distinguish and describe different groups within a specific type or serovar of bacteria. Although they overlap to some extent, they generally refer to slightly different concepts:

Bacterial Lineage

A bacterial lineage is a sequence of successive bacterial populations, strains, or variants that are descended from a common ancestor through reproduction over generations. Lineages are identified and classified based on their genetic makeup and phylogenetic relationships to one another. The study of bacterial lineages is a central aspect of

microbial phylogenetics and evolutionary biology, as it allows for the tracking of hereditary traits and the evolutionary history of bacteria. Within a given bacterial species, multiple lineages may exist, each with its genetic signatures that have emerged and been transmitted over time. A lineage usually refers to a sequence of genetically related strains that have evolved from a common ancestor. It emphasizes a long-term genetic continuity and a direct lineage relationship between strains. A lineage may represent a major branch of a particular serovar, which may contain many further differentiated sub-lineages. In usage, a lineage can be comparable to a “clan,” “line,” or “family,” encompassing a range of strains or variants that can be traced back to a common ancestor.¹⁵⁻¹⁸

Bacterial Variant

A bacterial variant typically refers to a bacterial cell or group of cells within a species that possess one or more genetic differences when compared to the reference strain of that species. These genetic differences could be in the form of mutations, insertions or deletions, gene acquisitions (such as through horizontal gene transfer), or chromosomal rearrangements that differentiate the variant from other strains. Bacterial variants may exhibit different phenotypic characteristics, which can include alterations in virulence, antibiotic resistance, metabolism, or other traits relevant to their survival and propagation.

A variant typically refers to a strain within a serovar that has one or more characteristic differences at the genetic level compared to other members of the same category. These differences could be due to gene mutations, the horizontal transfer of genes, or other mechanisms leading to specific phenotypic or genotypic traits. The concept of a variant tends to be more localized compared to a lineage, possibly pointing to a specific change, such as the acquisition of resistance genes, or the loss or mutation of virulence genes. In short, a lineage usually refers to a relatively larger, evolution-based, and long-term group of bacteria; whereas a variant emphasizes specific, possibly smaller or newer genetic changes.^{15,18}

To summarize, the term “variant” is more commonly used to describe individual or groups of bacteria that are slightly different from the main or reference strain, typically at the genomic level, whereas “lineage” is used to describe a broader ancestral-descendant sequence of bacteria, mapping out an evolutionary path across larger genetic distances and longer periods of time. When analyzing different populations of *Salmonella enterica* serovar Typhimurium or other bacteria, scientists use lineage studies to investigate long-term evolutionary patterns, while the study of different variants can help understand specific issues related to the pathogen’s spread, antibiotic resistance, and pathogenicity.

In the revised manuscript, we have provided clear definitions for the terms “variant” and “lineage” (lines 129-133 of the revised manuscript [clean version]). Specifically, in our study, the term “rough colony variants” refers to strains that exhibited a pronounced wrinkled morphology when inoculated and cultured on the modified Swarmagar medium. These strains demonstrated enhanced biofilm-forming ability at 28 °C and 37 °C compared to the typical rdar strains (reference strain ATCC14028). The term “novel lineage” refers to the L6.5 lineage identified in the phylogenetic analysis. Strains within this lineage frequently carried a distinct point mutation (-44G>T) in the *csgD* gene promoter. The estimated divergence time for this lineage was around 1977. Additionally, we used SeqSero2 to analyze the sequencing data and identified strains belonging to the monophasic variant of *S. Typhimurium*. Detailed results can be found in our response to your MINOR COMMENT 4.

We have ensured the appropriate use of these terms throughout the manuscript based on their defined meanings. Considering that sporadic rough colony strains have been previously reported, but not the large-scale prevalence or the emergence of a new evolutionary branch of these specific variants, we have revised the title of our study to “A novel lineage of *Salmonella enterica* serovar Typhimurium with a rough colony

morphology and multidrug resistance.” We hope that these revisions accurately reflect our findings and address your concerns.

MAJOR COMMENTS

1. The authors talk about a new “variant”, or “new variants” (in plural), a new “lineage”, a new “sublineage”, and a new “subpopulation”. All those terms have very precise and different meanings in phylogeny and taxonomy. The authors should better delimit what they actually describe in this manuscript.:

a) A new lineage with a particular genomic content and a particular phenotype, emerging in China.

Response: Thank you for your comment. In the revised manuscript, we have provided clear definitions for the terms “variant” and “lineage” (lines 129-133 of the revised manuscript [clean version]). Specifically, in our study, the term “rough colony variants” refers to strains that exhibited a pronounced wrinkled morphology when inoculated and cultured on the modified Swarmagar medium. These strains demonstrated enhanced biofilm-forming ability at 28 °C and 37 °C compared to the typical rdar strains (reference strain ATCC14028). The term “novel lineage” refers to the L6.5 lineage identified in the phylogenetic analysis. Strains within this lineage frequently carried a distinct point mutation (-44G>T) in the *csgD* gene promoter. The estimated divergence time for this lineage was around 1977. Additionally, we used SeqSero2 to analyze the sequencing data and identified strains belonging to the monophasic variant of *S. Typhimurium*. Detailed results can be found in our response to your MINOR COMMENT 4.

We have ensured the appropriate use of these terms throughout the manuscript based on their defined meanings. Considering that sporadic rough colony strains have been previously reported, but not the large-scale prevalence or the emergence of a new evolutionary branch of these specific variants, we have revised the title of our study to

“A novel lineage of *Salmonella enterica* serovar Typhimurium with a rough colony morphology and multidrug resistance.” We hope that these revisions accurately reflect our findings and address your concerns.

b) The particular morphology (and capacity of biofilm formation) shown by different strains, which is particularly predominant in a recent clade, prevalent in China in recent years, but also shown by strains belonging to other distant lineages.

Response: Thank you for your valuable feedback. Among the 654 mrdar strains identified, 15 belonged to the L1 lineage, six belonged to the L3 lineage, two belonged to the L4 lineage, and six belonged to the L5 lineage. The majority of these strains, 625 in total, belonged to the L6 lineage. The presence of mrdar strains in lineages other than L6 accounted for only a small proportion and did not form a distinct cluster on the evolutionary tree.

2. As in RESULTS line 107: “We identified a novel variant of *S. Typhimurium* with a rough colony morphology”.

What the authors have observed is a high number of isolates that show that particular morphology. But not being all of them genetically clustered together and separately from any other (smooth) isolate, that is not a new variant. It is a particular morphology that is frequently observed in *Salmonella*, particularly when strains are subcultured several times.

If the authors are trying to define a newly emerged subpopulation of ST34, with a particular phenotype (rough aspect and enhanced biofilm formation), it would be very interesting to add cgMLST information (Enterobase), as the scheme is universally used and could give a known reference to their distances in SNP, that mainly depend on the selected collection.

Response: Thank you for your suggestion. We have performed a cgMLST analysis as

per your recommendation. The results of the cgMLST analysis showed that the rough colony strains formed distinct clusters separate from other branches (Figure S3 in the supplementary appendix, as shown below), indicating considerable genetic divergence. This provides support for the emergence of a subpopulation of ST34 with a specific phenotype characterized by a rough colony and enhanced biofilm formation. This finding has been included in the revised manuscript in the Results section (lines 214-216 of the clean version) and interpreted in the Discussion section (lines 466-469 of the clean version).

Moreover, we would like to clarify that the rough colony strains observed in our study did not acquire this phenotype through multiple subculturing. Instead, they exhibited this characteristic upon initial isolation. Additionally, we conducted long-term subculturing experiments on these isolated rough colony strains, and their morphological features remained stable even after 60 subcultures.

MINOR COMMENTS

3. ABSTRACT line 39: what the authors recovered from publicly available

databases are “genomes”, not “isolates”. This is of importance when describing new phenotypes, as they cannot be known for the genomes exclusively.

Response: Thank you for your suggestion. We have replaced “isolates” with “genomes” in the Abstract section of the revised manuscript (line 40 of the clean version).

4. The authors describe a “rough” lineage (or variant, or subpopulation), that they serotyped by agglutination with sera from SSI. This is confusing, as usually the term “rough” is associated to the impossibility to sero-agglutinate, as the strains are self-agglutinating.

Most of the MRDAR isolates in the study belong to ST34, and the authors claim having performed seroagglutination following the Kauffmann-White scheme. However, they don’t specify whether the isolates in their study were monophasic, as it is usually the case for ST34.

Response: We apologize for the lack of clarity regarding the Kauffmann-White scheme in the original manuscript. We conducted spot inoculation of the strains on Swarmagar medium (Statens Serum Institut, Denmark) and modified Swarmagar medium. Based on the culture results, smooth colony strains were serotyped using the Kauffmann-White scheme. However, as you pointed out, the rough colony strains could not be serotyped using this method due to their self-agglutination. To address this, we retrospectively confirmed the serotypes of these self-agglutinating strains based on whole-genome sequencing with seqsero2¹⁹. The majority of these strains were identified as *Salmonella* Typhimurium, with only a few exceptions. We have revised the manuscript to include these details (lines 619-624 of the clean version).

In addition, to identify strains belonging to the monophasic variant 1,4,[5],12:i:- in this study, we used SeqSero2 to analyze the sequencing data¹⁹. The results showed that the monophasic variant 1,4,[5],12:i:- was primarily found in L6. Specifically, in L6.3, a

significant proportion of 95.53% (748 out of 783 strains) were identified as the monophasic variant. Within L6.5, 36.07% (303 out of 840 strains) belonged to the monophasic variant, with 217 strains being rough-monophasic. The results are presented in the figure below:

Tree scale: 0.01

5. ABSTRACT line 42: The authors use both terms « expansive multidrug resistance » and « extensive multidrug resistance ». The term generally used is « extensive multidrug resistance ». This reviewer would suggest the authors to use the later, and to describe what they exactly mean by that term the first time that it is used, as there is no universal consensus and different authors will mean different things with that.

Response: We appreciate your suggestion. We have consistently used the term “extensive multidrug resistance” throughout the revised manuscript and have provided a definition upon its first mention. For example, in lines 43-45 of the revised manuscript (clean version), we state: “The mrdar variants demonstrated extensive multidrug resistance, exhibiting significantly higher resistance to at least five classes of antimicrobial agents compared to non-mrdar variants.”

6. The authors identify 6 lineages of Salmonella Typhimurium among the > 3 000 Chinese and international genomes analysed. And they determine that most of the MRDAR genomes belonged to L6, and particularly to L6.5, as a very clonal population.

— **The authors cannot know whether there are many more MRDAR isolates in the other lineages, where the majority of genomes have been collected from other countries: they only studied the isolates from China, where most of them belonged to L6.5. Thus, ABSTRACT line 44-45: “... the majority of mrdar variants OBSERVED IN THIS STUDY in sublineage L6.5...”**

Response: Thank you for your comment. We have revised the sentence to include the qualifier “observed in this study.” In the revised manuscript (clean version), lines 47-49 now read “The majority of mrdar variants observed in this study belonged to sublineage L6.5, which originated from Chinese smooth colony strains and possibly

emerged circa 1977.”

— **How do the authors explain the rest of MRDAR, outside L.6?**

Response: Among the 654 *mrdaR* strains identified, 15 belonged to the L1 lineage and none of them had *csgD* mutations; six belonged to the L3 lineage, also without *csgD* mutations; two belonged to the L4 lineage, with one strain carrying *csgD* mutations; and six belonged to the L5 lineage, none of which had *csgD* mutations. The majority of these strains, 625 in total, belonged to the L6 lineage, and 603 (96.5%) of them had *csgD* mutations. The presence of *mrdaR* strains in lineages other than L6 was rare, accounting for a small proportion, with only one strain having *csgD* mutations. In addition, strains outside of the L6 lineage did not form a distinct cluster on the evolutionary tree, unlike those in the L6 lineage. In contrast, 96.5% of *mrdaR* strains in the L6 lineage had *csgD* mutations. We hypothesize that the *mrdaR* phenotype in the L6 lineage is mainly caused by mutations in the *csgD* gene, while the sporadic presence of *mrdaR* strains in other lineages is mainly due to mutations in genes other than *csgD*. To validate this hypothesis, we constructed a *csgD* knockout mutant (SF175- Δ *csgD*) and a single point mutant with -44T>G substitution in the *csgD* promoter (SF175-*csgD* with -44T>G). We observed that these two mutants were unable to form *rdar* colonies, showing substantially reduced biofilm formation at 28 °C and 37 °C, in contrast to the wild-type SF175 isolate (Figure 2, Figure S1). We have reported these findings in lines 398-416 of the revised manuscript (clean version).

— **There seems to be very little variability inside L.6.5, if compared to other lineages. Have the authors analysed a possible outbreak, regarding those isolates? Have the authors compared the cgMLST profiles of their isolates and those recovered from NCBI, by the HierCC EnteroBase scheme? That could be very helpful to give an idea of proximity.**

Response: Thank you for your insightful comment. We would like to clarify that the strains analyzed in our study were not isolated from large outbreaks. First, these strains were collected from CDCs in various regions and isolated over a broad time span, ranging from 2006 to 2018. Their background information did not indicate any apparent outbreaks. Second, the phylogenetic tree analysis revealed that strains from different regions tended to cluster together (Figure 3 in the revised manuscript, shown below), indicating no specific geographical clustering. Third, in response to your suggestion, we performed a cgMLST analysis. The rough colony strains formed a distinct large cluster containing many smaller branches (Figure S3 in the supplementary appendix, shown below). This pattern highlights the diversity within the rough colony strains. Finally, we analyzed the SNP differences among strains in L6.5 and a subset of strains within L6.5 that were isolated from the same geographic region and clustered together (as indicated by the red box in the figure below). We found that the SNP differences among strains in L6.5 ranged from 0 to 423, while the SNP differences among those within the red box were between 0 and 44. According to a previous study²⁰, SNP differences among outbreak *Salmonella* strains should be less than five. Therefore, this result confirms that the strains in our study did not originate from large outbreaks.

Figure 3. Population structure of the 3,951 *S. Typhimurium* genomes analyzed in this study.

Figure S3. Minimum spanning tree comparing core-genome allelic profiles among different groups (rough type, smooth type, and other).

— The authors infer dates of appearance of the different lineages in the phylogeny analysis. Also, they use the dates of collection known for the Chinese isolates in previous graphs. But it would be most interesting to show the collection dates in the phylogeny tree. That could perhaps help to identify (or to exclude) an outbreak that might explain the clonal expansion of the RDAR ST34 L6.5 subpopulation they describe here.

Response: We appreciate your suggestion to include the collection dates in the phylogeny tree. However, because there are too many collection dates, adding them to the tree would make the image complicated and hard to read. For clarity and accessibility, we have provided the collection dates in Table S1 (see the screenshot below). We hope that this table would provide a comprehensive overview of the data.

Table S1. The genome-sequenced *S. Typhimurium* isolates used in this study.

Sample	Collection date	Geographic location	Region	Source type	Source	diagnosis	Sex	Age Range	Age	Sequence type	Lineage-A270	Sublineage-A270	S-R phenotype	Triaxone	Cracycline
Z2016SF03	2016	Southern China	Guangdong	Food	Pork	NA	NA	NA	NA	ST19	L5	L5.2	Smooth	S	R
Z2016SF03	2016	Southern China	Guangdong	Food	Pork	NA	NA	NA	NA	ST19	L5	L5.2	Smooth	S	R
Z2016SF04	2016	Southern China	Guangdong	Food	Pork	NA	NA	NA	NA	ST34	L6	L6.3	Smooth	S	R
Z2016SF05	2016	Southern China	Guangdong	Food	Pork	NA	NA	NA	NA	ST19	L1	L1.2	Smooth	S	R
Z2016SF05	2016	Southern China	Guangdong	Food	Pork	NA	NA	NA	NA	ST34	L6	L6.3	Smooth	S	R
Z2016SF05	2016	Southern China	Guangdong	Food	Pork	NA	NA	NA	NA	ST34	L6	L6.3	Smooth	S	R
Z2016SF06	2016	Southern China	Guangdong	Food	Pork	NA	NA	NA	NA	ST34	L6	L6.3	Smooth	S	R
Z2016SF06	2016	Southern China	Guangdong	Food	Pork	NA	NA	NA	NA	ST34	L6	L6.3	Smooth	S	R
Z2016SF06	2016	Southern China	Guangdong	Food	Pork	NA	NA	NA	NA	ST34	L6	L6.3	Smooth	S	R
Z2016SF06	2016	Southern China	Guangdong	Food	Pork	NA	NA	NA	NA	ST19	L1	L1.2	Smooth	S	R
Z2016SF06	2016	Southern China	Guangdong	Food	Pork	NA	NA	NA	NA	ST19	L1	L1.2	Smooth	S	R
Z2016SF07	2016	Southern China	Guangdong	Food	Pork	NA	NA	NA	NA	ST34	L6	L6.3	Smooth	S	R
Z2016SF07	2016	Southern China	Guangdong	Food	Pork	NA	NA	NA	NA	ST34	L6	L6.3	Smooth	R	R
Z2016SF08	2016	Southern China	Guangdong	Food	Pork	NA	NA	NA	NA	ST34	L6	L6.5	Rough	S	R
Z2016SF11	2016	Southern China	Guangdong	Food	Pork	NA	NA	NA	NA	ST34	L6	L6.5	Rough	S	R
Z2016SF11	2016	Southern China	Guangdong	Food	Pork	NA	NA	NA	NA	ST34	L6	L6.4	Smooth	S	R
Z2016SF12	2016	Southern China	Guangdong	Food	Pork	NA	NA	NA	NA	ST34	L6	L6.5	Rough	S	R
Z2016SF12	2016	Southern China	Guangdong	Food	Pork	NA	NA	NA	NA	ST34	L6	L6.4	Smooth	S	R
Z2016SF12	2016	Southern China	Guangdong	Food	Pork	NA	NA	NA	NA	ST34	L6	L6.5	Rough	S	R
Z2016SF13	2016	Southern China	Guangdong	Food	Pork	NA	NA	NA	NA	ST34	L6	L6.3	Smooth	S	R
Z2016SF13	2016	Southern China	Guangdong	Food	Pork	NA	NA	NA	NA	ST34	L6	L6.4	Smooth	S	R
Z2016SF13	2016	Southern China	Guangdong	Food	Pork	NA	NA	NA	NA	ST34	L6	L6.4	Smooth	S	R
Z2016SF14	2016	Southern China	Guangdong	Food	Pork	NA	NA	NA	NA	ST34	L6	L6.3	Smooth	S	R
Z2016SF14	2016	Southern China	Guangdong	Food	Pork	NA	NA	NA	NA	ST34	L6	L6.3	Smooth	S	R
Z2016SF15	2016	Southern China	Guangdong	Food	Pork	NA	NA	NA	NA	ST34	L6	L6.5	Rough	S	R

7. As the authors aim to describe a particular, stable capacity of forming biofilm of the new variant they are describing, Figure S1-D and E should show the 20 SMOOTH and the 20 ROUGH isolates that the authors claim having tested. Biofilm formation can be very variable and so if the 20 isolates of each morphology behaved the same way, it would be most important to show it.

Response: Thank you for your comment. We have conducted crystal violet biofilm assays, examination of the red, dry and rough morphotype, and pellicle formation experiments to demonstrate the difference in biofilm formation ability between the 20 rough and 20 smooth colony isolates. The results of crystal violet biofilm assays are shown in Figures S1A-C. The results of examination of the red, dry and rough morphotype are shown in Figure S2A. The results of pellicle formation are shown in Figures S2B and S2C. In addition, we performed confocal laser scanning microscopy (CLSM) imaging on one rough colony isolate (SF175), one smooth colony isolate (2015114), the standard strain (14028), SF175-csgD (-44 T>G), and SF175-ΔcsgD. The

CLSM results are shown in Figures S1D and S1E.

We have included the results of biofilm formation tests under the heading “Biofilm formation” in lines 179-186 of the revised manuscript (clean version), as follows:

“Crystal violet biofilm assays and pellicle formation assays showed that the *mrda* variants examined in this study had greater biofilm-forming ability than the smooth variants; for example, the *mrda* colony from SF175 generated significantly more biofilm than the control ATCC strain and strain 2015114 at both 28 and 37 °C (Figure S1 and S2). Moreover, confocal laser scanning microscopy (CLSM) observations showed that the biofilm of the *mrda* colony morphotype had more EPSs (green color) than the smooth morphotype and ATCC 14028. Overall, the *mrda* variants exhibited greater biofilm-forming ability at 28 °C than at 37 °C (Figure S1).”

Figure S1. Biofilm formation by the representative rough and smooth colony strains.

Figure S2. Examination of the red, dry and rough morphotype and pellicle formation in representative rough and smooth colony strains.

8. Figures S1-A, B and C claim to be a representation of experiments repeated in triplicate. As biofilm formation is usually quite variable, it would be good if the figure included the results of the three times the analysis was done, with their error bars for each isolate.

Response: Thank you for your constructive feedback. The original Figure S1-A and B presented the mean values of results from the three repeated experiments. To provide a more comprehensive presentation of our data, we have revised Figures S1-A and B to include raw data from the three experiments. Additionally, we have added error bars to Figure S1-C to present the standard deviation of results from three experiments. We hope that these modifications will enhance the comprehensiveness and precision of our data presentation. Please find the updated Figures S1-A, B, and C below:

9. The Heatmap in figure 5-B shows a clear difference in the presence/absence of the mutation in *csgD* promoter.

— Did the authors identify the exact sequence of the promoter, or they refer to as so to the several base pairs prior to the genes CDS?

Response: Thank you for raising this point. We identified the exact sequence of the *csgD* promoter. Specifically, we employed the Snippy pipeline to identify the 755 bp intergenic region between the *csgBAC-csgDEFG* operon with the standard strain sequence as a reference. We validated the sequence through Sanger sequencing. Further details regarding this methodology can be found in the “Biofilm-related genes and mutation detection” section of the supplementary methods.

— Did they authors test the biofilm formation on isolates from the other lineages, when the mutation is not present, to prove its implication in the particular biofilm formation of the present variant?

Response: Thank you for your comment. We conducted the experiment at least three times to confirm the rough or smooth phenotypes for all 2,212 strains. Furthermore, we selected strains from lineages other than L6 and assessed their biofilm formation ability. Please see below for the results of some strains:

37 °C

Lineage-A270	Sublineage-A270	S-R phenotype	Biofilm forming ability	Parallel data 1	Parallel data 2	Parallel data 3	OD ₆₀₀
L5	L5.2	Smooth	No biofilm forming ability	0.3104	0.237	0.216	0.254466667
L3	L3.7	Smooth	No biofilm forming ability	0.2312	0.2466	0.2147	0.230833333
L1	L1.3	Smooth	No biofilm forming ability	0.2797	0.2566	0.2624	0.266233333
L1	L1.6	Smooth	No biofilm forming ability	0.24	0.2224	0.22	0.227466667
L1	L1.7	Smooth	No biofilm forming ability	0.2345	0.271	0.2808	0.2621
L3	L3.1	Smooth	No biofilm forming ability	0.1723	0.2534	0.2429	0.222866667
L4	L4.2	Smooth	No biofilm forming ability	0.2864	0.2568	0.268	0.2704
L3	L3.4	Smooth	No biofilm forming ability	0.2203	0.2323	0.2059	0.2195
L1	L1.4	Smooth	No biofilm forming ability	0.341	0.3272	0.409	0.359066667
L1	L1.5	Smooth	No biofilm forming ability	0.3262	0.3181	0.3153	0.319866667
L3	L3.2	Smooth	No biofilm forming ability	0.2918	0.261	0.283	0.2786
L3	L3.2	Smooth	No biofilm forming ability	0.3173	0.3591	0.3435	0.339966667
L2	L2.3	Smooth	No biofilm forming ability	0.2684	0.2488	0.248	0.255066667
L3	L3.4	Smooth	No biofilm forming ability	0.2398	0.1733	0.1748	0.195966667
L3	L3.5	Smooth	No biofilm forming ability	0.3	0.268	0.4328	0.3336
L4	L4.2	Smooth	No biofilm forming ability	0.3975	0.3479	0.2918	0.345733333
L1	L1.3	Smooth	No biofilm forming ability	0.3346	0.3172	0.2828	0.311533333
L1	L1.2	Smooth	No biofilm forming ability	0.3604	0.342	0.3419	0.3481
L1	L1.7	Smooth	No biofilm forming ability	0.3057	0.3004	0.2805	0.295533333
L5	L5.2	Smooth	No biofilm forming ability	0.2487	0.2513	0.3012	0.267066667
L4	L4.2	Rough	Weak biofilm forming ability	0.6161	0.5377	0.5544	0.5694
L3	L3.4	Rough	Weak biofilm forming ability	0.7378	0.6616	0.7221	0.707166667
				0.181000001	0.1705	0.1917	0.181066667

28 °C

ID	Sample	Lineage-A270	Sublineage-A270	S-R phenotype	Biofilm forming ability	Parallel data 1	Parallel data 2	Parallel data 3	OD ₆₀₀
1	GZ2016SF032	L5	L5.2	Smooth	No biofilm forming ability	0.211	0.2112	0.206	0.2094
2	SH06SF002	L3	L3.7	Smooth	Weak biofilm forming ability	0.3808	0.4652	0.4324	0.426133333
3	SH08SF032	L1	L1.3	Smooth	No biofilm forming ability	0.2254	0.2704	0.2564	0.250733333
4	SH09SF003	L1	L1.6	Smooth	No biofilm forming ability	0.1818	0.1954	0.1869	0.188033333
5	SH09SF004	L1	L1.7	Smooth	Strong biofilm formation ability	0.6926	0.6382	0.6237	0.6515
6	SH09SF029	L3	L3.1	Smooth	Weak biofilm forming ability	0.2578	0.262	0.2828	0.267533333
7	SH09SF095	L4	L4.2	Smooth	Weak biofilm forming ability	0.3359	0.2801	0.2586	0.291533333
8	SH12SF128	L3	L3.4	Smooth	Weak biofilm forming ability	0.4274	0.3902	0.3644	0.394
9	SH13SF733	L1	L1.4	Smooth	Weak biofilm forming ability	0.3131	0.2915	0.3466	0.317066667
10	SH18SF1760	L1	L1.5	Smooth	Weak biofilm forming ability	0.3429	0.2453	0.2162	0.268133333
11	2016126	L3	L3.2	Smooth	Weak biofilm forming ability	0.3714	0.4317	0.3806	0.394566667
12	SH16G2569	L3	L3.2	Smooth	Weak biofilm forming ability	0.3232	0.3941	0.3349	0.350733333
13	SH17G1353	L2	L2.3	Smooth	Weak biofilm forming ability	0.4285	0.3969	0.3829	0.402766667
14	SH18G2908	L3	L3.4	Smooth	Weak biofilm forming ability	0.3175	0.3543	0.4052	0.359
15	SH12SF119	L3	L3.5	Smooth	No biofilm forming ability	0.2838	0.201	0.2221	0.235633333
16	SH12G1124	L4	L4.2	Smooth	No biofilm forming ability	0.2373	0.2359	0.188	0.2204
17	SH12SF435	L1	L1.3	Smooth	No biofilm forming ability	0.1998	0.2363	0.2775	0.237866667
18	SH13SF577	L1	L1.2	Smooth	No biofilm forming ability	0.2251	0.2152	0.2346	0.224966667
19	SH14G1532	L1	L1.7	Smooth	Weak biofilm forming ability	0.3919	0.4133	0.4671	0.4241
20	SH16G1761	L5	L5.2	Smooth	Weak biofilm forming ability	0.2669	0.2462	0.4918	0.334966667
21	41-Sa-13D033	L4	L4.2	Rough	Strong biofilm formation ability	1.3089	1.2004	1.3814	1.2969
22	SH16G2665	L3	L3.4	Rough	Strong biofilm formation ability	0.6783	0.5791	0.7195	0.658966667
Blank control						0.1444	0.1163	0.1232	0.127966667

Responses to Reviewer #3

Within this study from Xiang et al. this team documents a novel *Salmonella enterica* serovar Typhimurium variant identified within Chinese samples collected over the last 15 years associated with a change in colony morphology. Using genomic approaches, this team assesses the relatedness and presence of virulence factors / AMR genes associated with the observed rough strains revealing the enrichment of these strains within the L6.5 sub-lineage. Using GWAS the team identifies a potential driver of the change in colony morphology and then using proteomic, phenotypic and mutagenesis experiments provide preliminary evidence for the involvement of the *csgDEFG* operons and specifically a single SNP in the promoter of *csgD* for this alteration. While the study is clearly a large body of work it does feel disjointed, and some sections do not appear to provide any additional progression in the narrative such as the sections on biofilm formation or the AMR profiles. Additionally, significantly more care is required in the description of the methods in particular the details provided on the proteomic analysis are grossly insufficient. Due to this I feel this study requires additional work to further support the claims (such as that these variants have increased stress resistance) meaning in its current form this work is unsuitable for publication.

Response: Thank you for your insightful comment and constructive criticisms on our study. We agree that our manuscript needs more coherence and details in certain sections. In response to your concerns, we have revised the manuscript. Specifically, we have rewritten the sections on biofilm formation and AMR profiles to better articulate their relevance to our primary findings and to the existing literature on *Salmonella* pathogenesis and evolution. Additionally, we have supplemented our statements with additional references to underscore the novelty and significance of our work. Furthermore, we have provided additional details on the proteomic analysis in the supplementary methods, including information on sample preparation, mass

spectrometry settings, data processing, statistical analysis, and functional annotation. We have included a supplementary table that lists all the proteins identified and quantified in our study, along with their p-values and functional categories. Finally, we have updated the results and figures, and discussed the implications of new findings. Please find our point-by-point responses below. We sincerely hope that these revisions have addressed your concerns and have improved the quality and clarity of our manuscript. Thank you again for your valuable feedback and suggestions.

Major concerns:

Arguably the first half of the results sections (line 100 to 307) could be re-written and streamlined as these are very descriptive summaries of the plasmids, virulence genes and AMR genes observed across strains but do not provide insights into the key biological question being asked, which is what the cause of the morphological change in the novel *S. Typhimurium* variants is.

Response: We appreciate your suggestion. Accordingly, we have revised and streamlined the first half of the results section. Given the extensive revisions, we kindly refer you to lines 136-348 of the revised manuscript (clean version).

The introduction is extremely brief and doesn't provide context on the enormous impact genomic analysis has had on *Salmonella* tracking or our understanding of the emergence of virulent *Salmonella* variants. This should at least be re-written to help put this work in context.

Response: Thank you for your suggestion. We have incorporated background information regarding the impact of genomic analysis on the emergence of virulent *Salmonella* variants in the second paragraph of the introduction section. The specific revisions can be found in lines 77-106 of the revised manuscript (clean version), as shown below:

“*S. Typhimurium* has shown remarkable genetic diversity and the ability to sense and respond to environmental stress. Mutation and evolution are primary drivers of its adaptation to environments²¹. Whole-genome sequencing technology has revolutionized *Salmonella* epidemiological research, enabling the identification of sequence types, the prediction of key genes, the tracking of transmission, the analysis of mutation evolution, and the examination of pathogen evolutionary patterns^{22,23}. In recent decades, several dominant *S. Typhimurium* variants, including definitive phage types DT9, DT204, DT29, DT104, and DT193, the monophasic variant 1,4,[5],12:i:-, and invasive NTS (iNTS), have been associated with successive human salmonellosis epidemics²⁴⁻²⁹. Moreover, studies have investigated *S. Typhimurium* genomic epidemiology at global and regional levels and identified several phylogenetic lineages characterized by multilocus sequence types (STs), such as ST19, ST34, ST36, and ST313²⁴⁻³⁴. Recently, *Salmonella enterica* serovar 1,4,[5],12:i:-, a monophasic variant of *S. Typhimurium* belonging to ST34, has emerged as a major global cause of NTS disease in animals and humans. Whole-genome evolutionary analysis has shown that this variant evolved into an independent clone and acquired a new genomic island SGI-4, which conferred resistance to multiple antibiotics and the heavy metal copper. Additionally, gene insertion and deletion events have played important roles in the microevolution of this clone²⁹. Furthermore, whole-genome analysis of ST313 African isolates has indicated the emergence of two different lineages from independent origins. Lineage I prevailed in the southwest region of Africa, while Lineage II prevailed in Malawi²⁸. Later research identified a sub-lineage, II.1, in Congo, which carried IncHI1 plasmids that conferred resistance to azithromycin and extended-spectrum beta-lactamases³⁴. Point mutations in the promoter region of the virulence gene *pgtE* have been found to upregulate its expression in ST313, contributing to increased virulence and thus distinguishing it from ST19 in phenotype³⁵. These dominant pathovariants and epidemic lineages exhibit distinct host ranges, niche adaptations, levels of pathogenicity, and risks to food safety. Their infections are associated with a higher probability of hospitalization and treatment failure, leading to a prolonged infection and potential

onward transmission. Therefore, they pose serious public health threats^{24,26,27,33,34}.”

The methods provided for the proteomic analysis are grossly insufficient with no details on how strains were grown, how the protein lysates were prepared or any details on the LC parameters (column used, gradient length, flow rate buffers etc) or MS instrument settings (DIA window width, maximum fill time, AGC, resolution etc) used provided.

Response: We are grateful for your insightful feedback. We have revised our manuscript and elaborated on the methods used for the proteomic analysis. We have included a new section titled “Sample preparation and fractionation for proteomics research” in the supplementary methods, which reads as follows:

“Sample collection and preparation

We inoculated 40 strains of rough and smooth *Salmonella* typhimurium onto the modified Swarmagar medium using the inoculation loop method for preliminary identification of *Salmonella*. After incubating them at 37 °C for 18 hours, we gently scraped off the colonies with an inoculation ring, avoiding the culture medium components. We transferred the scraped colonies into 2 mL EP tubes containing 1.5 mL PBS. To obtain enough samples, we repeated this process until the liquid inside the EP tubes became visibly turbid. We labeled the EP tubes with the strain numbers on the walls or caps and centrifuged them at 5000 rpm for 9 minutes. We discarded the supernatant and obtained the bacterial samples as the sediments in the EP tubes. We added 125 µL of lysate to each EP tube, vortexed them for 30 seconds, left them at room temperature for 5 minutes, froze them in liquid nitrogen for 5 minutes, and stored them in a -80 °C freezer.

Sample homogenization and lysate preparation

We first homogenized the samples using the MP FastPrep-24 homogenizer (24×2, 6.0M/S, 60s, twice), and then added SDT buffer (4% SDS, 100 mM Tris-HCl, pH 7.6). We sonicated the lysates and boiled them for 15 minutes. After centrifuging them at

14000 g for 40 minutes, we quantified the supernatants with the BCA Protein Assay Kit (Bio-Rad, USA). We mixed 20 µg of protein from each sample with 5X loading buffer and boiled them for 5 minutes. We separated the proteins on a 4%-20% SDS-PAGE gel (constant voltage 180 V, 45 minutes) and visualized the protein bands by Coomassie Blue R-250 staining.

DDA library generation and quality control

We pooled an equal aliquot from each sample for DDA library generation and quality control.

Sample treatment and peptide collection

We added the detergent DTT (final concentration 10 mM) to each sample and mixed them at 600 rpm for 1.5 hours (37°C). After cooling the samples to room temperature, we added IAA (final concentration 20 mM) to block the reduced cysteine residues and incubated the samples for 30 minutes in the dark. We transferred the samples to the filters (Microcon units, 10 kDa) and washed them with 100 µl UA buffer three times and then 100 µl 25mM NH₄HCO₃ buffer twice. We added trypsin to the samples (trypsin: protein [wt/wt] ratio 1:50) and incubated them at 37 °C for 15-18 hours (overnight). We collected the resulting peptides as a filtrate. We desalted the peptides of each sample on C18 Cartridges (Empore™ SPE Cartridges C18 [standard density], bed I.D. 7 mm, volume 3 ml, Sigma), concentrated them by vacuum centrifugation and reconstituted them in 40 µl of 0.1% (v/v) formic acid. We estimated the peptide content by UV light spectral density at 280 nm. For DIA experiments, we spiked iRT (indexed retention time) calibration peptides into the samples.

Peptide fractionation

We fractionated the digested pool peptides into 10 fractions using the Thermo Scientific™ Pierce™ High pH Reversed-Phase Peptide Fractionation Kit. We desalted each fraction on C18 Cartridges (Empore™ SPE Cartridges C18 [standard density], bed I.D. 7 mm, volume 3 ml, Sigma) and reconstituted them in 40 µl of 0.1% (v/v) formic acid. We spiked the iRT-Kits (Biognosys) peptides before data-dependent acquisition (DDA) analysis.”

In addition, we have incorporated a description of the mass spectrum in the supplementary methods, under the section “Proteomics sequencing of the rough and smooth colony strains,” which reads as follows:

“The detailed methodology and analysis process have been described previously³⁶. Briefly, all fractions for DDA library generation were injected on a Thermo Scientific Q-Exactive HF-X mass spectrometer connected to an Easy-nLC 1200 chromatography system (Thermo Scientific). The peptide was separated on a C18 Analytical Column (Thermo Scientific, ES802, 1.9 μm , 75 μm *20 cm) with a linear gradient of buffer B (84% acetonitrile in 0.1% formic acid) at a flow rate of 300 nl/min. The MS detection method was positive ion, the scan range was 350-1800 m/z, the resolution for the MS1 scan was 60000 at 200 m/z, the target of AGC (Automatic gain control) was 1e6, the maximum IT was 50 ms, and dynamic exclusion was 10.0 s. After each full MS-SIM scan, 20 ddMS2 scans followed according to the inclusion list. The isolation window was 1.5 m/z, the resolution for the MS2 scan was 30000 (@m/z 200), the AGC target was 1e5, the maximum IT was 50 ms, and the normalized collision energy was 30 eV. The peptides of each sample were analyzed by a Q-Exactive HF-X mass spectrometer connected to an Easy-nLC 1200 chromatography system in the DIA mode. Each DIA cycle contained one full MS-SIM scan, and 44 DIA scans covered a mass range of 350-1800 m/z with the following settings: SIM full scan resolution was 120,000 at 200 m/z; AGC 3e6; maximum IT 30ms; profile mode; DIA scans were set at a resolution of 30,000; AGC target 3e6; Max IT auto; MS2 Activation Type HCD; normalized collision energy was 30 eV.

For DDA library data, the FASTA sequence database was searched with SpectronautTM 14.4.200727.47784 (Biognosys) software. The database was downloaded from the website: <http://www.uniprot.org>. The iRT peptides sequence was added (Biognosys|iRT Kit). The parameters were set as follows: the enzyme was trypsin, the max missed cleavages was 1, the fixed modification was carbamidomethyl(C), and the dynamic modification was oxidation(M) and acetyl (Protein N-term). All reported data were

based on 99% confidence for protein identification as determined by false discovery rate (FDR) \leq 1%. DIA data was analyzed with Spectronaut™ 14.4.200727.47784, searching the above-constructed spectral library. The main software parameters were set as follows: the retention time prediction type was dynamic iRT, interference on MS2 level correction was enabled, and cross-run normalization was enabled. All results were filtered based on a Q value cutoff of 0.01 (equivalent to FDR < 1%).”

Additionally, further details are needed on how the DDA fractions were prepared (was this by basic reverse phase fractionation? how many fractions? What column was used for sample fractionation, was the sample used to create the DDA library a pooled reference of all the strains?)

Response: Thank you for your inquiry. We have included a new section titled “Sample preparation and fractionation for proteomics research” in the supplementary methods to provide more details. This section provides information about sample pooling, the fractionation process, and library construction. Please refer to our previous response for the specific methods.

On the bioinformatics associated with this, what were the search settings used within Spectronaut? As 40 different strains were used with 40 different proteomes were individual proteome built from the sequencing data and searched together or was a single reference strain used? was a minimum number of peptides required for consideration for analysis?

Response: Thank you for your question. We have included the search settings used within Spectronaut in the supplementary methods, under the title “Proteomics sequencing of the rough and smooth colony strains.” Please refer to the previous response for details.

In our study, we conducted a combined search for the 40 samples using the Uniprot sequence database, without considering the mutation for each strain. To ensure the reliability of our results, we set a 1% false discovery rate (FDR) at the protein level. We did not set a minimum peptide number. This approach aligns with the standard practice in proteomic research³⁷⁻³⁹.

if imputation was used how was it undertaken? Please provide details on the statistical test used to assess changes, was this done using a t-test and was multiple hypothesis correction applied?

Response: Thank you for your question. Given the large number of strains included in our analysis, there were sufficient data for most proteins to conduct a meaningful differential analysis even without imputation. Therefore, we did not perform imputation to avoid potential inaccuracies that could arise from the imputation process. Proteins with insufficient quantitative samples were excluded from the differential analysis.

Since the number of proteins identified in proteomics is generally lower than in transcriptomics, many proteomics studies⁴⁰⁻⁴², including ours, use p-value cut-off values instead of FDR cut-off values. In our analysis, we employed t-tests without multiple hypothesis correction to identify differential proteins. We have revised the Methods section to include these details (lines 734-736 of the clean version).

For the GO enrichment analysis of the proteomic data while it is stated in the main body methods that Fisher's exact test were used no output (supplementary tables) are provided with these results. Likewise, the Spectronaut search output from the proteomic experiment (all 2,998 proteins) should be provided not just a summary of the results.

Response: Thank you for your comment. We have revised Table S7 (previously referred

to as Table S6 in the original submission) and added a new supplementary table, Table S8, to present results of the Spectronaut search and Fisher’s exact tests.

For the Spectronaut search output, which includes all 2,998 proteins, we have included a new column titled “Significant difference” in Table S7. This column presents whether the protein expression was significantly up-regulated, down-regulated, or not significant when comparing the rough and smooth colony strains. Moreover, we have updated the title of Table S7 to “The identified proteins and differentially expressed proteins among the rough and smooth colony variants.” A screenshot of Table S7 is provided below:

S2	S3	S4	S5	S6	S7	S8	S9	average S	R/S	t test p value	Significant difference	
509.2420349	233.5640564	372.4873962	71.90001678	307.546875	176.8792725	254.3435516	183.0009918	416.3749378	28.82055439	3.51939E-24	Up	
614.4072266	267.7281799	486.2010498	123.7438736	316.96521		224.4969635		565.1315263	21.11744567	9.23827E-08	Up	
3328.036133	1180.304565	1707.000366	453.6679382	1268.308472	494.519043	1303.950073	581.1365967	1859.868813	18.10251831	9.18203E-19	Up	
165.9936371		170.9851227	67.25211334	55.45052338		48.536129	213.8700714	310.9166875	16.72408472	1.12839E-11	Up	
	155.0587311	118.4838562	51.02513504	60.19170761	383.4772949	105.4963837	65.01330566	219.5676734	16.04485251	1.39505E-10	Up	
402.2999268	337.7847595	292.2921382	571.0732422	338.6680908	176.7084503	290.481781		333.6857829	13.11582181	7.981E-15	Up	
1143.014893		360.0984497		286.2722778	223.0882568	512.4013062	257.8266296	574.3518239	11.43246329	3.7149E-10	Up	
289.4311218	229.2199402	411.4985657	281.401947	123.4505692	266.6456604	187.8126678	231.0639343	266.8469449	10.37210316	4.81922E-06	Up	
662.9952393	260.3855591	374.1190796	574.3115234	211.7725525	333.4150391	248.1086121	164.1917114	428.8876785	9.666836382	2.55411E-12	Up	
257.3962708		498.0480042	226.9034576	401.6342468	253.3670197	633.8950195	8710.556641	1671.868041	8.500410289	0.004829262	Up	
456.4040833	337.1806946	349.9984436	259.2275696	246.7051086	213.2797852	338.0570984	168.4997253	365.228582	5.024084609	5.64359E-12	Up	
622.84729	231.8461304	327.4237976	442.6315308	284.3116455	317.8868408	364.671936		178.6152954	424.1593124	4.960861966	8.97726E-12	Up
384.3231201	111.117012		41.54677963		142.9792938		37.58293152	108.9010507	4.774229201	0.003826542	Up	
3080.820313	1442.948364	1573.538208	894.5764771	1471.422852	1973.325684	1510.293091	1247.652344	1201.608135	4.30124851	6.90457E-05	Up	
534.5996704	251.1273193	193.5606689	384.6871338	295.3937378	276.3466492	401.2834778	69.06998444	320.3710709	3.879909883	2.18269E-07	Up	
2441.704834	981.2924194	593.3397827	1413.207764	1221.978516	929.6374512	2284.620361	512.0147095	1248.894937	3.790860875	2.09169E-06	Up	
595.0045776	320.8663635	247.543335	424.4527283	400.6943665	203.0409241	457.1225586	171.0980225	420.5305684	3.747845561	4.15192E-07	Up	
114.8455505	37.02613068	90.05728149	170.2617188	51.93901825	76.39193726	55.68525314		116.3770568	3.679445286	1.60492E-10	Up	
2990.20166	2683.616455	5786.848633	4828.558594	3545.825928		1598.106201	2287.853516	2171.560242	3.515926124	9.90095E-06	Up	
724.2134399	152.9148102	490.8237	298.5913696	301.0565491	1106.520752	231.9859924	602.9586792	431.9105869	3.464382276	0.000181762	Up	
1756.451904	433.8340149		886.3955078		734.3798218	257.2504272		845.9634219	3.405934849	0.006495662	Up	
852.465332	394.6954956	200.0504913	277.8392029	459.5766907	218.6374817	550.3657837	175.2865295	404.6296583	3.231431174	5.00468E-07	Up	
234.3905029	254.2096405	157.855011	288.729187	213.1729736	100.1700592	289.7410583	156.9784088	287.366757	3.163008341	1.71685E-11	Up	
1936.653564	447.696106	450.5215759	479.0079651	166.0447845	1235.647949	332.1816406	352.7712402	506.9962769	3.11597909	0.00011376	Up	
425.4919128	227.3491211	981.5248413	2309.052979	560.1936646	533.0383301	120.3998032	289.9900513	543.5247807	3.086542643	0.002463411	Up	
2541.12793	1458.751587	778.1484375	473.1547241	468.5661621	679.7339478	439.4355774	248.7431793	813.4394087	3.085262848	0.010741797	Up	
123.0046463	47.69697952	182.2969513	191.7729645	56.66240692	308.1329041	94.2726059	124.1948318	145.2967778	3.014601979	8.08594E-05	Up	
3550.486816	900.8314209	1136.28479	990.1865234	698.0634155	1119.490845	479.1716919	457.7549744	1016.071872	3.004784089	0.00059031	Up	
3245.044922	2293.629395	4233.115234	4796.472168	3375.64624	5422.582031	2479.399414	4128.672363	3444.294696	2.799652848	1.7106E-08	Up	
528.2659912	284.8307495	594.3388062	298.2188416	260.390625	1448.734619	432.3329163	412.8874207	504.8289017	2.774374365	0.000116753	Up	
256.4862976	283.8537598	146.9075317		66.91038513	101.8289642	199.7737122	117.7169189	142.6951028	2.73173308	0.010755061	Up	

Furthermore, we have introduced Table S8 to present the results of the GO and KEGG enrichment analysis. This table lists the top 50 enriched terms for KEGG pathways and the three categories of GO terms (biological process, molecular function, and cellular component), arranged in ascending order of p-values. A screenshot of Table S8 is provided below:

Level1	Level2	Map_ID	Map_Name	Test	TestAll	Ref	RefAll	Test_Seq	Ref_Seq	P value	FDR	richFactor
Cellular Processes	Cell motility	ko02040	Flagellar assembly	19	424	53	3539	AA0AF6B014AA0DD6IA	5.05141E-06	0.000449575	0.358490566	
Cellular Processes	Cell motility	ko02030	Bacterial chemotaxis	9	424	23	3539	AA0AD61C48AA0DD6IC	0.000614448	0.036242975	0.391304348	
Genetic Information Translation	Ribosome	ko03010	Ribosome	12	424	55	3539	AA0AF6BAC1AA0DF6AX	0.026548033	0.787591658	0.218181818	
Genetic Information Folding, sorting and degradation	Sulfur relay system	ko04122	Sulfur relay system	5	424	17	3539	AA0AD618G7AA0DD6HV	0.043915694	0.85222265	0.294117647	
Cellular Processes	Cellular community - prokaryotes	ko05111	Biofilm formation - Vibrio cf	5	424	18	3539	AA0AB5KATCADA5K1VP	0.050368822	0.85222265	0.277777778	
Cellular Processes	Cellular community - prokaryotes	ko02025	Biofilm formation - Pseudor	4	424	13	3539	AA0AF6B5N2AA0DF6B2	0.060205162	0.85222265	0.307692308	
Metabolism	Lipid metabolism	ko00592	alpha-Linolenic acid metabi	2	424	4	3539	AA0AH3NJNEADA0H3NJ	0.072883028	0.85222265	0.5	
Genetic Information Replication and repair	Base excision repair	ko03410	Base excision repair	4	424	14	3539	AA0AF6B579AA0DF6B0	0.078604283	0.85222265	0.285714286	
Metabolism	Metabolism of terpenoids and polyketides	ko00281	Geraniol degradation	2	424	5	3539	AA0AD61H210AA0DD6FB	0.112016364	0.932633414	0.4	
Environmental Info	Membrane transport	ko02010	ABC transporters	22	424	141	3539	AA0AD61RN6AA2AJOR8	0.113620472	0.932633414	0.156028369	
Cellular Processes	Cellular community - prokaryotes	ko02026	Biofilm formation - Escheric	7	424	35	3539	AA0AD61GJ51AA0DD6IH	0.117198751	0.932633414	0.2	
Metabolism	Metabolism of cofactors and vitamins	ko00780	Biotin metabolism	3	424	11	3539	AA0AD61G72AA0DF7J6I	0.135863986	0.932633414	0.272727273	
Metabolism	Metabolism of terpenoids and polyketides	ko00900	Terpenoid backbone biosyr	3	424	13	3539	AA0AD61FV95AA0DD6IH	0.197552672	0.932633414	0.230769231	
Environmental Info	Signal transduction	ko04068	HIF-1 signaling pathway	2	424	7	3539	AA0A718YDX1AA0DF6B0	0.200608186	0.932633414	0.285714286	
Metabolism	Carbohydrate metabolism	ko00630	Glyoxylate and dicarboxylat	7	424	41	3539	AA0AF6AYNKAA0DD6ITC	0.21356741	0.932633414	0.170731707	
Metabolism	Lipid metabolism	ko00564	Glycerophospholipid metab	5	424	27	3539	AA0AH3NJNEADA0DD6IG	0.215654478	0.932633414	0.185185185	
Metabolism	Carbohydrate metabolism	ko00620	Pyruvate metabolism	9	424	56	3539	AA0AD61FZL4AA0A484VJ	0.221714839	0.932633414	0.160714286	
Metabolism	Lipid metabolism	ko00565	Ether lipid metabolism	1	424	2	3539	AA0AH3NJNEADA0H3NJ	0.225291595	0.932633414	0.5	
Metabolism	Lipid metabolism	ko00591	Linoleic acid metabolism	1	424	2	3539	AA0AH3NJNEADA0H3NJ	0.225291595	0.932633414	0.5	
Metabolism	Xenobiotics biodegradation and metabolism	ko00642	Ethylbenzene degradation	1	424	2	3539	AA0AF6B9E5AA0DD6FB	0.225291595	0.932633414	0.5	
Cellular Processes	Cell growth and death	ko04214	Apoptosis - fly	1	424	2	3539	AA0AF6AY97AA0DF6AX	0.225291595	0.932633414	0.5	
Metabolism	Lipid metabolism	ko00061	Fatty acid biosynthesis	3	424	14	3539	AA0AF6B482AA0DF7J6I	0.230538597	0.932633414	0.214285714	
Metabolism	Metabolism of other amino acids	ko00450	Selenocysteine metabolism	3	424	15	3539	AA0AD61R51AA0DD6IRI	0.264425981	0.985235847	0.2	
Environmental Info	Signal transduction	ko02020	Two-component system	23	424	167	3539	AA0AD61C18AA0SK1VP	0.265681577	0.985235847	0.137724551	
Cellular Processes	Cellular community - prokaryotes	ko02024	Quorum sensing	8	424	52	3539	AA0AB5KATCADA0B5KA	0.279831435	0.991327186	0.153846154	
Metabolism	Lipid metabolism	ko00590	Arachidonic acid metabolis	1	424	3	3539	AA0AH3NJNEADA0H3NJ	0.31816023	0.991327186	0.333333333	
Metabolism	Lipid metabolism	ko01040	Biosynthesis of unsaturate	1	424	3	3539	AA0AF6B228AA0DD6H9	0.31816023	0.991327186	0.333333333	
Genetic Information Translation	RNA transport	ko03013	RNA transport	1	424	3	3539	AA0AF7J8G6AA0DF7J6I	0.31816023	0.991327186	0.333333333	
Metabolism	Xenobiotics biodegradation and metabolism	ko00983	Drug metabolism - other en	3	424	17	3539	AA0AF6B9D9AA0DD6FF	0.333601554	0.991327186	0.176470588	
Metabolism	Energy metabolism	ko00190	Oxidative phosphorylation	6	424	41	3539	AA0AD61F81AA0DF6A1Y	0.367137725	0.991327186	0.146341463	
Metabolism	Metabolism of cofactors and vitamins	ko00790	Folate biosynthesis	4	424	26	3539	AA0AF7JD42AA0DD6IP	0.379911838	0.991327186	0.153846154	
Metabolism	Metabolism of other amino acids	ko00473	D-Alanine metabolism	1	424	4	3539	AA0AD61F81AA0DD6HJ	0.399919298	0.991327186	0.25	
Metabolism	Amino acid metabolism	ko00330	Arginine and proline metab	4	424	27	3539	AA0AD61F82AA0DD6GK	0.408478335	0.991327186	0.148148148	
Metabolism	Carbohydrate metabolism	ko00640	Propanoate metabolism	5	424	35	3539	AA0AD61D83AA0DD6FI	0.411163699	0.991327186	0.142857143	
Metabolism	Metabolism of other amino acids	ko00480	Glutathione metabolism	4	424	28	3539	AA0AD61F81AA0DD6IC	0.436772345	0.991327186	0.142857143	

Additionally, all raw proteomic data must be uploaded to a centralised database, currently no details are provided confirming the submission of this dataset to such a repository.

Response: Thank you for reminding us about this requirement. We have uploaded all the raw proteomic data to the iProX website, a publicly accessible proteomics data repository that is a member of the ProteomeXchange Consortium. The submission details are as follows:

- Website: <https://www.iprox.cn/page/PSV023.html?url=1701656331492mfKE>
- iProX ID: IPX0007671000
- ProteomeXchange ID: PXD047483
- Password: ANM2

We have included this information in the Data Availability section of the revised manuscript.

The methods provided for the construction of the csgD knockout promoter point mutation are insufficient. Please provide the name of the vector and reference for this CRISPR-BTM system as well as a reference for the pH73sacB vector. It should be noted the provided reference, reference 26 the authors highlighted for the

CRISPR-BTM mutagenesis approach (also a Nature publishing group publication) lacks details on this vector so the statement “as previously described” is false.

Response: Thank you for your comment. We have included the company information for the gene-edited strains and updated the references for the CRISPR-BTM system in the supplementary methods under the section “Construction of the mutants with *csgD* gene knockout and a single nucleotide replacement (-44 T>G) in its promoter.” The *csgD* gene knockout mutant was constructed by Ubigene Biosciences Co., Ltd. (Guangzhou, China) using a CRISPR-BTM system, as previously described⁴³⁻⁴⁵. The CRISPR-BTM system used three vectors: CRISPR-B_CR, CRISPR-B_G, and CRISPR-B_D. More details can be found at <https://m.ubigene.com/service/crispr-b/2715.html>. The pH73sacB vector was supplied by Micro Foresight Biosciences Co., Ltd. (Tianjin, China).

Line 216 to 244 the discussion of the AMR determinants is very descriptive without any deeper analysis. Wouldn't a better use of this data be to combine it with the AMR profile data from Line 150 to 169 and use those direct assessments of resistance to confirm the function of AMR determinants and/or reveal AMR determinants being missed using the genomic analysis.

Response: Thank you for your suggestion. We have included a discussion on the correlation between AMR genes and AMR profiles in this paragraph. The updated paragraph is as follows (lines 263-293 of the revised manuscript [clean version]):

“The acquisition of AMR determinants may act as a major driving force for the adaptive evolution of the MDR *S. Typhimurium* lineages. A total of 152 different AMR determinants were identified, and each of the isolates contained numerous AMR determinants (ranging from 20 to 50; Figure S6A-C, Table S1). The six lineages contained different contents of AMR determinants, and the isolates in Lineage L6 contained more AMR determinants than those in other lineages (Figure S6A, Figure

S7). Moreover, the mrdar variants carried more AMR determinants than the smooth colony isolates (Figure S6B), and the mrdar variants in Lineage L6 carried more AMR determinants than the smooth colony strains in L6 (Figure S6C). In particular, the rough colony variants in sublineage L6.5 carried more AMR determinants (Figure S7), including 12 aminoglycoside resistance genes (*AAC(3)-IV*, *AAC(6')-Ib7*, *AAC(6')-Ib-cr6*, *aadA1*, *aadA2*, *aadA3*, *aadA12*, *aadA25*, *strA*, *strB*, *APH(3')-Ia*, and *APH(4)-Ia*), three beta-lactam resistance genes (*bla_{OXA-1}*, *bla_{TEM-1}*, and *bla_{TEM-60}*), two efflux pump-mediated disinfectant resistance genes (*qacEdelta1* and *qacL*), three phenicol resistance genes (*catB3*, *floR*, and *cmlA1*), one rifamycin resistance gene (*arr-3*), three sulfonamide resistance genes (*sul1*, *sul2*, and *sul3*), two tetracycline resistance genes (*tetB* and *tetR*), one trimethoprim resistance gene (*dfrA12*), and two quinolone resistance genes (*oqxA* and *oqxB*) (Figure S7). The differences in resistance phenotypes between mrdar variants and smooth strains were largely due to the presence of specific resistance genes. For example, aminoglycoside resistance genes can confer resistance to gentamicin, and beta-lactam resistance genes can confer resistance to ampicillin. Moreover, the identification of resistance genes such as *floR* and *arr3* suggests that there may be differences in resistance to other antibiotics that have not been tested yet, such as fluphenicol and rifampicin. Resistance to fluoroquinolones is commonly associated with point mutations in quinolone resistance-determining regions (QRDRs) of the *gyrA*, *gyrB*, *parC*, and *parE* genes⁴⁶. The QRDR mutations were mainly observed in Chinese organisms, and the sublineage L6.5 mrdar variants mostly harbored the D87N or D87Y mutation in the *gyrA* gene (Figure S7). However, we did not find differences in ciprofloxacin resistance between mrdar variants and smooth strains, possibly because some smooth strains also harbored the D87N or D87Y mutation in the *gyrA* gene, along with others that harbored the S83L mutation in the *gyrA* gene.”

Line 433/434 the author state “we identified an mrdar colonial phenotype with better multicellular stress resistance.” No stress assays are undertaken within this work to support this claim.

Response: Thank you for your comment. Extensive literature supports that biofilm formation is a preferred survival strategy for the majority of microorganisms³⁻⁹. The formation of microbial biofilms is known to play an important role in cell survival across diverse environments and to serve as a key pathogenic factor in many localized chronic infections³⁻⁹. To ensure accuracy, we have revised the manuscript, such as replacing “enhanced environmental adaptation” and “stress resistance” with “a strong biofilm-forming ability”. Furthermore, the mentioned sentence has been deleted from the revised manuscript, because, as you pointed out, the use of the word “multicellular” is inappropriate for bacteria.

Minor concerns:

Line 107 the author state “We identified a novel variant of *S. Typhimurium* with a rough colony morphology.” Please provide an image to show this morphology compared to a typing strain on a LB plates and/or the selective media used for the initial identification of *S. Typhimurium* used within this study.

Response: Thank you for your comment. The image provided below presents the colony morphology of the novel lineage of *S. Typhimurium* on the modified Swarmagar medium (Statens Serum Institut, Denmark), in comparison to a typing strain.

A. smooth colony morphology

B. rough colony morphology

Line 112/113 the author state “Moreover, this variant was mostly recovered from humans (64.1%), followed by food 113 (27.2%) and the environment (8.7%)” is the difference in the rate of rough colonies observed in human statistically significant?

Response: Thank you for your question. The differences in the rate of rough colonies observed in humans, food, and the environment are indeed statistically significant. The results of chi-square tests for pairwise comparisons are as follows: humans vs. food, $p=9.22e-116$; humans vs. environment, $p=2.01e-78$; food vs. environment, $p=9.39e-43$. We have revised the relevant section (lines 146-152 of the clean version) in the manuscript to provide a clearer description. The updated section reads as follows: “A total of 654 strains exhibiting the rough colony variant were successfully identified and isolated from various samples, including humans (419, 64.1%), food (178, 27.2%), and environment (57, 8.7%). Furthermore, it was observed that the prevalence of the rough colony variant significantly varied across different isolation sources, with environmental sources (40.7%) exhibiting a higher proportion compared to human sources (29.4%) and food sources (27.6%) ($P < 0.05$) (Figure 1C, Table 1).”

Line 117 /118 Please define or provide context on why the morphology of colonies was assessed using Luria-Bertani (LB) agar containing Congo red and Coomassie blue without salt at 28C?

Response: We appreciate your comment. The use of Luria-Bertani (LB) agar with Congo red and Coomassie blue, without salt, at 28°C to assess colony morphology is a well-established and frequently used method in previous studies¹¹⁻¹⁴. We used this method for its proven effectiveness and reliability in previous studies.

Line 138 the author state “noticeable multicellular behavior.” Please re-write this statement are as the use of the word multicellular is confusing when referring to

bacteria.

Response: Thank you for your suggestion. We have revised the manuscript to avoid any mention of “multicellular behavior” throughout the text.

Line 173 the authors state “single nucleotide polymorphisms (SNPs) identified from the 3,951 genome dataset.” More details should be provided here to make it clear the authors are referring to their dataset supplemented with 1739 publicly available genomes. Also, a discussion is needed on why these 1739 publicly available genomes were selected.

Response: Thank you for your suggestion. We have revised the manuscript to provide more details about how we selected the 1,739 publicly available genomes. The revised text is as follows (lines 206-214 of the clean version):

“The maximum likelihood (ML) phylogenetic tree was constructed based on 85,508 core single nucleotide polymorphisms (SNPs) identified from a dataset of 3,951 genomes. This dataset comprised 2,212 strains that we sequenced and analyzed ourselves, and 1,739 strains whose sequencing data were downloaded from the NCBI *Salmonella* Typhimurium Genome database (Table S1). We downloaded all available whole genome sequencing data of *Salmonella* Typhimurium from the NCBI database in August 2021, and after excluding records with incomplete strain background information, such as location and collection dates, we were left with data for 1,739 strains.”

Line 377, please avoid the phrase “multicellular behavior”.

Response: Thank you for your suggestion. We have revised the manuscript to avoid any mention of “multicellular behavior” throughout the text.

Line 423 to 425. The authors state “Despite the burden of *S. Typhimurium* infections on public health systems in China and worldwide, relatively little is known about the genomic changes that contribute to the emergence and microevolution of pathovariants.” This is not true *Salmonella* is arguably the best understood human pathogen and many groups have made significant strides in understanding the microevolution of pathovariants which you cite in the proceeding sentences.

Response: We appreciate your valuable suggestion. We acknowledge that *Salmonella* is one of the most extensively researched human pathogens. In response to your comment, we have removed the mentioned sentence from the revised manuscript.

Line 448 / 449 the authors state “The data from the proteomics analysis and mRNA expression experiments confirmed that the mrdar variants had higher expression levels” re-word as proteomics does not assess expression it assesses protein abundance.

Response: Thank you for your comment. We have revised this sentence according to your suggestion. The updated sentence is in lines 496-498 of the revised manuscript (clean version), which reads as follows: “Data from the proteomics and transcriptomics analysis confirmed that the mrdar variants had a higher abundance of *csgDEFG* operons than the smooth colony strains.”

Line 487 what is meant by “promol secreted proteins”

Response: We apologize for this typographical error. The word “promol” has been removed from the revised manuscript, and the revised sentence reads as follows: “We observed clear upregulation of the expression of flagellar-associated proteins, including FliACDGKLMN and FlgBCDEGIKLM; MCPs, including Tsr, Tar, Trg, and MglB; and

secreted proteins, including SipABCD, sopB, SopE, SopE2, SptP, and InvG (Table S7).”
 (lines 535-538 of the clean version)

Figure 1A) Please increase the size of the pie charts shown as some are unreadable (for example inner monogolia). Also, each region colored should correspond to a single pie chart, some have multiple pie charts which makes this hard to understand.

Response: Thank you for your suggestion. We have revised Figure 1A by enlarging the pie charts. Additionally, in the original Figure 1A, the seven colors represented seven geographical regions of China, with each region comprising several provinces. Each province had a corresponding pie chart, with the province name annotated below. To enhance clarity, we have removed the colors representing geographical regions while retaining the pie charts with province names. The revised Figure 1A is presented below:

Figure 1B and C) increase font size and provide percentage of rough colonies on the figure so readers can independently assess these details.

Response: Thank you for your suggestion. We have increased the font size and included the percentage of rough colony strains in Figure 1B and C. Please find the revised Figure 1B and C below:

Figure 2 C/D. Within the images shown for SF175 significantly more cells are shown is this due to a growth difference between the strains? Base on the methods no normalisation based on optimal density was undertaken. Additionally, are these images representative of multiple biological replicates? It is unclear from the method how many times this assay was undertaken.

Response: Thank you for your comment on Figure 2 C/D. The higher cell count for SF175 is not the result of a growth difference between the strains. To maintain consistency across different strains, we conducted the following steps in our experiment: First, we normalized the sample concentration before fixation by measuring the OD₆₀₀ of bacterial suspensions with an enzyme-linked immunosorbent assay (ELISA). We then adjusted the OD₆₀₀ with PBS to match the concentrations of different strains.

Next, we transferred 100 uL of the adjusted bacterial suspension into 2 mL EP tubes and added 100 uL of glutaraldehyde to each tube. After thorough mixing by pipetting, the samples were fixed overnight at 4 °C.

Then, the fixed samples were sent to the Institute of Microbiology, Chinese Academy of Sciences, where they were prepared for scanning electron microscopy (SEM) by trained staff. During SEM observation and photography, we chose various fields of view and magnifications, capturing at least 10 photos for each strain. This whole experiment was repeated three times.

We have included these details in the supplementary methods under the “Scanning electron microscopy (SEM) examination” section.

Figure 3A is very complex this could be simplified to improve readability, for example does color coding the Chinese regions the strain in this study come from add anything? Additional all the sublineage details are provided yet this is not discussed therefore could be removed to improve readability.

Response: We appreciate your suggestions on how to enhance the readability of Figure 3A. However, we think that the colors indicating regions from which the strains originated are important for visually representing the geographical distribution of the strains. This information is useful in determining whether an outbreak has occurred. Therefore, we respectfully retain these colors.

Furthermore, we agree with your that including sublineage details may not be necessary. We have simplified Figure 3A by removing these details. Please find the revised Figure 3A below:

Crystal violet assays. Within Supplementary Figure 1, the data from Crystal violet assays are given the y axis states at OD₆₀₀ has been used, typically 570 nm is used. Is there a rationale why an alternative OD was used?

Response: Thank you for your comment on the crystal violet assays. In previous studies, different research groups have chosen different OD values for evaluating the biofilm formation ability of *Salmonella* in crystal violet assays. OD₆₀₀ is a common choice⁴⁷⁻⁴⁹. In the preliminary stage of our experiment, we found no significant difference between the readings at OD₆₀₀, OD₅₇₀, and OD₅₈₀. Therefore, we decided to use OD₆₀₀ for our assays.

References

- 1 Liu, H. *et al.* The exopolysaccharide gene cluster *pea* is transcriptionally controlled by RpoS and repressed by AmrZ in *Pseudomonas putida* KT2440. *Microbiol Res* **218**, 1-11 (2019). <https://doi.org/10.1016/j.micres.2018.09.004>

- 2 Fernández-Gómez, P., López, M., Prieto, M., González-Raurich, M. & Alvarez-Ordóñez, A. The
role of the general stress response regulator RpoS in Cronobacter sakazakii biofilm formation.
Food Res Int **136**, 109508 (2020). <https://doi.org/10.1016/j.foodres.2020.109508>
- 3 Yin, W., Wang, Y., Liu, L. & He, J. Biofilms: The Microbial "Protective Clothing" in Extreme
Environments. *Int J Mol Sci* **20** (2019). <https://doi.org/10.3390/ijms20143423>
- 4 Crabbé, A., Jensen, P., Bjarnsholt, T. & Coenye, T. Antimicrobial Tolerance and Metabolic
Adaptations in Microbial Biofilms. *Trends Microbiol* **27**, 850-863 (2019).
<https://doi.org/10.1016/j.tim.2019.05.003>
- 5 Ciofu, O., Moser, C., Jensen, P. & Høiby, N. Tolerance and resistance of microbial biofilms. *Nat
Rev Microbiol* **20**, 621-635 (2022). <https://doi.org/10.1038/s41579-022-00682-4>
- 6 Van Acker, H. & Coenye, T. The Role of Efflux and Physiological Adaptation in Biofilm
Tolerance and Resistance. *J Biol Chem* **291**, 12565-12572 (2016).
<https://doi.org/10.1074/jbc.R115.707257>
- 7 Charron, R., Boulanger, M., Briandet, R. & Bridier, A. Biofilms as protective cocoons against
biocides: from bacterial adaptation to One Health issues. *Microbiology (Reading)* **169** (2023).
<https://doi.org/10.1099/mic.0.001340>
- 8 Ricciardelli, A. *et al.* Environmental conditions shape the biofilm of the Antarctic bacterium
Pseudoalteromonas haloplanktis TAC125. *Microbiol Res* **218**, 66-75 (2019).
<https://doi.org/10.1016/j.micres.2018.09.010>
- 9 Du, B., Wang, S., Chen, G., Wang, G. & Liu, L. Nutrient starvation intensifies chlorine
disinfection-stressed biofilm formation. *Chemosphere* **295**, 133827 (2022).
<https://doi.org/10.1016/j.chemosphere.2022.133827>
- 10 Kirk, M. D. *et al.* World Health Organization Estimates of the Global and Regional Disease
Burden of 22 Foodborne Bacterial, Protozoal, and Viral Diseases, 2010: A Data Synthesis. *PLoS
Med* **12**, e1001921 (2015). <https://doi.org/10.1371/journal.pmed.1001921>
- 11 Osland, A. M. *et al.* Evaluation of Disinfectant Efficacy against Biofilm-Residing Wild-Type
Salmonella from the Porcine Industry. *Antibiotics (Basel)* **12** (2023).
<https://doi.org/10.3390/antibiotics12071189>
- 12 Jain, S. & Chen, J. Antibiotic resistance profiles and cell surface components of *Salmonellae*. *J
Food Prot* **69**, 1017-1023 (2006). <https://doi.org/10.4315/0362-028x-69.5.1017>
- 13 Fàbrega, A. *et al.* Impact of quinolone-resistance acquisition on biofilm production and fitness
in *Salmonella enterica*. *J Antimicrob Chemother* **69**, 1815-1824 (2014).
<https://doi.org/10.1093/jac/dku078>
- 14 Römling, U. *et al.* Occurrence and regulation of the multicellular morphotype in *Salmonella*
serovars important in human disease. *Int J Med Microbiol* **293**, 273-285 (2003).
<https://doi.org/10.1078/1438-4221-00268>
- 15 Michael T. Madigan *et al.* *Brock Biology of Microorganisms*. 14th edn, (Pearson, 2014).
- 16 Didelot, X. & Maiden, M. C. Impact of recombination on bacterial evolution. *Trends Microbiol*
18, 315-322 (2010). <https://doi.org/10.1016/j.tim.2010.04.002>
- 17 Comas, I. *et al.* Out-of-Africa migration and Neolithic coexpansion of *Mycobacterium*
tuberculosis with modern humans. *Nat Genet* **45**, 1176-1182 (2013).
<https://doi.org/10.1038/ng.2744>
- 18 Kaper, J. B., Nataro, J. P. & Mobley, H. L. Pathogenic *Escherichia coli*. *Nat Rev Microbiol* **2**,
123-140 (2004). <https://doi.org/10.1038/nrmicro818>

- 19 Zhang, S. *et al.* SeqSero2: Rapid and Improved Salmonella Serotype Determination Using Whole-Genome Sequencing Data. *Appl Environ Microbiol* **85** (2019). <https://doi.org/10.1128/aem.01746-19>
- 20 Bayliss, S. C. *et al.* Rapid geographical source attribution of Salmonella enterica serovar Enteritidis genomes using hierarchical machine learning. *Elife* **12** (2023). <https://doi.org/10.7554/eLife.84167>
- 21 Norris, T. L. & Baumler, A. J. Phase variation of the *lpf* operon is a mechanism to evade cross-immunity between Salmonella serotypes. *Proc Natl Acad Sci U S A* **96**, 13393-13398 (1999). <https://doi.org/10.1073/pnas.96.23.13393>
- 22 Ellington, M. J. *et al.* The role of whole genome sequencing in antimicrobial susceptibility testing of bacteria: report from the EUCAST Subcommittee. *Clin Microbiol Infect* **23**, 2-22 (2017). <https://doi.org/10.1016/j.cmi.2016.11.012>
- 23 Diep, B. *et al.* Salmonella Serotyping; Comparison of the Traditional Method to a Microarray-Based Method and an in silico Platform Using Whole Genome Sequencing Data. *Front Microbiol* **10**, 2554 (2019). <https://doi.org/10.3389/fmicb.2019.02554>
- 24 Ingle, D. J. *et al.* Evolutionary dynamics of multidrug resistant Salmonella enterica serovar 4,[5],12:i:- in Australia. *Nat Commun* **12**, 4786 (2021). <https://doi.org/10.1038/s41467-021-25073-w>
- 25 Kingsley, R. A. *et al.* Genome and transcriptome adaptation accompanying emergence of the definitive type 2 host-restricted Salmonella enterica serovar Typhimurium pathovar. *mBio* **4**, e00565-00513 (2013). <https://doi.org/10.1128/mBio.00565-13>
- 26 Mather, A. E. *et al.* New Variant of Multidrug-Resistant Salmonella enterica Serovar Typhimurium Associated with Invasive Disease in Immunocompromised Patients in Vietnam. *mBio* **9** (2018). <https://doi.org/10.1128/mBio.01056-18>
- 27 Mather, A. E. *et al.* Distinguishable epidemics of multidrug-resistant Salmonella Typhimurium DT104 in different hosts. *Science* **341**, 1514-1517 (2013). <https://doi.org/10.1126/science.1240578>
- 28 Okoro, C. K. *et al.* Intracontinental spread of human invasive Salmonella Typhimurium pathovariants in sub-Saharan Africa. *Nat Genet* **44**, 1215-1221 (2012). <https://doi.org/10.1038/ng.2423>
- 29 Petrovska, L. *et al.* Microevolution of Monophasic Salmonella Typhimurium during Epidemic, United Kingdom, 2005-2010. *Emerg Infect Dis* **22**, 617-624 (2016). <https://doi.org/10.3201/eid2204.150531>
- 30 Arai, N. *et al.* Phylogenetic Characterization of Salmonella enterica Serovar Typhimurium and Its Monophasic Variant Isolated from Food Animals in Japan Revealed Replacement of Major Epidemic Clones in the Last 4 Decades. *J Clin Microbiol* **56** (2018). <https://doi.org/10.1128/JCM.01758-17>
- 31 Hayden, H. S. *et al.* Genomic Analysis of Salmonella enterica Serovar Typhimurium Characterizes Strain Diversity for Recent U.S. Salmonellosis Cases and Identifies Mutations Linked to Loss of Fitness under Nitrosative and Oxidative Stress. *MBio* **7**, e00154 (2016). <https://doi.org/10.1128/mBio.00154-16>
- 32 Lu, X. *et al.* Epidemiologic and genomic insights on *mcr-1*-harbouring Salmonella from diarrhoeal outpatients in Shanghai, China, 2006-2016. *EBioMedicine* **42**, 133-144 (2019). <https://doi.org/10.1016/j.ebiom.2019.03.006>

- 33 Pulford, C. V. *et al.* Stepwise evolution of Salmonella Typhimurium ST313 causing bloodstream infection in Africa. *Nat Microbiol* **6**, 327-338 (2021). <https://doi.org/10.1038/s41564-020-00836-1>
- 34 Van Puyvelde, S. *et al.* An African Salmonella Typhimurium ST313 sublineage with extensive drug-resistance and signatures of host adaptation. *Nat Commun* **10**, 4280 (2019). <https://doi.org/10.1038/s41467-019-11844-z>
- 35 Hammarlöf, D. L. *et al.* Role of a single noncoding nucleotide in the evolution of an epidemic African clade of Salmonella. *Proc Natl Acad Sci U S A* **115**, E2614-e2623 (2018). <https://doi.org/10.1073/pnas.1714718115>
- 36 Shen, B., Dong, X., Yuan, B. & Zhang, Z. Molecular Markers of MDR of Chemotherapy for HSCC: Proteomic Screening With High-Throughput Liquid Chromatography-Tandem Mass Spectrometry. *Front Oncol* **11**, 687320 (2021). <https://doi.org/10.3389/fonc.2021.687320>
- 37 Jiang, Y. *et al.* Proteomics identifies new therapeutic targets of early-stage hepatocellular carcinoma. *Nature* **567**, 257-261 (2019). <https://doi.org/10.1038/s41586-019-0987-8>
- 38 Huang, C. *et al.* Proteogenomic insights into the biology and treatment of HPV-negative head and neck squamous cell carcinoma. *Cancer Cell* **39**, 361-379.e316 (2021). <https://doi.org/10.1016/j.ccell.2020.12.007>
- 39 Clark, D. J. *et al.* Integrated Proteogenomic Characterization of Clear Cell Renal Cell Carcinoma. *Cell* **179**, 964-983.e931 (2019). <https://doi.org/10.1016/j.cell.2019.10.007>
- 40 Enríquez-Vázquez, D. *et al.* Non-invasive electromechanical assessment during atrial fibrillation identifies underlying atrial myopathy alterations with early prognostic value. *Nat Commun* **14**, 4613 (2023). <https://doi.org/10.1038/s41467-023-40196-y>
- 41 Qiu, J. *et al.* FAM210A is essential for cold-induced mitochondrial remodeling in brown adipocytes. *Nat Commun* **14**, 6344 (2023). <https://doi.org/10.1038/s41467-023-41988-y>
- 42 Agarwal, S. *et al.* VapBC22 toxin-antitoxin system from Mycobacterium tuberculosis is required for pathogenesis and modulation of host immune response. *Sci Adv* **6**, eaba6944 (2020). <https://doi.org/10.1126/sciadv.aba6944>
- 43 Tang, X. *et al.* Structural basis for bacterial lipoprotein relocation by the transporter LolCDE. *Nature structural & molecular biology* **28**, 347-355 (2021). <https://doi.org/10.1038/s41594-021-00573-x>
- 44 Yu, Z. *et al.* Membrane translocation process revealed by in situ structures of type II secretion system secretins. *Nat Commun* **14**, 4025 (2023). <https://doi.org/10.1038/s41467-023-39583-2>
- 45 Hu, J. *et al.* Dietary D-xylose promotes intestinal health by inducing phage production in Escherichia coli. *NPJ Biofilms Microbiomes* **9**, 79 (2023). <https://doi.org/10.1038/s41522-023-00445-w>
- 46 Turner, A. K., Nair, S. & Wain, J. The acquisition of full fluoroquinolone resistance in Salmonella Typhi by accumulation of point mutations in the topoisomerase targets. *J Antimicrob Chemother* **58**, 733-740 (2006). <https://doi.org/10.1093/jac/dkl333>
- 47 Thames, H. T. *et al.* Salmonella Biofilm Formation under Fluidic Shear Stress on Different Surface Materials. *Foods* **12** (2023). <https://doi.org/10.3390/foods12091918>
- 48 Baugh, S., Ekanayaka, A. S., Piddock, L. J. & Webber, M. A. Loss of or inhibition of all multidrug resistance efflux pumps of Salmonella enterica serovar Typhimurium results in impaired ability to form a biofilm. *J Antimicrob Chemother* **67**, 2409-2417 (2012). <https://doi.org/10.1093/jac/dks228>

- 49 Ramachandran, G., Aheto, K., Shirtliff, M. E. & Tennant, S. M. Poor biofilm-forming ability and long-term survival of invasive *Salmonella* Typhimurium ST313. *Pathog Dis* **74** (2016). <https://doi.org:10.1093/femspd/ftw049>

REVIEWER COMMENTS

Reviewer #1 (Remarks to the Author):

Xiang et al. have described the spread of a particular variant of Salmonella Typhimurium heavily throughout China and a little bit in the rest of the world (39 of 654 isolates). It is a thorough body of work and they have done a good job establishing this as a new lineage, tracing its origins in China and characterizing the differences between the rough variant and the smooth relative that also occurs in the L6.5 lineage. The authors have done a satisfactory job answering my questions, they have improved the description of numerous sections and the Figures are better laid out and much clearer now.

However, I still feel think they are missing important points when it comes to the key mutation contributing to the rough mrdar morphotype. Apologies for repeating parts of this from my previous review, but I felt I need to explain it better.

1. Transcriptional regulation of the csgDEFG operon has been very well described in the past. It is probably one of the best characterized regulatory networks in Gram-negative enteric bacteria. It is possible that the authors did not know this before.
2. For normal regulation of csgDEFG, RNA transcription is controlled by the RpoS sigma factor and the activation of transcription is highly sensitive to osmolarity, nutrient levels in the cell, temperature (below 30°C only) and growth phase (i.e., activated as cells start to enter stationary phase). Osmolarity feeds into the system through the DNA binding protein OmpR. There have been 2 OmpR binding sites identified that are involved in activation of csgDEFG transcription via RpoS.
3. The mutation that the authors found (-44 T>G) in their mrdar lineage was first characterized back in 1998 (Romling et al. – in your reference list already). This -44 point mutation is located in the high affinity OmpR binding site. The consequence of this mutation in *S. Typhimurium* is that csgDEFG transcription now becomes independent of RpoS but dependent on sigma70 – the dominant “housekeeping” sigma factor in the cell.
4. What this mutation causes in phenotype is curli/cellulose/biofilm formation that is independent of temperature (28°C or 37°C), earlier in the growth phase (not restricted to stationary phase anymore), not as sensitive to osmolarity, that is based on higher levels of csgDEFG transcription for longer. The biofilm/wrinkly spreader morphology that the authors have described and is shown in Figure 2 (SF175) was first shown by Ute Romling in 1998, attributed to the -44 T>G mutation

The authors need to be clear on these points and not give the impression that they have ‘discovered’ this mutation or that it is new. What IS new (and very cool) is that they have discovered a lineage of *S. Typhimurium* with this mutation that is present within a “wild” population of *S. Typhimurium* and that appears to be spreading within China and the rest of the world.

Why is this particular mrdar variant spreading? What fitness advantages/costs are associated with the -44 T>G mutation? and What is the role of other SNPs identified in this L6.5 lineage? These are things that need to be investigated further.

Other Important Points

1. I don’t think the last sentence in the Abstract is necessary
2. Lines 324-327 – I found this sentence confusing – needs to be reworded
3. Lines 601-603 – As I stated above, this mutation is well-established, and its effects have been well-characterized. If anything, it’s the other mrdar strains that do not have this mutation that should be investigated further. Is there a chance that these other strains could have sequencing errors with regards to the -44 T>G SNP?

Reviewer #2 (Remarks to the Author):

The authors have addressed most of the comments on the first version.

However, there are still important issues to assess, prior to consider this manuscript for publication in Nature Communications.

Please find the comments to this second version in the attached document.

Reviewer #3 (Remarks to the Author):

The authors have addressed many of my concerns yet there are several outstanding issues associated with this study including:

CRITICAL CORRECTION REQUIRED:

Please update the proteomic methods to accurately reflect the data associated with iProX ID associated with IPX0007671000. Examination of the data uploaded reveals this data was collected on a Bruker instrument not a Thermo HF-X with the HyStar report (see attached) also highlighting the LC is a Evosep not the reported Easy-nLC 1200. Either the incorrect dataset has been uploaded or description of the method is grossly inaccurate.

MINOR CORRECTIONS:

“However, in our analysis of proteomic data from 40 strains, we found no statistically significant difference in the expression level of RpoS between rough and smooth colony strains”.

to

“However, in our analysis of proteomic data from 40 strains, we found no statistically significant difference in the abundance of RpoS between rough and smooth colony strains”

As I highlighted in my previous reviewer comments proteomics provides measurements of abundance not expression.

Additional corrections needed for the proteomic methods, please fix the following oversights:

- It is currently stated “We discarded the supernatant and obtained the bacterial samples as the sediments in the EP tubes. We added 125 μ L of lysate to each EP tube, vortexed them for 30 seconds” What are the cells resuspend in to generate the 125 μ L of lysate?

- Please update the methods and provide the complete name of chemicals used including DTT – Dithiothreitol

IAA -iodoacetamide

- Please change “detergent DTT” to DTT as DTT is not a detergent.

- Please define what the buffer composition of the “UA buffer” used for proteomic sample prep was. Additionally, this method appears to be similar to the FASP (doi: 10.1038/nmeth.1322.) protocol, please reference the method used for this sample preparation approach.

- Please change NH_4HCO_3 to NH_4HCO_3

- Please provide the vendor for the trypsin used for proteomics.

- Please provide the length of the analytical gradient used for peptide separation. Currently the methods states “The peptide was separated on a C18 Analytical Column (Thermo Scientific, ES802,

1.9 μm , 75 μm *20 cm) with a linear gradient of buffer B (84% acetonitrile in 0.1% formic acid) at a flow rate of 300 nl/min.” but no details on the composition of Buffer A or the length/time of the analytical gradient are provided.

- Please provide the details of the Uniprot database used for proteomic database searching including the Uniprot accession number.
- Please also change the title within the supplementary from “Proteomics sequencing of the rough and smooth colony strains” to “Proteomic analysis of the rough and smooth colony strains”
Proteomics is not sequencing samples per say.

NCOMMS-23-40287A — Second revision

The authors have addressed all the comments and in most of cases the answers are satisfactory.

There are however some points that need further consideration.

The present study revises a vast collection of *Salmonella* Typhimurium isolated in China through recent years, where a high prevalence of the MRDAR phenotype and a high proportion of strains resistant to ≥ 5 families of antibiotics can be highlighted.

The authors have observed that in the majority of the cases, the rough phenotype is associated to a particular mutation of the *csgD* promoter. Compared to *S. Typhimurium* genomes from international databases, this mutation is proportionally much more frequent in the Chinese bacterial population vs international genomes available online. Thus, the authors hypothesize that the strains from which the genomes did not contain the mutated operon, shall probably be smooth.

However, only Chinese strains were available to compare the presence of the mutated promoter of *csgD* with the rough/smooth phenotype and the capacity of producing biofilm. Due to this limitation, the authors could not confirm the relationship between the presence/absence of the *csgD* mutation with the aspect of the colonies (rough/smooth) in foreign strains.

Also, not all the MRDAR from China carried the mutated operon, and not all the strains with the mutated operon tested expressed an MRDAR phenotype.

Therefore, the authors can argue that from the observations in the strains from China analyzed, an association can be suspected, and laboratory analyses showed that for the tested strains, the phenotype changed when the operon changed. But further investigations, and notably the phenotype information from the foreign strains should be needed to confirm the hypothesis.

The provided data allow establishing that the *S. Typhimurium* population in China is quite diverse but remains well differentiated from other populations identified in different regions of the globe. Also, that Chinese *S. Typhimurium* are frequently MRDAR, produce biofilm and very often carry a particular mutation of the *csgD* promoter, which may be involved in the described phenotype.

The description of a new lineage remains unclear, as the suggested L6 sublineage includes strains from very distant MLST types (ST19, ST34, ST36). Accordingly, it is suggested to change the title to something like: « High frequency of rough morphology and enhanced biofilm producing capacity in *Salmonella enterica* serovar Typhimurium in China and the possible role of the *csgD* operon in these features. »

Following are more specific comments on the second version of this manuscript.

1. It is suggested to compile in one single table, where each line would correspond to one genome/strain, all the information about that genome/strain.

INTRODUCTION

2. Line 113-114:

Since the term “extensive MDR” appears here for the first time in the text, it should be here defined.

TABLES

3. Sentence in lines 143-144 and Table 1, Figure 1.

The percentages shown in the table and the figure are calculated on the total number of isolates of the same phenotype. But to compare prevalence by years, the percentages should be calculated based on the total number of isolates in the year, and this way, the prevalence of the rough phenotype, has quite decreased since 2014. This is how figure 1B would look like if the percentages are calculated over the totals of each year:

This same way of expressing prevalence is suggested for Figure 1C.

4. Tables with dataset for Figure 1.

Tables with datasets for figures 1A to 1F are identical and so they don't correspond to the 4 different sub-figures. It is suggested to prepare one single table containing all the data shown in Figure 1.

5. Tables with dataset for Figure 3.

Also, in that repeated table (3A-3C), there are regions of China and China itself described in the same column, which suggests a possible redundancy in the values shown in the figure.

RESULTS

6. Lines 166-168:

This reviewer pointed out that the rough phenotype in *Salmonella* is often shown after subcultures, in comment #2.

The authors answered that the rough phenotype in their strains existed already at the initial isolation.

However, the authors describe an increase in this phenotype in lines 166-168 of the new version of the manuscript.

This is in agreement to this reviewer's comment, that the roughness of *Salmonella* has a tendency to appear (or increase) over generations reproduced *in vitro*.

DISCUSSION

7. Lines 420 to 422:

Agreed with the first sentence:

« The major finding of this study is the identification of a macro-rdar colony morphology of *S. Typhimurium* that emerged and became prevalent in China. »

Not agreed with the second sentence:

« We provided a comprehensive description of the genomic and proteomic characteristics of these mrdar variants and observed that they formed a novel lineage on the phylogenetic tree. »

There are some strains OUTSIDE the lineage with these characteristics and strains INSIDE the lineage without these properties. There are strains of different ST (ST34, ST36, which are at 4/7 AD on 7g-MLST and > 400 AD on cgMLST) in the lineage L6, even in L6.5. It is striking that genomes of so different MLST types can constitute a lineage, while other genomes from those ST remain so distant in the tree, that are considered as different lineages (also with mixed STs). How do the authors explain the structure of the tree? What are the genomic traits of the Chinese genomes when compared to those obtained from online databases?

Following the authors' responses to the comments in the first review, it is clearer that what is here described are phenotypic — mrdar — variants; but not a lineage.

8. Monophasic ≠ ST34 ≠ lack of *fljA/fljB*

As the authors know, these three concepts are not synonymous: not all monophasic strains belong to ST34, nor do they owe their phenotype to the lack of *fljA/fljB*; not all strains of ST34 are monophasic, nor do they lack *fljA/fljB*; not all strains lacking *fljA/fljB* belong to ST34. The referred data in figure

9. Biofilm formation imaging

This reviewer asked the authors to show the confocal imaging, or the EPS and fibrillar network results (like figures 2 and S1) of all the other 19 strains tested of Rough and Smooth phenotype, as a variability in biofilm formation was expected.

The authors did not reproduce the imaging as suggested, but they performed crystal violet and pellicle formation tests. The new figure S2 (B and C) is not very clear, but allows to observe the expected variability, as rough strains R12 and R20 show similar degrees of biofilm and pellicle formation as smooth strains S2, S8 and S18. This is why the confocal imaging in all of them would have been very important.

The authors have performed the assays also in strains from other lineages than L6. However, their results are only described in a table, and not shown in an image.

10. Line 192:

Please avoid using terms as “important” when describing results. Such valorization expressions should be left to the Discussion, and only if well justified.

11. Line 196:

Like in the previous comment, avoid using valuating expressions in the Results. Because the authors say that « [...] the mrdar variants had “more severe” AMR profiles than the smooth colony isolates, but if an evaluation of clinical severity must be expressed on these results, it must be noted that none of the antibiotics referred in this paragraph (tetracycline, ampicillin, sulfisoxazole, nalidixic acid, chloramphenicol, trimethoprim/sulfamethoxazole or gentamycin) is of choice to treat *Salmonella*. Cefoxitin, ciprofloxacin and azithromycin are the main therapeutic options to treat *Salmonella* infections, and against those three antibiotics, the smooth colonies were more resistant than the rough ones.

12. Population structure of global *S. Typhimurium*, including Chinese isolates

The authors did not detail the number of genes included in the core genome on which the SNPs were identified. Taking in account that the number of genomes from China (3,851, 57.8% of the total of 3,851 shown in table 450756_1_data_set_8614497_s83rx8) largely outnumber the genomes from other countries, the core genome could be disregarding important genes that could have provided a higher variability to the phylogenetic analysis.

13. Line 250:

« L6 lineage [...] has been particularly successful at global dissemination, comprising 44,4% of all isolates in this study. »

Since the authors explain that this L6 comprises the latest genomes, this “success” in dissemination must take in account that sequencing all over the world has increased dramatically in recent years, so it is to be expected that most genomes will be recent. This “success” is a value added to the result that should be left to the Discussion.

14. Line 258:

« Furthermore, the mrdar variants “seemingly” also presented in other regions of the world [...]».

The authors should avoid hypothesizing in the Results. Also, there are no phenotypic information on the morphology of strains abroad since the study only included phenotypic analyses on Chinese strains.

15. Lines 263-264:

The sentence should be kept for the Discussion. Those are not results of the present study.

The authors should describe what “AMR determinants” are. Genes? Mutations (including different mutations in the same gene?).

16. Line 264-266:

Table S1 does not content the information expressed in the referring line.

Figure S6A and parenthesis in line 266: according to the authors, all strains carried AMR determinants, which is strange, given that the wildtype phenotype of *S. Typhimurium* is pan-susceptible. Also, the authors describe that there is at least one strain in L1 with > 40 resistance AMR determinants and at least one strain in L6 with > 50. Those data require further . The authors may have considered as different determinants the

different denominations, or the different <100% hits to BLAST hits on the same genomic region. Those numbers of AMR determinants are unprecedented. The number of genes of resistance by lineage depends greatly on the number of genomes in the lineage, which renders figure S6A very difficult to evaluate.

17. Line 276:

When referring to disinfectants, it is strongly suggested not to use terms of “resistance”, but of “decreased susceptibility”, as there are not cut-off values, yet.

18. Lines 290-293:

It should be noted that strains from the sublineage L1.3, where strains were mostly smooth, not only carried a mutation on *gyrA* (D87N) but also a second mutation, on *parC* (S80R). Actually, while single QRDR mutations (associated to decreased susceptibility to ciprofloxacin) were observed in more rough strains (n = 604; 92.4% of all rough) than smooth strains (n = 190; 12.2% of all smooth strains), double or triple or triple QRDR mutations, associated to high level of resistance to ciprofloxacin, were ten times more frequent among smooth (n = 91; 5.84% of all smooth) than in rough (n = 9; 1.37% of all rough) isolates, according to table 450756_1_data_set_8614497_s83rx8. Thus, a higher level of resistance to ciprofloxacin could be expected among those smooth-colony strains than among the L6.5, presenting with only one *gyrA* mutation. But globally, a higher proportion of rough strains than smooth strains should have shown lower levels of susceptibility to ciprofloxacin (due to the one mutation in *gyrA*), as 1,227 smooth strains (78% of all smooth strains) did not present any QRDR mutations.

19. Identification of AMR- and virulence-associated plasmids

It would be important to show the Inc of plasmids associated with the different AMR determinants appearing in table 450756_1_data_set_8614497_s83rx8.

In that table, what are the figures in the AMR determinants? Percentage of identity? Percentage of coverage? For some genetic determinants the figures shown are < 90%. Although there is not a standard threshold, experience shows that the predicted associated phenotype rarely appears when the identified gene is lower than 90%, sometimes even if < 95%.

In addition to that, some determinants described in this manuscript have no activity in *Salmonella*. For example, the authors include among the *mcr* genes (associated to resistance to colistin) the gene *mcr9*, which is known NOT to cause this resistance in *Salmonella*. And they identified *mefB* in three strains, of which two showed to be susceptible to azithromycin, and the one being resistant carried also the gene *mphA* that is well known to cause this phenotype in *Salmonella*. And they.

These considerations should be highlighted. Otherwise, the “take-home” message would be the high number of AMR determinants observed, when in reality the AMR profiles were much less striking.

20. Pangenome analysis

It is surprising that the core genome of all the *S. Typhimurium* in the study was only 1,821 genes (roughly 3274 if adding the soft-core genes), while the cgMLST for the entire genus *Salmonella* is of 3,002 genes

(<http://www.genome.org/cgi/doi/10.1101/gr.251678.119>,
<https://doi.org/10.3389/fmicb.2017.01345>).

21. Lines 364 and 642 / Table S5:

Table S5 doesn't explain which were there 20 S + 20 R "selected strains". And nowhere in the text appear the criteria for this selection.

22. Line 422:

« [...] and observed that they formed a novel lineage on the phylogenetic tree. »

The mrdar colony-forming strains may constitute a variant of *S. Typhimurium* and its monophasic variant. But they do not constitute an evolutionary lineage. It is very surprising that such a restrict sublineage like L6.5 may include genomes of ST19, ST34 and ST36.

23. Line 450:

« The majority of the mrdar variants were found in sublineage L6.5. ». To be noted that L6.5 contains, in general, the highest number of strains (n = 708, rough an smooth together) from this study, while there lineages with 0, 1 or 2 isolates with the morphotype tested.

In fact, since only the phenotype of the strains from this study have been studied, the discussion should be focused on the presence/absence of the mutations that the authors associate with the described phenotypes: not about the evolution of "rough/mrdar" but about the evolution of "-44G>T substitution in the *csgD* gene promoter".

24. Line 608:

The authors should define "severe MDR" profiles. Also, while it is true that for instance, the proportion of ESBL or AmpC among the rough colonies was higher than among the smooth strains, the difference was not that striking: 17,7% of rough showed resistance to 3dr generation cephalosporins, while 14,7% of smooth colonies did. Is this a significant difference?

There were no carbapenem-resistant strains identified in the study, and only 10 were resistant to the three molecules used for treating infections by *Salmonella*, of which 7 were smooth and three were rough.

25. Line 609:

The definition of a new lineage, where ST34 and ST19 or ST36 coexist is difficult to understand.

METHODS

26. AMR gene and VF detection.

CARD is a very exhaustive but not curated database, and the use of its results must be done with care. There are plenty of genes with proven absence of activity, in all bacteria or at least in *Salmonella*.

Also, the threshold of 80% of coverage and identity is too low. Many genes have similar structures, and this threshold can bring to false identifications. Standard protocols suggest not to go under 95%.

Responses to Reviewers' Comments

We sincerely appreciate the reviewers for their insightful and constructive feedback. Carefully incorporating their advice into the revised manuscript has significantly enhanced the quality of our work.

Below, we provide a detailed explanation of how we have addressed each comment in the revision of our manuscript. The reviewers' comments are presented in bold, followed by our responses in standard font. All changes have been tracked in the revised manuscript. We also offer a clean version of the revised manuscript. The referenced line numbers in our responses correspond to those in this clean version.

Responses to Reviewer #1

Xiang et al. have described the spread of a particular variant of Salmonella Typhimurium heavily throughout China and a little bit in the rest of the world (39 of 654 isolates). It is a thorough body of work and they have done a good job establishing this as a new lineage, tracing its origins in China and characterizing the differences between the rough variant and the smooth relative that also occurs in the L6.5 lineage. The authors have done a satisfactory job answering my questions, they have improved the description of numerous sections and the Figures are better laid out and much clearer now.

Response:

We sincerely appreciate your positive feedback on our manuscript. We agree with the insightful comments you provided, and we have prepared a point-by-point response below, outlining the specific revisions we made to the manuscript based on your recommendations. These revisions have undoubtedly improved the quality and clarity of our work. Once again, we would like to express our heartfelt gratitude for your time and effort in reviewing our manuscript. Your continued support is immensely

appreciated.

However, I still feel think they are missing important points when it comes to the key mutation contributing to the rough mrdar morphotype. Apologies for repeating parts of this from my previous review, but I felt I need to explain it better.

1. Transcriptional regulation of the csgDEFG operon has been very well described in the past. It is probably one of the best characterized regulatory networks in Gram-negative enteric bacteria. It is possible that the authors did not know this before.

2. For normal regulation of csgDEFG, RNA transcription is controlled by the RpoS sigma factor and the activation of transcription is highly sensitive to osmolarity, nutrient levels in the cell, temperature (below 30°C only) and growth phase (i.e., activated as cells start to enter stationary phase). Osmolarity feeds into the system through the DNA binding protein OmpR. There have been 2 OmpR binding sites identified that are involved in activation of csgDEFG transcription via RpoS.

3. The mutation that the authors found (-44 T>G) in their mrdar lineage was first characterized back in 1998 (Romling et al. – in your reference list already). This -44 point mutation is located in the high affinity OmpR binding site. The consequence of this mutation in S. Typhimurium is that csgDEFG transcription now becomes independent of RpoS but dependent on sigma70 – the dominant “housekeeping” sigma factor in the cell.

4. What this mutation causes in phenotype is curli/cellulose/biofilm formation that is independent of temperature (28°C or 37°C), earlier in the growth phase (not restricted to stationary phase anymore), not as sensitive to osmolarity, that is based on higher levels of csgDEFG transcription for longer. The biofilm/wrinkly spreader morphology that the authors have described and is shown in Figure 2 (SF175) was first shown by Ute Romling in 1998, attributed to the -44 T>G mutation

The authors need to be clear on these points and not give the impression that they

have ‘discovered’ this mutation or that it is new. What IS new (and very cool) is that they have discovered a lineage of *S. Typhimurium* with this mutation that is present within a “wild” population of *S. Typhimurium* and that appears to be spreading within China and the rest of the world.

Response:

Thank you for your insightful comments and guidance on our research. We acknowledge the extensive understanding of factors that would activate CsgD transcription, including low osmolarity, low temperatures, and restrictive nutritional conditions¹. These environmental factors, through the action of RNA polymerase sigma factor RpoS (σ 38), various transcription factors, and small nucleotides, regulate the expression of CsgD^{2,3}. To date, at least 14 transcription factors have been identified as being involved in the regulation of the *csgD* promoter, including BasR, BtsR, CpxR, Cra, Crl, CRP, CsgD, H-NS, IHF, MlrA, MqsA, OmpR, RstA, and RcdA^{4,5}.

The -44 position in the *csgD* promoter is the binding site not only for OmpR, but also for several other transcription factors such as H-NS, CpxR, IHF, CRP, RcdA, and RsrA⁶. Mutations at this site can affect various environmental signals, thereby leading to multifaceted response changes within this complex regulatory network. Previous studies have not fully elucidated the overall impact of point mutations at this site on the bacterial regulatory network.

To address these points and for clearer communication, we have made the following modifications in the discussion section:

Original: The typical rdar morphotype was first described by Römmling et al. during their research on the multicellular behavior of *S. Typhimurium* ATCC 14028.

Revised: The typical mrdar morphotype was first described by Römmling et al. during their investigation of the multicellular behavior of two unrelated *S. Typhimurium* strains⁷. They observed that these two strains spontaneously developed this phenotypic variation during laboratory passage and identified mutations in the *csgD* promoter as

the underlying cause. Specifically, one strain had a single T insertion after position -17, while the other had a -44 T>G mutation in the *csgD* promoter. In our study, we report a novel lineage of *S. Typhimurium* carrying the -44 T>G mutation in the *csgD* promoter within a wild *S. Typhimurium* population. Notably, this lineage appeared to be spreading not only within China but also across multiple regions of the world. (lines 427-436 in the clean version of the revised manuscript)

Why is this particular mrdar variant spreading? What fitness advantages/costs are associated with the -44 T>G mutation? and What is the role of other SNPs identified in this L6.5 lineage? These are things that need to be investigated further.

Response:

We wholeheartedly agree with your suggestions and have already incorporated them into our future research plans, as follows: “Further international epidemiological studies are required to understand the spread of the mrdar variant, assess the selective benefits and drawbacks of the -44 T>G mutation, and elucidate the role of other SNPs identified in the L6.5 lineage.” (lines 618-621 in the clean version of the revised manuscript)

Other Important Points

1. I don’t think the last sentence in the Abstract is necessary

Response:

Thank you very much for your suggestion. The mentioned sentence has been appropriately removed from the Abstract section in the revised manuscript.

2. Lines 324-327 – I found this sentence confusing – needs to be reworded

Response:

Thank you for providing valuable feedback. We have revised the sentence accordingly,

as follows:

Original: The plasmid pYY-2016-057-2 belonging to the IncHI2 replicon 183,430 bp in size was prevalent in sublineage L6.5, which is similar to a previous plasmid p2016089-170 (accession CP090536) identified from a *Salmonella* 4,[5],12:i:- strain in China.

Revised: The plasmid pYY-2016-057-2, which was frequently found in the L6.5 sublineage, measured 183,430 bp and belonged to the IncHI2 replicon. Its structure was similar to that of plasmid p2016089-170 (accession CP090536), which was previously identified from a *Salmonella* 4,[5],12:i:- strain in China. (lines 327-331 in the clean version of the revised manuscript)

3. Lines 601-603 – As I stated above, this mutation is well-established, and its effects have been well-characterized. If anything, it’s the other mrdar strains that do not have this mutation that should be investigated further. Is there a chance that these other strains could have sequencing errors with regards to the -44 T>G SNP?

Response:

Thank you for your insightful comment. In response, we performed Sanger sequencing to verify the presence or absence of the -44 T>G mutation in strains that exhibited the mrdar phenotype but were previously identified as lacking this mutation, as well as those with a smooth phenotype that were previously identified as harboring the mutation. Inconsistencies were found in 33 strains (please refer to the table below), resulting in an error rate of 1.49% (33/2212), which is comparable to the reported error level for base substitution in next-generation sequencing⁸. The verified sequencing data confirmed that all mrdar strains within the L6.5 lineage do indeed carry the -44 T>G mutation. Accordingly, we have updated Figure 5B (provided below) to present the verified results.

TreeID	Lineage	Sublineage	Status	S-R phenotype	csgD promoter-44G>T	Check result : csgD promoter-44G>T
SH16SF1301	L6	L6.5	This work	Smooth	100	0
SH16SF1310	L1	L1.7	This work	Rough	0	100
SH17SF1557	L6	L6.3	This work	Rough	0	100
SH17SF1563	L6	L6.5	This work	Smooth	100	0
SH17SF1577	L6	L6.3	This work	Rough	0	100
SH17SF292	L6	L6.5	This work	Smooth	100	0
SH18SF797	L6	L6.5	This work	Rough	0	100
SH18SF823	L6	L6.5	This work	Rough	0	100
2015SM082	L6	L6.5	This work	Smooth	100	0
41-Sa-09D101	L6	L6.5	This work	Rough	0	100
41-Sa-12D114	L6	L6.5	This work	Rough	0	100
41-Sa-15D125	L6	L6.5	This work	Rough	0	100
HN-Sa-06019	L6	L6.5	This work	Rough	0	100
SH09G113	L6	L6.5	This work	Smooth	100	0
SH11G1268	L6	L6.5	This work	Rough	0	100
SH12G1031	L6	L6.5	This work	Rough	0	100
SH14G1809	L6	L6.5	This work	Rough	0	100
SH15G1428	L6	L6.5	This work	Smooth	100	0
SH15G1745	L6	L6.5	This work	Rough	0	100
SH15G1766	L6	L6.5	This work	Smooth	100	0
SH15G1927	L6	L6.5	This work	Smooth	100	0
SH15G2328	L6	L6.5	This work	Rough	0	100
SH17G2228	L6	L6.3	This work	Rough	0	100
SH17G2249	L6	L6.5	This work	Smooth	100	0
SH17G341	L6	L6.5	This work	Smooth	100	0
SH17G347	L3	L3.7	This work	Rough	0	100
SH18G2595	L6	L6.5	This work	Smooth	100	0
SH18G2603	L6	L6.5	This work	Smooth	100	0
SH18G2604	L6	L6.5	This work	Smooth	100	0
SH12G495	L6	L6.5	This work	Smooth	100	0
SH14G381	L6	L6.3	This work	Rough	0	100
SH15G950	L5	L5.2	This work	Rough	0	100
SH16G364	L5	L5.2	This work	Rough	0	100

Thirty-three strains with inconsistent results between Sanger sequencing and next-generation sequencing

Responses to Reviewer #2

The authors have addressed most of the comments on the first version.

However, there are still important issues to assess, prior to consider this manuscript for publication in Nature Communications.

Please find the comments to this second version in the attached document.

Response:

Thank you for your thorough review of the second version of our manuscript. We are encouraged to know that the majority of concerns from the initial review have been addressed to your satisfaction.

To further refine the manuscript, we have made the following key revisions based on your insightful feedback:

- Limitation to Chinese Strains: We have acknowledged in the limitations section that our experimental analysis is limited to analyzed Chinese strains, which may impact the generalizability of our findings.
- Diverse ST Types in Lineage L6: We have provided clarification to explain that the presence of different ST types within lineage L6 is within the normal range, particularly when a large number of strains are analyzed.
- Core Genes in Pangenome Analysis: We have clarified the specific number of core genes identified in the pangenome analysis, highlighting its relevance to the strain dataset used in our study.
- Inactive Resistance Determinants: We have revised the Discussion section to acknowledge that some identified resistance determinants may not be active in *Salmonella* but are still meaningful as potential reservoirs for horizontal transfer.
- Threshold for Resistance Gene Detection: We have justified the use of 80% identity and coverage thresholds for resistance gene detection based on alignment with default Abricate settings, and conducted a supplemental analysis showing minimal impact when increasing the threshold to 95%.

- **Additional Revisions:** We have revised figures and tables for better clarity and precision, defined terms more clearly, and made textual modifications to address specific comments.

For a detailed response to each of your comments, please refer to the point-by-point explanations provided below. We extend our heartfelt gratitude for your thoughtful and constructive feedback, which has significantly enhanced the quality of our work.

The attached document:

The authors have addressed all the comments and in most of cases the answers are satisfactory.

There are however some points that need further consideration.

The present study revises a vast collection of Salmonella Typhimurium isolated in China through recent years, where a high prevalence of the MRDAR phenotype and a high proportion of strains resistant to ≥ 5 families of antibiotics can be highlighted. The authors have observed that in the majority of the cases, the rough phenotype is associated to a particular mutation of the csgD promoter. Compared to S. Typhimurium genomes from international databases, this mutation is proportionally much more frequent in the Chinese bacterial population vs international genomes available online. Thus, the authors hypothesize that the strains from which the genomes did not contain the mutated operon, shall probably be smooth.

However, only Chinese strains were available to compare the presence of the mutated promoter of csgD with the rough/smooth phenotype and the capacity of producing biofilm. Due to this limitation, the authors could not confirm the relationship between the presence/absence of the is mutation with the aspect of the colonies (rough/smooth) in foreign strains.

Also, not all the MRDAR from China carried the mutated operon, and not all the strains with the mutated operon tested expressed an MRDAR phenotype.

Therefore, the authors can argue that from the observations in the strains from China analyzed, an association can be suspected, and laboratory analyses showed that for the tested strains, the phenotype changed when the operon changed. But further investigations, and notably the phenotype information from the foreign strains should be needed to confirm the hypothesis.

Response:

Thank you for your insightful comment. We acknowledge that our analysis is limited to Chinese *Salmonella* Typhimurium isolates, which prevents us from verifying associations between the *csgD* promoter mutation and colony morphology in strains from other geographical areas. This limitation has been clearly stated in the revised manuscript (lines 614-624 in the clean version). We hope that our findings will inspire similar studies worldwide, as collaborative efforts to share and gather phenotypic and genetic data from diverse geographical regions are important for understanding the broader implications of mutations in *Salmonella* Typhimurium. We eagerly anticipate further exploration of this important topic.

For the Chinese strains we analyzed, our laboratory investigation has revealed a strong association between the *csgD* promoter mutation and the rough phenotype. While not all mrdar strains carried the mutated promoter, a substantial majority, 95.25% (623/654), do harbor this mutation. Moreover, all mrdar strains within the L6.5 lineage had the mutated promoter, and the presence of this mutation was remarkably higher in the L6.5 lineage compared to others. Specifically, the mutation rates were as follows: L1 at 1.64% (8/489), L2 at 0.0% (0/530), L3 at 0.72% (5/697), L4 at 1.96% (1/51), L5 at 0.46% (2/430), L6 at 44.58% (782/1754), and L6.5 at 92.50% (777/840).

The provided data allow establishing that the *S. Typhimurium* population in China is quite diverse but remains well differentiated from other populations identified in different regions of the globe. Also, that Chinese *S. Typhimurium* are frequently MRDAR, produce biofilm and very often carry a particular mutation

of the *csgD* promoter, which may be involved in the described phenotype.

The description of a new lineage remains unclear, as the suggested L6 sublineage includes strains from very distant MLST types (ST19, ST34, ST36). Accordingly, it is suggested to change the title to something like: « High frequency of rough morphology and enhanced biofilm producing capacity in *Salmonella enterica* serovar Typhimurium in China and the possible role of the *csgD* operon in these features. »

Response:

We appreciate your valuable suggestion. The discovery of the *csgD* promoter point mutation leading to changes in colony morphology was initially documented by Römling et al. in 1998⁷. However, this observation was originally limited to only a few strains. Our study represents the first identification of a *Salmonella* Typhimurium lineage harboring this mutation, which is present within the wild population and appears to be disseminating across China and other global regions. Therefore, we propose retaining the current title to highlight the importance of our findings.

In addition, regarding your concern about the inclusion of strains from distant MLST types in the L6 sublineage, we have provided a detailed explanation in the response to your specific comment #7. Please refer to that specific response for further details.

Following are more specific comments on the second version of this manuscript.

1. It is suggested to compile in one single table, where each line would correspond to one genome/strain, all the information about that genome/strain.

Response:

We appreciate your suggestion. We have revised Table S1 to consolidate all the information regarding genomes and strains. Please refer to the screenshot below.

Sample	Collection date	Geographic location	Region	Source type	Source	diagnosis	Sex	Age Range	Age	Sequence type	Lineage-A270	Sublineage-A270	S-R phenotype	triaxone, Cr
GZ2016SF032	2016	Southern China	Guangdong	Food	Pork	NA	NA	NA	NA	ST19	L5	L5.2	Smooth	S
GZ2016SF033	2016	Southern China	Guangdong	Food	Pork	NA	NA	NA	NA	ST19	L5	L5.2	Smooth	S
GZ2016SF049	2016	Southern China	Guangdong	Food	Pork	NA	NA	NA	NA	ST34	L6	L6.3	Smooth	S
GZ2016SF052	2016	Southern China	Guangdong	Food	Pork	NA	NA	NA	NA	ST19	L1	L1.2	Smooth	S
GZ2016SF054	2016	Southern China	Guangdong	Food	Pork	NA	NA	NA	NA	ST34	L6	L6.3	Smooth	S
GZ2016SF055	2016	Southern China	Guangdong	Food	Pork	NA	NA	NA	NA	ST34	L6	L6.3	Smooth	S
GZ2016SF061	2016	Southern China	Guangdong	Food	Pork	NA	NA	NA	NA	ST34	L6	L6.3	Smooth	S
GZ2016SF062	2016	Southern China	Guangdong	Food	Pork	NA	NA	NA	NA	ST34	L6	L6.3	Smooth	S
GZ2016SF063	2016	Southern China	Guangdong	Food	Pork	NA	NA	NA	NA	ST19	L1	L1.2	Smooth	S
GZ2016SF064	2016	Southern China	Guangdong	Food	Pork	NA	NA	NA	NA	ST19	L1	L1.2	Smooth	S
GZ2016SF072	2016	Southern China	Guangdong	Food	Pork	NA	NA	NA	NA	ST34	L6	L6.3	Smooth	S
GZ2016SF075	2016	Southern China	Guangdong	Food	Pork	NA	NA	NA	NA	ST34	L6	L6.3	Smooth	R
GZ2016SF089	2016	Southern China	Guangdong	Food	Pork	NA	NA	NA	NA	ST34	L6	L6.5	Rough	S
GZ2016SF114	2016	Southern China	Guangdong	Food	Pork	NA	NA	NA	NA	ST34	L6	L6.5	Rough	S
GZ2016SF117	2016	Southern China	Guangdong	Food	Pork	NA	NA	NA	NA	ST34	L6	L6.4	Smooth	S
GZ2016SF121	2016	Southern China	Guangdong	Food	Pork	NA	NA	NA	NA	ST34	L6	L6.5	Rough	S
GZ2016SF128	2016	Southern China	Guangdong	Food	Pork	NA	NA	NA	NA	ST34	L6	L6.4	Smooth	S
GZ2016SF129	2016	Southern China	Guangdong	Food	Pork	NA	NA	NA	NA	ST34	L6	L6.5	Rough	S
GZ2016SF130	2016	Southern China	Guangdong	Food	Pork	NA	NA	NA	NA	ST34	L6	L6.3	Smooth	S
GZ2016SF131	2016	Southern China	Guangdong	Food	Pork	NA	NA	NA	NA	ST34	L6	L6.4	Smooth	S

INTRODUCTION

2. Line 113-114:

Since the term “extensive MDR” appears here for the first time in the text, it should be here defined.

Response:

Thank you for your suggestion. We have included the definition of “extensive MDR” in the revised sentence, as follows:

Original: These rough colony variants exhibited enhanced biofilm-forming ability and extensive multidrug resistance (MDR), indicating a potentially improved capacity for environmental persistence.

Revised: These rough colony variants exhibited an enhanced biofilm-forming ability and extensive multidrug resistance (MDR), defined as resistance to at least five classes of antimicrobial agents. This indicates a potentially improved capacity for environmental persistence. (lines 110-113 in the clean version of the revised manuscript)

TABLES

3. Sentence in lines 143-144 and Table 1, Figure 1.

The percentages shown in the table and the figure are calculated on the total number of isolates of the same phenotype. But to compare prevalence by years, the percentages should be calculated based on the total number of isolates in the year, and this way, the prevalence of the rough phenotype, has quite decreased

since 2014. This is how figure 1B would look like if the percentages are calculated over the totals of each year:

This same way of expressing prevalence is suggested for Figure 1C.

Response:

We are grateful for your guidance in improving the clarity and precision of our data representation. Following your advice, we have revised Figures 1B and C. Please find the revised figures provided below for your review.

Revised Figures 1B and C

4. Tables with dataset for Figure 1.

Tables with datasets for figures 1A to 1F are identical and so they don't correspond to the 4 different sub-figures. It is suggested to prepare one single table containing all the data shown in Figure 1.

Response:

Thank you for highlighting this issue. We have now resolved it by compiling all the data presented in Figure 1 into a single sheet labeled "Figure 1" within the Source Data file. Additionally, to align with the policy requirements of *Nature Communications*, we have provided separate sub-datasets for Figures 1A to 1F, each on their respective sheets. The screenshot below illustrates this organization.

Sample	S-R phenotype	Region	Collection date	Source type		
GZ2016SF032	Smooth	Guangdong	2016	Food		
GZ2016SF033	Smooth	Guangdong	2016	Food		
GZ2016SF049	Smooth	Guangdong	2016	Food		
GZ2016SF052	Smooth	Guangdong	2016	Food		
GZ2016SF054	Smooth	Guangdong	2016	Food		
GZ2016SF055	Smooth	Guangdong	2016	Food		
GZ2016SF061	Smooth	Guangdong	2016	Food		
GZ2016SF062	Smooth	Guangdong	2016	Food		
GZ2016SF063	Smooth	Guangdong	2016	Food		
GZ2016SF064	Smooth	Guangdong	2016	Food		
GZ2016SF072	Smooth	Guangdong	2016	Food		
GZ2016SF075	Smooth	Guangdong	2016	Food		
GZ2016SF089	Rough	Guangdong	2016	Food		
GZ2016SF114	Rough	Guangdong	2016	Food		
GZ2016SF117	Smooth	Guangdong	2016	Food		
GZ2016SF121	Rough	Guangdong	2016	Food		
GZ2016SF128	Smooth	Guangdong	2016	Food		
GZ2016SF129	Rough	Guangdong	2016	Food		
GZ2016SF130	Smooth	Guangdong	2016	Food		
GZ2016SF131	Smooth	Guangdong	2016	Food		
GZ2016SF133	Smooth	Guangdong	2016	Food		
GZ2016SF142	Smooth	Guangdong	2016	Food		
GZ2016SF148	Smooth	Guangdong	2016	Food		
GZ2016SF150	Rough	Guangdong	2016	Food		
GZ2016SF151	Smooth	Guangdong	2016	Food		
GZ2016SF161	Smooth	Guangdong	2016	Food		
GZ2016SF165	Smooth	Guangdong	2016	Food		
GZ2016SF169	Smooth	Guangdong	2016	Food		

Source data sheets for Figure 1 and its panels (Figures 1A-1F)

5. Tables with dataset for Figure 3.

Also, in that repeated table (3A-3C), there are regions of China and China itself described in the same column, which suggests a possible redundancy in the values shown in the figure.

Response:

Thank you for your comment. The inclusion of panels A, B, and C in Figure 3 is to provide a clearer visualization of data. Figure 3A presents a substantial amount of data and may not effectively present details of sub-lineages within the L6 branch. To address this, Figure 3B is added to provide a more detailed perspective. Figure 3C is a zoomed-in view of the L6.5 branch to showcase detailed information within this sub-lineage. In addition, the corresponding sheets in the Source Data file have been revised to eliminate redundancy and now each sheet corresponds to a single panel in Figure 3. The screenshot below illustrates this organization.

TreeID	Lineage	Sublineage	Status	Geographic location	Sequence type	S-R phenotype	Strain
GZ2016SF032	L5	L5.2	This work	Southern China	ST19	Smooth	/
GZ2016SF033	L5	L5.2	This work	Southern China	ST19	Smooth	/
GZ2016SF049	L6	L6.3	This work	Southern China	ST34	Smooth	/
GZ2016SF052	L1	L1.2	This work	Southern China	ST19	Smooth	/
GZ2016SF054	L6	L6.3	This work	Southern China	ST34	Smooth	/
GZ2016SF055	L6	L6.3	This work	Southern China	ST34	Smooth	/
GZ2016SF061	L6	L6.3	This work	Southern China	ST34	Smooth	/
GZ2016SF062	L6	L6.3	This work	Southern China	ST34	Smooth	/
GZ2016SF063	L1	L1.2	This work	Southern China	ST19	Smooth	/
GZ2016SF064	L1	L1.2	This work	Southern China	ST19	Smooth	/
GZ2016SF072	L6	L6.3	This work	Southern China	ST34	Smooth	/
GZ2016SF075	L6	L6.3	This work	Southern China	ST34	Smooth	/
GZ2016SF089	L6	L6.5	This work	Southern China	ST34	Rough	/
GZ2016SF114	L6	L6.5	This work	Southern China	ST34	Rough	/
GZ2016SF117	L6	L6.4	This work	Southern China	ST34	Smooth	/
GZ2016SF121	L6	L6.5	This work	Southern China	ST34	Rough	/
GZ2016SF128	L6	L6.4	This work	Southern China	ST34	Smooth	/
GZ2016SF129	L6	L6.5	This work	Southern China	ST34	Rough	/
GZ2016SF130	L6	L6.3	This work	Southern China	ST34	Smooth	/
GZ2016SF131	L6	L6.4	This work	Southern China	ST34	Smooth	/
GZ2016SF133	L6	L6.4	This work	Southern China	ST34	Smooth	/
GZ2016SF142	L6	L6.3	This work	Southern China	ST34	Smooth	/
GZ2016SF148	L6	L6.3	This work	Southern China	ST34	Smooth	/
GZ2016SF150	L6	L6.5	This work	Southern China	ST34	Rough	/
GZ2016SF151	L5	L5.2	This work	Southern China	ST19	Smooth	/
GZ2016SF161	L6	L6.3	This work	Southern China	ST34	Smooth	/
GZ2016SF165	L5	L5.2	This work	Southern China	ST19	Smooth	/
GZ2016SF169	L6	L6.3	This work	Southern China	ST34	Smooth	/

< > | **Figure 3A** Figure 3B Figure 3C | Figure 4A Figure 4B Figure 4C Figure 5A Figur ... + ||

Source data sheets for Figures 3A-3C

RESULTS

6. Lines 166-168:

This reviewer pointed out that the rough phenotype in Salmonella is often shown

after subcultures, in comment #2. The authors answered that the rough phenotype in their strains existed already at the initial isolation.

However, the authors describe an increase in this phenotype in lines 166-168 of the new version of the manuscript. This is in agreement to this reviewer's comment, that the roughness of Salmonella has a tendency to appear (or increase) over generations reproduced in vitro.

Response:

The statements in lines 166-168 of the second version manuscript highlight the increased colony size and more distinct differences of rough colony strains over extended cultivation periods, compared to the control ATCC 14028 strain and the smooth isolates. It is important to clarify that, in our experimental observations, the phenotype of rough colony strains was clearly evident at the initial isolation and remained stable through multiple passages. For a visual demonstration of this stability, please refer to the figure below.

To improve clarity, we have revised the sentence you mentioned, as follows:

Original: Furthermore, over time, the mrdar colony variants developed increasingly larger colonies and more obvious wrinkles than the control ATCC 14028 strain and the smooth isolates (Figure 2A and B).

Revised: Furthermore, over extended cultivation periods, the mrdar colony variants developed increasingly larger colonies with more pronounced wrinkles than the control ATCC 14028 strain and the smooth isolates (Figure 2A and B). (lines 165-168 in the clean version of the revised manuscript)

DISCUSSION

7. Lines 420 to 422:

Agreed with the first sentence: « The major finding of this study is the identification of a macro-rdar colony morphology of *S. Typhimurium* that emerged and became prevalent in China. »

Not agreed with the second sentence: « We provided a comprehensive description of the genomic and proteomic characteristics of these mrdar variants and observed that they formed a novel lineage on the phylogenetic tree. »

There are some strains OUTSIDE the lineage with these characteristics and strains INSIDE the lineage without these properties. There are strains of different ST (ST34, ST36, which are at 4/7 AD on 7g-MLST and > 400 AD on cgMLST) in the lineage L6, even in L6.5. It is striking that genomes of so different MLST types can constitute a lineage, while other genomes from those ST remain so distant in the tree, that are considered as different lineages (also with mixed STs). How do the authors explain the structure of the tree? What are the genomic traits of the Chinese genomes when compared to those obtained from online databases?

Following the authors' responses to the comments in the first review, it is clearer that what is here described are phenotypic — mrdar — variants; but not a lineage.

Response:

Thank you for your valuable feedback. We have revised the second sentence you mentioned, as follows:

Original: We provided a comprehensive description of the genomic and proteomic characteristics of these mrdar variants and observed that they formed a novel lineage on the phylogenetic tree.

Revised: We provided a comprehensive description of the genomic and proteomic characteristics of these mrdar variants and observed that most of them clustered in a novel sublineage on the phylogenetic tree. (lines 424-427 in the clean version of the revised manuscript)

We would like to clarify that MLST typing and phylogenetic tree analysis are two different methods. MLST sequence typing is an early typing method based on

housekeeping gene loci, and may not accurately represent the evolutionary relationships between strains. On the other hand, phylogenetic tree analysis is based on SNP differences and provides a better representation of the evolutionary relationships between strains⁹. Although there are some correlations between the results of MLST typing and phylogenetic analysis¹⁰, phylogenetic tree analysis has advantages over molecular typing methods like MLST in providing phylogenetic information¹¹.

When a mutation occurs at a housekeeping gene locus, the ST typing result for a strain may change. For example, a strain with a *dnaN* gene mutation at position 496, changing from A to G, would shift its typing result from ST34 to ST19. Nonetheless, in phylogenetic tree analyses, the strain would still cluster with strains that are most closely related in terms of SNP distances. In phylogenetic tree analyses, it is common for a lineage to primarily consist of strains of a particular ST type, while also including a small proportion of strains of other ST types, especially when a substantial number of strains are analyzed¹²⁻¹⁵. In our study, the sublineage L6.5 comprised 840 strains, with 834 (99.26%) being ST34, 4 (0.48%) being ST19, 1 (0.12%) being ST36, and 1 (0.12%) being ST6020, which can be regarded as within the normal range.

Furthermore, when constructing the phylogenetic tree, *S. Heidelberg* (accession number NC_011083.1)¹⁶ was also used as the outgroup strain to root the tree. We found that the tree based on *S. Heidelberg* showed the same topology as the tree based on *S. Paratyphi* A270 (accession number ERR326600). Please refer to the figure below for a visual representation. We selected the tree constructed with *S. Paratyphi* A270 as the reference strain, and then used the hierBAPS software [DOI: 10.1093/molbev/mst028] for lineage assignment.

Phylogenetic tree with *S. Heidelberg* as the outgroup reference strain

8. Monophasic \neq ST34 \neq lack of fljA/fljB

As the authors know, these three concepts are not synonymous: not all monophasic strains belong to ST34, nor do they owe their phenotype to the lack of fljA/fljB; not all strains of ST34 are monophasic, nor do they lack fljA/fljB; not all strains lacking fljA/fljB belong to ST34. The referred data in figure

Response:

We agree with your observation that monophasic strains are mostly associated with ST34, but there are also other types such as ST19. As highlighted in existing literature, it is evident that monophasic strains are not equivalent to ST34, nor does they automatically indicate the absence of fljA/fljB^{17,18}. In our study, we used the *Salmonella*

serotyping prediction software, SeqSero2, to identify monophasic strains, a method that has been widely used in previous studies¹⁹.

9. Biofilm formation imaging

This reviewer asked the authors to show the confocal imaging, or the EPS and fibrillar network results (like figures 2 and S1) of all the other 19 strains tested of Rough and Smooth phenotype, as a variability in biofilm formation was expected. The authors did not reproduce the imaging as suggested, but they performed crystal violet and pellicle formation tests. The new figure S2 (B and C) is not very clear, but allows to observe the expected variability, as rough strains R12 and R20 show similar degrees of biofilm and pellicle formation as smooth strains S2, S8 and S18. This is why the confocal imaging in all of them would have been very important.

The authors have performed the assays also in strains from other lineages than L6. However, their results are only described in a table, and not shown in an image.

Response:

Thank you very much for your valuable recommendations. In response, we have now included confocal imaging results for the 40 strains (20 rough and 20 smooth) that were the focus of our analysis after 24 hours of cultivation at 28 °C in the revised supplementary appendix. These results are presented in the new Figure S3, which you can find below.

Figure S3. Biofilm production by representative rough and smooth colony strains determined by confocal laser scanning microscopy (CLSM) at 28 °C.

We have also revised Figure S2. Considering that panels B and C in the original Figure S2 represented only differences before and after staining, we have removed panel B and enlarged panel C to better display the results. The revised Figure S2 is as follows:

Figure S2. Examination of the red, dry, and rough morphotype and pellicle formation in representative rough and smooth colony strains.

Furthermore, as per your suggestion, we have created a bar graph to illustrate the biofilm formation results for strains from lineages other than L6. The graph is presented below.

Biofilm formation by 22 strains from lineages other than L6

10. Line 192:

Please avoid using terms as “important” when describing results. Such valorization expressions should be left to the Discussion, and only if well justified.

Response:

Thank you for your suggestion. We have removed the term “important.” The sentence now reads, “The majority of these isolates exhibited MDR, with 79.5% demonstrating resistance to three or more Clinical and Laboratory Standards Institute (CLSI) classes, including MDR phenotypes such as ASSuT (45.5%, resistance to at least ampicillin, streptomycin, sulfisoxazole, and tetracycline) and ACSSuT (22.0%, resistance to at least ampicillin, chloramphenicol, streptomycin, sulfisoxazole, and tetracycline) (Table 2).” (lines 189-194 in the clean version of the revised manuscript)

11. Line 196:

Like in the previous comment, avoid using valuating expressions in the Results. Because the authors say that « [...] the mrdar variants had “more severe” AMR profiles than the smooth colony isolates, but if an evaluation of clinical severity must be expressed on these results, it must be noted that none of the antibiotics referred in this paragraph (tetracycline, ampicillin, sulfisoxazole, nalidixic acid, chloramphenicol, trimethoprim/sulfamethoxazole or gentamycin) is of choice to treat Salmonella. Cefoxitin, ciprofloxacin and azithromycin are the main therapeutic options to treat Salmonella infections, and against those three antibiotics, the smooth colonies were more resistant than the rough ones.

Response:

We sincerely appreciate your valuable suggestion. In response, we have revised the sentence, as follows:

Original: Notably, the mrdar variants had more severe AMR profiles than the smooth colony isolates, ...

Revised: Notably, the mrdar variants exhibited higher rates of resistance to multiple antibiotics than the smooth colony isolates, ... (lines 194-196 in the clean version of the revised manuscript)

We tested for the susceptibility of strains to a panel of 14 and 16 antimicrobials by broth

microdilution using the National Antimicrobial Resistance Monitoring System (NARMS) Gram Negative CMV3AGNF plate. Therefore, the antibiotics selected for testing in our study are under the surveillance of the NARMS program. Additionally, the plate we used for antimicrobial susceptibility testing is widely used in research on *Salmonella* antimicrobial resistance²⁰⁻²³. Although antibiotics such as tetracycline and chloramphenicol are not the first-line treatment options for clinical *Salmonella* infections currently, they remain extensively used in animal husbandry.

12. Population structure of global *S. Typhimurium*, including Chinese isolates

The authors did not detail the number of genes included in the core genome on which the SNPs were identified. Taking in account that the number of genomes from China(3,851, 57.8% of the total of 3,851 shown in table 450756_1_data_set_8614497_s83rx8)largely outnumber the genomes from other countries, the core genome could be disregarding important genes that could have provided a higher variability to the phylogenetic analysis.

Response:

Thank you for raising these important points. In our study, we utilized the widely-used Snippy pipeline^{24,25} to identify core SNPs for phylogenetic tree construction. The Snippy output provided the number of differential core SNPs, but did not specify the total number of genes in the core genome. For detailed information on the capabilities and methodology of the Snippy pipeline, please refer to its documentation at <https://github.com/tseemann/snippy>.

We performed phenotypic identification for the Chinese isolates in our study, with particular emphasis on the emergence of a large number of strains exhibiting the mrdar phenotype. The phylogenetic analysis showed that mrdar strains clustered mainly within the L6.5 lineage. As phenotypic identification could not be performed for isolates from other countries, Chinese isolates constituted a large proportion of our dataset. We acknowledge the possibility that genes potentially providing greater

variability to the phylogenetic analysis may have been overlooked in the core genome. We hope our work will inspire further studies integrating both phenotypic and genomic data from diverse geographic regions, enabling the construction of a more comprehensive phylogenetic tree.

13. Line 250:

« L6 lineage [...] has been particularly successful at global dissemination, comprising 44,4% of all isolates in this study. »

Since the authors explain that this L6 comprises the latest genomes, this “success” in dissemination must take in account that sequencing all over the world has increased dramatically in recent years, so it is to be expected that most genomes will be recent.

This “success” is a value added to the result that should be left to the Discussion.

Response:

We appreciate your insightful feedback. We agree that the description of the L6 lineage’s “success” in dissemination requires careful consideration, particularly given the limited historical strain sequencing data included in our study. To improve the accuracy of our statement, we have removed the phrase “particularly successful at global dissemination” from the sentence, as follows:

Original: The L6 lineage, a recently derived clade that developed circa 1884 (95% CI: 1874-1894), has been particularly successful at global dissemination, comprising 44.4% of all isolates in this study.

Revised: The L6 lineage, a recently derived clade that developed circa 1884 (95% CI: 1874-1894), comprised 44.4% of all isolates in this study. (lines 249-250 in the clean version of the revised manuscript)

14. Line 258:

« Furthermore, the mrdar variants “seemingly” also presented in other regions of the world [...]».

The authors should avoid hypothesizing in the Results. Also, there are no phenotypic information on the morphology of strains abroad since the study only included phenotypic analyses on Chinese strains.

Response:

Thank you for your suggestion. We have revised the description of results, as follows:

Original: From the regional distribution data, the mrdar variants were observed to be widely distributed in diverse regions of China, indicating that rough colony *S. Typhimurium* spread more frequently in China. Furthermore, the mrdar variants seemingly also presented in other regions of the world, as the isolates from Europe (23 isolates), North America (11 isolates), Oceania (four isolates), and South America (one isolate) were clustered in this sublineage (Figure 3C, Table S1).

Revised: Based on the regional distribution data, the sublineage L6.5 was primarily composed of strains from China, including strains from various regions within the country. Additionally, this sublineage included strains from other parts of the world, such as Europe (23 isolates), North America (11 isolates), Oceania (four isolates), and South America (one isolate) (Figure 3C, Table S1). (lines 254-259 in the clean version of the revised manuscript)

15. Lines 263-264:

The sentence should be kept for the Discussion. Those are not results of the present study.

The authors should describe what “AMR determinants” are. Genes? Mutations (including different mutations in the same gene?).

Response:

Thank you for your valuable suggestion. As per your recommendation, we have relocated the sentence “The acquisition of AMR determinants may act as a major driving force for the adaptive evolution of MDR *S. Typhimurium* lineages” to the Discussion section (lines 586-587 in the clean version of the revised manuscript).

Additionally, we have provided a description of AMR determinants, as follows: “We examined the presence of genetic determinants of AMR, including the acquired resistance gene content and the presence of QRDR mutations²⁶.” (lines 262-263 in the clean version of the revised manuscript)

16. Line 264-266:

Table S1 does not contain the information expressed in the referring line.

Figure S6A and parenthesis in line 266: according to the authors, all strains carried AMR determinants, which is strange, given that the wildtype phenotype of *S. Typhimurium* is pan-susceptible. Also, the authors describe that there is at least one strain in L1 with > 40 resistance AMR determinants and at least one strain in L6 with > 50. Those data require further . The authors may have considered as different determinants the different denominations, or the different <100% hits to BLAST hits on the same genomic region. Those numbers of AMR determinants are unprecedented.

The number of genes of resistance by lineage depends greatly on the number of genomes in the lineage, which renders figure S6A very difficult to evaluate.

Response:

Thank you for your insightful comments. We appreciate the opportunity to address the points you have raised and provide further clarification. Firstly, we can confirm that Table S1 contains the information referred to in the corresponding line, specifically the identification of 152 resistance genes and the number of resistance genes identified in each strain.

Secondly, it is important to highlight that with the widespread use of antibiotics, the dissemination of resistance genes, and the ongoing genetic evolution of *S. Typhimurium*, a large portion of *Salmonella Typhimurium* strains are not pan-susceptible to antibiotics. For example, a study conducted in Bangladesh found that 94% of *Salmonella* strains isolated from broiler chickens exhibited multidrug resistance²⁷. Similarly, in Taiwan

between 2017 and 2018, 47.3% of non-typhoidal *Salmonella* (NTS) strains were reported to be multidrug-resistant²⁸.

Lastly, Figure S6A illustrates the number of resistance gene types carried by strains across different lineages. Each lineage possesses distinct genomic characteristics, while strains within the same lineage share similar genomic features. This can lead to the clustering of resistance genes, virulence factors, or plasmid replicons within a particular lineage. The number of resistance genes is closely associated with the genomic characteristics of a lineage, as evidenced in Figure S7 (provided below). However, as you pointed out, the number of resistance genes is also influenced by the number of genomes within each lineage. Consequently, the observed differences in the number of resistance gene types among lineages may be partly attributed to variations in the number of genomes within each lineage. We acknowledge that this potential confounding factor should be taken into consideration when interpreting the data presented in Figure S6A.

Figure S7. Maximum likelihood phylogeny of the global *S. Typhimurium* isolates with a heatmap showing the distribution of the antimicrobial-resistant determinants.

17. Line 276:

When referring to disinfectants, it is strongly suggested not to use terms of “resistance”, but of “decreased susceptibility”, as there are not cut-off values, yet.

Response:

Thank you for your valuable suggestion. We have revised the relevant expression according to your advice.

Original: two efflux pump-mediated disinfectant resistance genes (*qacEdelta1* and *qacL*)

Revised: two efflux pump-mediated genes with decreased susceptibility to disinfectants (*qacEdelta1* and *qacL*) (lines 274-275 in the clean version of the revised manuscript)

18. Lines290-293:

It should be noted that strains from the sublineage L1.3, where strains were mostly smooth, not only carried a mutation on *gyrA* (D87N) but also a second mutation, on *parC* (S80R). Actually, while single QRDR mutations (associated to decreased susceptibility to ciprofloxacin) were observed in more rough strains (n = 604; 92.4% of all rough) than smooth strains (n = 190; 12.2% of all smooth strains), double or triple or triple QRDR mutations, associated to high level of resistance to ciprofloxacin, were ten times more frequent among smooth (n = 91; 5.84% of all smooth) than in rough (n = 9; 1.37% of all rough) isolates, according to table 450756_1_data_set_8614497_s83rx8.

Thus, a higher level of resistance to ciprofloxacin could be expected among those smooth-colony strains than among the L6.5, presenting with only one *gyrA* mutation. But globally, a higher proportion of rough strains than smooth strains should have shown lower levels of susceptibility to ciprofloxacin (due to the one mutation in *gyrA*), as 1,227 smooth strains (78% of all smooth strains) did not present any QRDR mutations.

Response:

We appreciate your insights and have accordingly revised our manuscript to reflect these observations more accurately.

Original: The QRDR mutations were mainly observed in Chinese organisms, and the sublineage L6.5 *mrdr* variants mostly harbored the D87N or D87Y mutation in the *gyrA* gene (Figure S7). However, we did not find differences in ciprofloxacin resistance between *mrdr* variants and smooth strains, possibly because some smooth strains also harbored the D87N or D87Y mutation in the *gyrA* gene, along with others that harbored the S83L mutation in the *gyrA* gene.

Revised: The QRDR mutations were mainly observed in Chinese strains, particularly within sublineages L6.5 and L1.3. Within sublineage L6.5, the *mrdr* variants mostly harbored the D87N or D87Y mutation in the *gyrA* gene, while sublineage L1.3 strains, characterized by their smooth morphology, carried mutations not only in the *gyrA* gene (S83L and/or D87N) but also in the *parC* gene (S80R) (Figure S7). The presence of double or triple mutations in the QRDR is correlated with a high level of resistance to ciprofloxacin, and single mutations are associated with decreased sensitivity to this antibiotic²⁹. Consequently, compared to smooth colony strains, the *mrdr* variants exhibited a lower resistance rate (3.98% vs. 7.18%) and a higher intermediate rate (10.70% vs. 2.50%) to ciprofloxacin. (lines 287-296 in the clean version of the revised manuscript)

19. Identification of AMR- and virulence-associated plasmids

It would be important to show the Inc of plasmids associated with the different AMR determinants appearing in table 450756_1_data_set_8614497_s83rx8.

In that table, what are the figures in the AMR determinants? Percentage of identity? Percentage of coverage? For some genetic determinants the figures shown are < 90%. Although there is not a standard threshold, experience shows that the predicted associated phenotype rarely appears when the identified gene is lower than 90%, sometimes even if < 95%.

Response:

Thank you for your valuable comments. The information on plasmid replicons and the coverage for each resistance gene has been included in Table S1. Please refer to the screenshot below.

A screenshot for Table S1

In this study, we used ABRicate [<https://github.com/tseeman/abricate/>] and the CARD database³⁰ to identify resistance genes, which is a commonly adopted method. There is currently no consensus in the scientific community on the optimal identity and coverage thresholds. Many studies adhered to the default minimum identity and coverage thresholds of 80% in the ABRicate scheme^{31,32}. Alternatively, some other studies chose a minimum identity of 80% and a minimum coverage of 50%³³. In our study, we set the minimum identity and coverage thresholds at 80% to align with the default settings of the Abricate scheme, which is considered reasonable. Upon reviewing the data, we found that increasing the coverage threshold to 95% only impacted two resistance genes: *aadA1-ANT(3'')-IIa* (85% coverage) and *marA* (91% coverage).

In addition to that, some determinants described in this manuscript have no activity in Salmonella. For example, the authors include among the mcr genes (associated to resistance to colistin) the gene mcr9, which is known NOT to cause this resistance in Salmonella. And they identified mefB in three strains, of which two showed to be susceptible to azithromycin, and the one being resistant carried

also the gene *mphA* that is well known to cause this phenotype in *Salmonella*. And they.

These considerations should be highlighted. Otherwise, the “take-home” message would be the high number of AMR determinants observed, when in reality the AMR profiles were much less striking.

Response:

Thank you for your insightful comments. We recognize that in vitro resistance is largely explained by the presence of known genetic determinants of AMR³⁴, and we agree that some of the resistance determinants in our study do not exhibit activity in *Salmonella*. Nonetheless, we think that documenting the presence of these resistance genes remains meaningful. The majority of these resistance determinants are harbored by mobile genetic elements, such as plasmids, which turns these strains into potential reservoirs of resistance genes and promotes the spread of antibiotic resistance among different bacterial hosts through horizontal transfer³⁵.

To address your concerns, we have included additional descriptions in the Discussion section, as follows: “Although certain resistance genes, such as *mcr-9*, do not necessarily confer polymyxin-resistant phenotypes in *Salmonella*, the majority of these genes are located on plasmids and transposable elements³⁶. This facilitates the transfer of these genes to distantly related bacteria through mechanisms such as conjugation, transduction, or transformation, ultimately leading to the dissemination of antibiotic resistance^{35,37}.” (lines 598-602 in the clean version of the revised manuscript)

20. Pangenome analysis

It is surprising that the core genome of all the *S. Typhimurium* in the study was only 1,821 genes (roughly 3274 if adding the soft-core genes), while the cgMLST for the entire genus *Salmonella* is of 3,002 genes

(<http://www.genome.org/cgi/doi/10.1101/gr.251678.119>,

<https://doi.org/10.3389/fmicb.2017.01345>)

Response:

Thank you for bringing this to our attention. The difference in the number of core genes between the pangenome analysis and cgMLST can be attributed to the distinct methodologies employed by each analysis. In our study, we used the prokka software [DOI: 10.1093/bioinformatics/btu153] for genome annotation and subsequently applied Roary for pangenome analysis. Roary categorizes genes as core if they are present in 99% or more of strains, and as soft-core if they are present in 95% to 99% of strains³⁸. Therefore, the number of core genes in the pangenome analysis is specific to our strain dataset. In contrast, cgMLST identifies core genes based on a background database containing a total of 120,471 strains³⁹.

21. Lines 364 and 642/Table S5:

Table S5 doesn't explain which were there 20 S + 20 R "selected strains". And nowhere in the text appear the criteria for this selection.

Response:

Thank you for your inquiry. To address your concern, we have annotated the 20 rough strains and 20 smooth strains included in Table S5 with circular symbols in revised Figure 5B, as shown below. Furthermore, we have added the criteria for strain selection in the "Biofilm formation" section of the Methods, as follows: "These 40 representative strains were randomly selected from the L6 strains based on the results of phylogenetic analysis." (lines 665-666 in the clean version of the revised manuscript)

B

Revised Figure 5B

22. Line 422:

« [...] and observed that they formed a novel lineage on the phylogenetic tree. »

The *mrda*r colony-forming strains may constitute a variant of *S. Typhimurium* and its monophasic variant. But they do not constitute an evolutionary lineage. It is very surprising that such a restrict sublineage like L6.5 may include genomes of ST19, ST34 and ST36.

Response:

Thank you for your insightful comment. We would like to clarify that MLST typing and phylogenetic tree analysis are two different methods. MLST sequence typing is an early typing method based on housekeeping gene loci, and may not accurately represent the

evolutionary relationships between strains. On the other hand, phylogenetic tree analysis is based on SNP differences and provides a better representation of the evolutionary relationships between strains⁹. Although there are some correlations between the results of MLST typing and phylogenetic analysis¹⁰, phylogenetic tree analysis has advantages over molecular typing methods like MLST in providing phylogenetic information¹¹.

When a mutation occurs at a housekeeping gene locus, the ST typing result for a strain may change. For example, a strain with a *dnaN* gene mutation at position 496, changing from A to G, would shift its typing result from ST34 to ST19. Nonetheless, in phylogenetic tree analyses, the strain would still cluster with strains that are most closely related in terms of SNP distances. In phylogenetic tree analyses, it is common for a lineage to primarily consist of strains of a particular ST type, while also including a small proportion of strains of other ST types, especially when a substantial number of strains are analyzed¹²⁻¹⁵. In our study, the sublineage L6.5 comprised 840 strains, with 834 (99.26%) being ST34, 4 (0.48%) being ST19, 1 (0.12%) being ST36, and 1 (0.12%) being ST6020, which can be regarded as within the normal range.

23. Line 450:

« The majority of the *mrda*r variants were found in sublineage L6.5. ». To be noted that L6.5 contains, in general, the highest number of strains (n = 708, rough an smooth together) from this study, while there lineages with 0, 1 or 2 isolates with the morphotype tested.

In fact, since only the phenotype of the strains from this study have been studied, the discussion should be focused on the presence/absence of the mutations that the authors associate with the described phenotypes: not about the evolution of “rough/*mrda*r” but about the evolution of “-44G>T substitution in the *csgD* gene promoter”.

Response:

Thank you for your insightful comments. We greatly appreciate the opportunity to address your concerns and provide further clarification.

The emergence of a new lineage of *S. Typhimurium* is indeed a complex process. For example, the initial belief that the absence of *fljA/fljB* was the primary reason for the emergence of monophasic variants has been challenged by recent findings¹⁷. In our investigation of the genomes of rough/mrdar *S. Typhimurium*, we found that the vast majority (95.25%, 623/654) of strains exhibiting this phenotype harbored the -44G>T mutation. Moreover, all mrdar strains within the L6.5 lineage had the mutated operon, and the presence of this mutation was remarkably higher in the L6.5 lineage compared to others. Specifically, the mutation rates were as follows: L1 at 1.64% (8/489), L2 at 0.0% (0/530), L3 at 0.72% (5/697), L4 at 1.96% (1/51), L5 at 0.46% (2/430), L6 at 44.58% (782/1754), and L6.5 at 92.50% (777/840). The phenotypic alteration was confirmed in laboratory-constructed mutant strains. Therefore, we hypothesize that this mutation is highly likely to be the cause of the observed rough phenotype. However, as you pointed out, there were a few rough/mrdar strains lacking this mutation in their genomes. The reasons and mechanisms behind this phenomenon are intriguing questions that we aim to address in our future work.

In response to your comments, we have revised relevant statements in the manuscript. For example, in the discussion section, we have revised the following sentence:

Original: The ancestor of the mrdar variants was traced back to circa 1977 (95% CI: 1966-1987), showing that the emergence and clonal expansion of mrdar variants occurred in recent evolutionary history.

Revised: The ancestor of the L6.5 strains was traced back to circa 1977 (95% CI: 1966-1987), showing that the clonal expansion of mrdar variants occurred in recent evolutionary history. (lines 477-479 in the clean version of the revised manuscript)

24. Line 608:

The authors should define “severe MDR” profiles. Also, while it is true that for

instance, the proportion of ESBL or AmpC among the rough colonies was higher than among the smooth strains, the difference was not that striking: 17,7% of rough showed resistance to 3dr generation cephalosporins, while 14,7% of smooth colonies did. Is this a significant difference?

There were no carbapenem-resistant strains identified in the study, and only 10 were resistant to the three molecules used for treating infections by *Salmonella* , of which 7 were smooth and three were rough.

Response:

Thank you for your insightful comments. We would like to clarify that we did observe a notably higher proportion of resistance to several older antibiotics among the mrdar strains, particularly nalidixic acid, gentamicin, and chloramphenicol. Although these antibiotics are not typically used as first-line treatments for *Salmonella* infections in clinical settings, they are still widely used in the livestock industry. Therefore, resistance patterns to these antibiotics are relevant to our understanding of the epidemiology and potential environmental adaptation of these strains. The detailed comparison results of antibiotic resistance profiles between rough and smooth strains, along with the corresponding statistical analysis (P-values), can be found in Table 2 (provided below).

Table 2. Antimicrobial resistance of the Chinese *S. Typhimurium* strains in this study.

Antimicrobial	Antimicrobial resistance (%)			P value*
	Total (n = 2,212)	Rough (n = 654)	Smooth (n = 1,558)	
Ceftriaxone	15.6	17.7	14.7	0.087
Tetracycline	76.7	90.5	71.1	< 0.001
Ceftiofur	16.0	17.7	15.2	0.163
Cefoxitin	3.7	2.4	4.2	0.063
Gentamicin	28.9	57.3	17.0	< 0.001
Ampicillin	76.2	89.4	70.7	< 0.001
Chloramphenicol	46.8	71.6	36.4	< 0.001
Ciprofloxacin	6.2	4.0	7.2	0.006
Trimethoprim/sulfamethoxazole	39.3	58.3	31.3	< 0.001
Sulfisoxazole	75.9	93.7	68.5	< 0.001

Nalidixic acid	52.3	95.3	34.3	< 0.001
Streptomycin	54.2	56.0	53.4	0.272
Azithromycin	9.7	9.2	9.9	0.651
Amoxicillin/clavulanic	2.9	3.2	2.8	0.730
≥ 3 CLSI classes	79.5	95.4	72.8	< 0.001
≥ 4 CLSI classes	73.6	92.5	65.7	< 0.001
≥ 5 CLSI classes	45.2	83.9	28.9	< 0.001
≥ 6 CLSI classes	27.6	58.1	14.8	< 0.001
ASSuT	45.5	49.7	43.8	0.011
ACSSuT	22.0	34.6	16.8	< 0.001
ACSSuTCRO	5.8	6.1	5.6	0.935
ACSSuTCIP	4.1	2.0	5.0	0.002
ACSSuTAZI	3.7	4.1	3.5	0.533

* $P < 0.05$ denotes that there is a clear significant difference in resistance to the corresponding antimicrobials. CLSI, Clinical and Laboratory Standard Institute; ASSuT, resistance to ampicillin, streptomycin, sulfamethoxazole/sulfisoxazole, and tetracycline; ACSSuT, resistance to ampicillin, chloramphenicol, streptomycin, sulfamethoxazole/sulfisoxazole, and tetracycline; CRO, ceftriaxone; CIP, ciprofloxacin; AZI, azithromycin.

Considering that the difference in resistance to the third-generation cephalosporins between rough and smooth strains was not statistically significant, we agree that the wording “severe MDR” may not accurately represent our findings. To avoid confusion and ensure clarity, we have revised the relevant statements in the manuscript, as follows:

Original: This variant exhibited a potentially enhanced capacity for environmental adaptation, including severe MDR and enhanced biofilm-forming ability.

Revised: This variant exhibited a potentially enhanced capacity for environmental adaptation, including a high proportion of MDR and enhanced biofilm-forming ability.

(lines 627-629 in the clean version of the revised manuscript)

25. Line 609:

The definition of a new lineage, where ST34 and ST19 or ST36 coexist is difficult to understand.

Response:

Thank you for your valuable feedback. We would like to clarify that MLST typing and

phylogenetic tree analysis are two different methods. MLST sequence typing is an early typing method based on housekeeping gene loci, and may not accurately represent the evolutionary relationships between strains. On the other hand, phylogenetic tree analysis is based on SNP differences and provides a better representation of the evolutionary relationships between strains⁹. Although there are some correlations between the results of MLST typing and phylogenetic analysis¹⁰, phylogenetic tree analysis has advantages over molecular typing methods like MLST in providing phylogenetic information¹¹.

When a mutation occurs at a housekeeping gene locus, the ST typing result for a strain may change. For example, a strain with a *dnaN* gene mutation at position 496, changing from A to G, would shift its typing result from ST34 to ST19. Nonetheless, in phylogenetic tree analyses, the strain would still cluster with strains that are most closely related in terms of SNP distances. In phylogenetic tree analyses, it is common for a lineage to primarily consist of strains of a particular ST type, while also including a small proportion of strains of other ST types, especially when a substantial number of strains are analyzed¹²⁻¹⁵. In our study, the sublineage L6.5 comprised 840 strains, with 834 (99.26%) being ST34, 4 (0.48%) being ST19, 1 (0.12%) being ST36, and 1 (0.12%) being ST6020, which can be regarded as within the normal range.

METHODS

26.AMR gene and VF detection.

CARD is a very exhaustive but not curated database, and the use of its results must be done with care. There are plenty of genes with proven absence of activity, in all bacteria or at least in Salmonella.

Also, the threshold of 80% of coverage and identity is too low. Many genes have similar structures, and this threshold can bring to false identifications. Standard protocols suggest not to go under 95%.

Response:

Thank you for your valuable comment. In this study, we used ABRicate [<https://github.com/tseeman/abricate/>] and the CARD database³⁰ to identify resistance genes, which is a commonly adopted method. There is currently no consensus in the scientific community on the optimal identity and coverage thresholds. Many studies adhered to the default minimum identity and coverage thresholds of 80% in the ABRicate scheme^{31,32}. Alternatively, some other studies chose a minimum identity of 80% and a minimum coverage of 50%³³. In our study, we set the minimum identity and coverage thresholds at 80% to align with the default settings of the ABRicate scheme, which is considered reasonable. Upon reviewing the data, we found that increasing the coverage threshold to 95% only impacted two resistance genes: *aadA1-ANT(3'')-IIa* (85% coverage) and *marA* (91% coverage).

Responses to Reviewer #3

The authors have addressed many of my concerns yet there are several outstanding issues associated with this study including:

CRITICAL CORRECTION REQUIRED:

Please update the proteomic methods to accurately reflect the data associated with iProX ID associated with IPX0007671000. Examination of the data uploaded reveals this data was collected on a Bruker instrument not a Thermo HF-X with the HyStar report (see attached) also highlighting the LC is a Evosep not the reported Easy-nLC 1200. Either the incorrect dataset has been uploaded or description of the method is grossly inaccurate.

Response:

We sincerely apologize for the substantial oversight in our documentation of proteomic methods. Upon receiving your feedback, we conducted a thorough review of our analysis procedures, during which we confirmed the accuracy of the raw data files, search databases, and the final results. However, the description of methods was indeed incorrect.

To address this issue, we have carefully revised the relevant methods description in the supplementary appendix and made corresponding revisions to the information uploaded to iProX. For your convenience, we have included the corrected information below. We are fully committed to ensuring the highest level of accuracy and integrity in our research. We are grateful for your patience and the opportunity to correct this mistake.

Project ID: IPX0007671000

ProteomeXchange ID: PXD047483

<https://www.iprox.cn/page/PSV023.html?url=1709700388071SfYu>

Code: fG6t

Revised version of “Proteomic analysis of the rough and smooth colony strains” in the supplementary appendix:

Sample collection and preparation

Forty strains of rough and smooth *Salmonella* typhimurium were inoculated onto the multiple soft agar medium using the inoculation loop method for preliminary identification of *Salmonella*. After incubating them at 37 °C for 18 hours, the colonies were gently scraped off with an inoculation ring, avoiding the culture medium components. The scraped colonies were transferred into 2 mL EP tubes containing 1.5 mL PBS. To obtain enough samples, this process was repeated until the liquid inside the EP tubes became visibly turbid. The EP tubes were labeled with the strain numbers on the walls or caps and centrifuged at 5000 rpm for 9 minutes. The bacterial samples were collected as sediments within the EP tubes.

The protein extraction process was conducted as follows⁴⁰. Initially, three steel balls, each with a diameter of 3 mm, were mixed with an appropriate amount of quartz sand in a 2 ml thickened centrifuge tube, which was then placed on ice for later use. Thallus samples were collected and stored temporarily in a 2 ml screw-cap centrifuge tube at -80 °C, with the remaining samples also stored at -80 °C. Following this, 600 µL of lysate [SDT buffer (4% Sodium dodecyl sulfate, 100 mM Tris-HCl, 1 mM Dithiothreitol, pH 8.0)] was added, and the cap of the EP tube was tightly secured. The sample then underwent a thorough freeze-grinding process. Ultrasonication was employed in intervals of 30 seconds, alternated with 30-second breaks, accumulating to a total duration of 40 minutes. Afterward, the sample was heated in a boiling water bath at 95 °C for 8 minutes. The sample was then centrifuged at 20 °C and 14,000 g for 30 minutes, after which the supernatant was removed and transferred to a new centrifuge tube.

Dithiothreitol (final concentration 10 mM) was added to each 100 µg of the above sample and mixed at 600 rpm for 1.5 hours at 37 °C. After cooling the samples to room

temperature, Iodoacetamide (final concentration 20 mM) was added to block the reduced cysteine residues and incubated the samples for 30 minutes in the dark. The samples were transferred to filters (Microcon units, 10 kDa) and washed with 100 μ l UA buffer (8M urea, 150 mM Tris-HCl, pH 8.0) three times and then 100 μ l 25 mM NH_4HCO_3 buffer twice. Trypsin (HLS TRY001C, No. 020201308) was added to the samples (trypsin: protein [wt/wt] ratio 1:50) and the samples were incubated at 37 $^\circ\text{C}$ for 15-18 hours (overnight). The resulting peptides were collected as a filtrate. The peptides of each sample were desalted on C18 Cartridges (EmporeTM SPE Cartridges C18 [standard density], bed I.D. 7 mm, volume 3 ml, Sigma), concentrated by vacuum centrifugation and reconstituted in 40 μ l of 0.1% (v/v) formic acid. The peptide content was estimated by UV light spectral density at 280 nm. For DIA experiments, iRT (indexed retention time) calibration peptides were spiked into the samples.

Data-dependent acquisition (DDA) mass spectrometry assay

The digested peptide pool was separated into 10 fractions using the Thermo ScientificTM PierceTM High pH Reversed-Phase Peptide Fractionation Kit. Each fraction was desalted and reconstituted on C18 Cartridges (EmporeTM SPE Cartridges C18 [standard density], bed I.D. 7 mm, volume 3 ml, Sigma) and then eluted in 40 μ l of 0.1% (v/v) formic acid. Buffer A consisted of a 0.1% formic acid aqueous solution, with formic acid supplied by Fluka (565302-50mL-GL). The analytical gradient was set as follows: 0.00-3.00 minutes, Buffer B linearly increased from 5% to 8%; 3.00-43.00 minutes, Buffer B linearly increased from 8% to 26%; 43.00-48.00 minutes, Buffer B linearly increased from 26% to 40%; 48.00-50.00 minutes, Buffer B linearly increased from 40% to 90%; 50.00-60.00 minutes, Buffer B was maintained at 90%. iRT-Kits (Biognosys) peptides were spiked into the samples before DDA analysis. All sample fractions were analyzed on a TIMSTOF mass spectrometer (Bruker) connected to an Evosep One liquid chromatography system (Denmark). The mass spectrometer was operated in data-dependent mode for ion mobility-enhanced spectral library generation. The accumulation and ramp time were both set to 100 ms, capturing mass spectra within the m/z range of 100-1700 in positive electrospray mode, with a dynamic exclusion

duration of 24.0 seconds. The ion source voltage was set to 1500 V, the temperature to 180 °C, and the dry gas flow to 3 L/min. Ion mobility scanning ranged from 0.75 to 1.35 Vs/cm², followed by eight cycles of PASEF MS/MS.

Mass spectrometry assay for data-independent acquisition (DIA)

Peptides from each sample were analyzed using a TIMSTOF mass spectrometer (Bruker) connected to an Evosep One liquid chromatography system (Denmark) in DIA mode. The mass spectrometer collected ion mobility MS spectra across a mass range of m/z 100-1700. Up to four windows were defined for single 100 ms TIMS scans according to the m/z-ion mobility plane. During PASEF MSMS scanning, the collision energy linearly increased as a function of mobility, from 20 eV at 1/K0 = 0.85 Vs/cm² to 59 eV at 1/K0 = 1.30 Vs/cm².

Mass spectrometry data analysis

To analyze DDA library data, the Spectronaut™ 14.4.200727.47784 (Biognosys) software was used to search a FASTA sequence database downloaded from the Uniprot website (<http://www.uniprot.org>). The iRT peptides sequence (Biognosys|iRT Kit) was added to the database. The specific FASTA file of the sequence database was downloaded under the accession number uniprot_Salmonella_typhimurium_140545_0417 (20210417). Additionally, the database file has been uploaded and is available on iProX. Search parameters were set as follows: enzyme - trypsin; maximum missed cleavages - 1; fixed modification - carbamidomethyl(C); dynamic modifications - oxidation(M) and acetylation (Protein N-terminus). Data reporting for protein identification was based on 99% confidence, as determined by a false discovery rate (FDR) ≤ 1%.

DIA data was analyzed with Spectronaut™ 14.4.200727.47784 by searching the previously constructed spectral library. Key software parameters were set as follows: retention time prediction type - dynamic iRT; interference correction at MS2 level enabled; and cross-run normalization enabled. Results were filtered based on a Q value

cutoff of 0.01 (equivalent to FDR < 1%). The proteomics methodologies were provided by Shanghai Applied Protein Technology Co., Ltd.

Validation of proteomic differential proteins

According to the proteomic results, the significantly differential proteins, including the *csg* operons *csgBAC* and *csgDEFG*, were selected to detect their mRNA abundance through real-time quantitative polymerase chain reaction (RT-qPCR) assays. The primer sequences used for RT-qPCR are shown in Table S9. Total RNA was extracted using an RNA prep Cell/Bacteria Kit (Tiangen, Beijing, China), and the one-step TB Green® PrimeScript™ RT-PCR Kit II (Perfect Real Time; TaKaRa, Dalian, China) was used to perform RT-qPCR. The rough colony strain SF175, smooth colony strain 2015114, and control strain ATCC 14028 were selected as representatives, and the experiments were repeated three times for each sample.

MINOR CORRECTIONS:

“However, in our analysis of proteomic data from 40 strains, we found no statistically significant difference in the expression level of RpoS between rough and smooth colony strains” to “However, in our analysis of proteomic data from 40 strains, we found no statistically significant difference in the abundance of RpoS between rough and smooth colony strains”

As I highlighted in my previous reviewer comments proteomics provides measurements of abundance not expression.

Response:

Thank you for your attention to this matter. Following your suggestion, we have revised the sentence (lines 516-518 in the clean version of the revised manuscript). In addition, we have made necessary revisions to the relevant content in the main text and the supplementary appendix.

Additional corrections needed for the proteomic methods, please fix the following

oversights:

• It is currently stated “We discarded the supernatant and obtained the bacterial samples as the sediments in the EP tubes. We added 125 µL of lysate to each EP tube, vortexed them for 30 seconds” What are the cells resuspend in to generate the 125 µL of lysate?

Response:

Thank you for your comment. To address the oversight you highlighted, we have provided a more detailed description of the protein extraction process in the “Proteomic analysis of the rough and smooth colony strains” section of the revised supplementary appendix.

In short, the lysate was prepared by suspending the cells in an SDT buffer solution, which consisted of 4% Sodium dodecyl sulfate, 100 mM Tris-HCl, 1 mM Dithiothreitol, pH 8.0. This buffer was used to solubilize the cellular components and facilitate the extraction of proteins. The EP tubes were then vortexed to ensure thorough mixing before proceeding to next steps of the protein extraction process.

• Please update the methods and provide the complete name of chemicals used including

DTT – Dithiothreitol

IAA -iodoacetamide

• Please change “detergent DTT” to DTT as DTT is not a detergent.

Response:

We are grateful for your constructive feedback. We have carefully incorporated your suggestions into the “Proteomic analysis of the rough and smooth colony strains” section of the revised supplementary appendix. Specifically, we have provided the complete names of chemicals you mentioned, with DTT referred to as Dithiothreitol and IAA as Iodoacetamide. Furthermore, we have corrected the terminology by

removing the incorrect reference to DTT as a detergent, now identifying it simply as Dithiothreitol.

• Please define what the buffer composition of the “UA buffer” used for proteomic sample prep was. Additionally, this method appears to be similar to the FASP (doi: 10.1038/nmeth.1322.) protocol, please reference the method used for this sample preparation approach.

Response:

We sincerely appreciate your thorough review and apologize for this oversight. We have taken note of your suggestion and have now included the composition of the UA buffer used in our proteomic sample preparation: 8M urea, 150 mM Tris-HCl, pH 8.0. This information can be found in the “Proteomic analysis of the rough and smooth colony strains” section of the revised supplementary appendix. Furthermore, we recognize that our method is similar to the FASP protocol. We have included a reference (supplementary reference #25) to the FASP method to acknowledge the original source and strengthen the scientific basis of our research.

• Please change NH_4HCO_3 to NH_4HCO_3

Response:

Thank you for your suggestion. We have revised NH_4HCO_3 to NH_4HCO_3 in the “Proteomic analysis of the rough and smooth colony strains” section of the revised supplementary appendix.

• Please provide the vendor for the trypsin used for proteomics.

Response:

Thank you for your suggestion. The vendor for the trypsin used in our study is HLS TRY001C, No. 020201308. This information has been included in the “Proteomic

analysis of the rough and smooth colony strains” section of the revised supplementary appendix.

• **Please provide the length of the analytical gradient used for peptide separation. Currently the methods states “The peptide was separated on a C18 Analytical Column (Thermo Scientific, ES802, 1.9 μm, 75 μm*20 cm) with a linear gradient of buffer B (84% acetonitrile in 0.1% formic acid) at a flow rate of 300 nl/min.” but no details on the composition of Buffer A or the length/time of the analytical gradient are provided.**

Response:

Thank you for helping us improve the clarity of our manuscript. We have now provided details for the composition of Buffer A, which consisted of a 0.1% formic acid aqueous solution, with the formic acid supplied by Fluka (565302-50mL-GL). Additionally, we have specified the analytical gradient conditions as follows: from 0.00 to 3.00 minutes, Buffer B was linearly increased from 5% to 8%; from 3.00 to 43.00 minutes, Buffer B was linearly increased from 8% to 26%; from 43.00 to 48.00 minutes, Buffer B was linearly increased from 26% to 40%; from 48.00 to 50.00 minutes, Buffer B was linearly increased from 40% to 90%; and from 50.00 to 60.00 minutes, Buffer B was maintained at 90%. These specific details have been included in the “Proteomic analysis of the rough and smooth colony strains” section of the revised supplementary appendix.

• **Please provide the details of the Uniprot database used for proteomic database searching including the Uniprot accession number.**

Response:

Thank you for your inquiry. The FASTA file of the sequence database used for proteomic database searching was downloaded from the Uniprot website (<http://www.uniprot.org>) under the accession number uniprot_Salmonella_typhimurium_140545_0417 (20210417). Additionally, the

database file has been uploaded and is available on iProX. We have included this information in the “Proteomic analysis of the rough and smooth colony strains” section of the revised supplementary appendix.

- **Please also change the title within the supplementary from “Proteomics sequencing of the rough and smooth colony strains” to “Proteomic analysis of the rough and smooth colony strains” Proteomics is not sequencing samples per say.**

Response:

Thank you for your attention to this detail. In response to your suggestion, the section title in the revised supplementary appendix has been updated to “Proteomic analysis of the rough and smooth colony strains”. Additionally, the same title in the main text has also been revised accordingly.

References

- 1 Gerstel, U. & Römling, U. The *csgD* promoter, a control unit for biofilm formation in *Salmonella typhimurium*. *Res Microbiol* **154**, 659-667 (2003).
<https://doi.org:10.1016/j.resmic.2003.08.005>
- 2 Srivastava, A., Varshney, R. K. & Shukla, P. Sigma Factor Modulation for Cyanobacterial Metabolic Engineering. *Trends Microbiol* **29**, 266-277 (2021).
<https://doi.org:10.1016/j.tim.2020.10.012>
- 3 Gualdi, L., Tagliabue, L. & Landini, P. Biofilm formation-gene expression relay system in *Escherichia coli*: modulation of sigmaS-dependent gene expression by the CsgD regulatory protein via sigmaS protein stabilization. *J Bacteriol* **189**, 8034-8043 (2007).
<https://doi.org:10.1128/jb.00900-07>
- 4 Mika, F. & Hengge, R. Small RNAs in the control of RpoS, CsgD, and biofilm architecture of *Escherichia coli*. *RNA Biol* **11**, 494-507 (2014). <https://doi.org:10.4161/rna.28867>
- 5 Ma, Z. *et al.* A Novel LysR Family Factor STM0859 is Associated with The Responses of *Salmonella Typhimurium* to Environmental Stress and Biofilm Formation. *Pol J Microbiol* **70**, 479-487 (2021). <https://doi.org:10.33073/pjm-2021-045>
- 6 Ogasawara, H. *et al.* Novel regulators of the *csgD* gene encoding the master regulator of biofilm formation in *Escherichia coli* K-12. *Microbiology (Reading)* **166**, 880-890 (2020).
<https://doi.org:10.1099/mic.0.000947>
- 7 Römling, U., Sierralta, W. D., Eriksson, K. & Normark, S. Multicellular and aggregative behaviour of *Salmonella typhimurium* strains is controlled by mutations in the *agfD* promoter. *Mol Microbiol* **28**, 249-264 (1998). <https://doi.org:10.1046/j.1365-2958.1998.00791.x>
- 8 Schirmer, M. *et al.* Insight into biases and sequencing errors for amplicon sequencing with the

- Illumina MiSeq platform. *Nucleic Acids Res* **43**, e37 (2015).
<https://doi.org:10.1093/nar/gku1341>
- 9 Yan, S. *et al.* Genomes-based MLST, cgMLST, wgMLST and SNP analysis of Salmonella Typhimurium from animals and humans. *Comp Immunol Microbiol Infect Dis* **96**, 101973 (2023). <https://doi.org:10.1016/j.cimid.2023.101973>
- 10 Champion, O. L. *et al.* Comparative phylogenomics of the food-borne pathogen *Campylobacter jejuni* reveals genetic markers predictive of infection source. *Proc Natl Acad Sci U S A* **102**, 16043-16048 (2005). <https://doi.org:10.1073/pnas.0503252102>
- 11 Ranieri, M. L., Shi, C., Moreno Switt, A. I., den Bakker, H. C. & Wiedmann, M. Comparison of typing methods with a new procedure based on sequence characterization for Salmonella serovar prediction. *J Clin Microbiol* **51**, 1786-1797 (2013). <https://doi.org:10.1128/jcm.03201-12>
- 12 Higgs, C. *et al.* Optimising genomic approaches for identifying vancomycin-resistant *Enterococcus faecium* transmission in healthcare settings. *Nat Commun* **13**, 509 (2022). <https://doi.org:10.1038/s41467-022-28156-4>
- 13 Berçot, B. *et al.* Ceftriaxone-resistant, multidrug-resistant *Neisseria gonorrhoeae* with a novel mosaic penA-237.001 gene, France, June 2022. *Euro Surveill* **27** (2022). <https://doi.org:10.2807/1560-7917.Es.2022.27.50.2200899>
- 14 Arai, N. *et al.* Phylogenetic Characterization of Salmonella enterica Serovar Typhimurium and Its Monophasic Variant Isolated from Food Animals in Japan Revealed Replacement of Major Epidemic Clones in the Last 4 Decades. *J Clin Microbiol* **56** (2018). <https://doi.org:10.1128/jcm.01758-17>
- 15 Pulford, C. V. *et al.* Stepwise evolution of Salmonella Typhimurium ST313 causing bloodstream infection in Africa. *Nat Microbiol* **6**, 327-338 (2021). <https://doi.org:10.1038/s41564-020-00836-1>
- 16 Almeida, F. *et al.* Phylogenetic and antimicrobial resistance gene analysis of Salmonella Typhimurium strains isolated in Brazil by whole genome sequencing. *PLoS One* **13**, e0201882 (2018). <https://doi.org:10.1371/journal.pone.0201882>
- 17 Gymoese, P. *et al.* Investigation of Outbreaks of Salmonella enterica Serovar Typhimurium and Its Monophasic Variants Using Whole-Genome Sequencing, Denmark. *Emerg Infect Dis* **23**, 1631-1639 (2017). <https://doi.org:10.3201/eid2310.161248>
- 18 Sun, H., Wan, Y., Du, P. & Bai, L. The Epidemiology of Monophasic Salmonella Typhimurium. *Foodborne Pathog Dis* **17**, 87-97 (2020). <https://doi.org:10.1089/fpd.2019.2676>
- 19 Zhang, S. *et al.* SeqSero2: Rapid and Improved Salmonella Serotype Determination Using Whole-Genome Sequencing Data. *Appl Environ Microbiol* **85** (2019). <https://doi.org:10.1128/aem.01746-19>
- 20 Ge, B. *et al.* Genomic analysis of azithromycin-resistant Salmonella from food animals at slaughter and processing, and retail meats, 2011-2021, United States. *Microbiol Spectr* **12**, e0348523 (2024). <https://doi.org:10.1128/spectrum.03485-23>
- 21 Edirmanasinghe, R. *et al.* A Whole-Genome Sequencing Approach To Study Cefoxitin-Resistant Salmonella enterica Serovar Heidelberg Isolates from Various Sources. *Antimicrob Agents Chemother* **61** (2017). <https://doi.org:10.1128/aac.01919-16>
- 22 Reimschuessel, R. *et al.* Multilaboratory Survey To Evaluate Salmonella Prevalence in

- Diarrheic and Nondiarrheic Dogs and Cats in the United States between 2012 and 2014. *J Clin Microbiol* **55**, 1350-1368 (2017). <https://doi.org:10.1128/jcm.02137-16>
- 23 Cho, S. *et al.* Analysis of Salmonella enterica Isolated from a Mixed-Use Watershed in Georgia, USA: Antimicrobial Resistance, Serotype Diversity, and Genetic Relatedness to Human Isolates. *Appl Environ Microbiol* **88**, e0039322 (2022).
<https://doi.org:10.1128/aem.00393-22>
- 24 Meumann, E. M. *et al.* Emergence of Burkholderia pseudomallei Sequence Type 562, Northern Australia. *Emerg Infect Dis* **27**, 1057-1067 (2021).
<https://doi.org:10.3201/eid2704.202716>
- 25 Thorpe, H. A. *et al.* Repeated out-of-Africa expansions of Helicobacter pylori driven by replacement of deleterious mutations. *Nat Commun* **13**, 6842 (2022).
<https://doi.org:10.1038/s41467-022-34475-3>
- 26 Baker, K. S. *et al.* Horizontal antimicrobial resistance transfer drives epidemics of multiple Shigella species. *Nat Commun* **9**, 1462 (2018). <https://doi.org:10.1038/s41467-018-03949-8>
- 27 Das, T. *et al.* Antimicrobial resistance profiling and burden of resistance genes in zoonotic Salmonella isolated from broiler chicken. *Vet Med Sci* **8**, 237-244 (2022).
<https://doi.org:10.1002/vms3.648>
- 28 Chiou, C. S. *et al.* Antimicrobial Resistance and Mechanisms of Azithromycin Resistance in Nontyphoidal Salmonella Isolates in Taiwan, 2017 to 2018. *Microbiol Spectr* **11**, e0336422 (2023). <https://doi.org:10.1128/spectrum.03364-22>
- 29 Morgan-Linnell, S. K. & Zechiedrich, L. Contributions of the combined effects of topoisomerase mutations toward fluoroquinolone resistance in Escherichia coli. *Antimicrob Agents Chemother* **51**, 4205-4208 (2007). <https://doi.org:10.1128/aac.00647-07>
- 30 Jia, B. *et al.* CARD 2017: expansion and model-centric curation of the comprehensive antibiotic resistance database. *Nucleic Acids Res* **45**, D566-d573 (2017).
<https://doi.org:10.1093/nar/gkw1004>
- 31 Jin, Y. *et al.* Genomic Epidemiology and Characterization of Methicillin-Resistant Staphylococcus aureus from Bloodstream Infections in China. *mSystems* **6**, e0083721 (2021).
<https://doi.org:10.1128/mSystems.00837-21>
- 32 Arcadi, E. *et al.* Shallow-Water Hydrothermal Vents as Natural Accelerators of Bacterial Antibiotic Resistance in Marine Coastal Areas. *Microorganisms* **10** (2022).
<https://doi.org:10.3390/microorganisms10020479>
- 33 Nguyen, Q. H. *et al.* Large-scale analysis of putative plasmids in clinical multidrug-resistant Escherichia coli isolates from Vietnamese patients. *Front Microbiol* **14**, 1094119 (2023).
<https://doi.org:10.3389/fmicb.2023.1094119>
- 34 Ingle, D. J., Levine, M. M., Kotloff, K. L., Holt, K. E. & Robins-Browne, R. M. Dynamics of antimicrobial resistance in intestinal Escherichia coli from children in community settings in South Asia and sub-Saharan Africa. *Nat Microbiol* **3**, 1063-1073 (2018).
<https://doi.org:10.1038/s41564-018-0217-4>
- 35 Zhou, Y. *et al.* Exploiting a conjugative endogenous CRISPR-Cas3 system to tackle multidrug-resistant Klebsiella pneumoniae. *EBioMedicine* **88**, 104445 (2023).
<https://doi.org:10.1016/j.ebiom.2023.104445>
- 36 Tyson, G. H. *et al.* The mcr-9 Gene of Salmonella and Escherichia coli Is Not Associated with Colistin Resistance in the United States. *Antimicrob Agents Chemother* **64** (2020).

<https://doi.org:10.1128/aac.00573-20>

- 37 Bourouis, A., Ben Moussa, M. & Belhadj, O. Multidrug-resistant phenotype and isolation of a novel SHV- beta-Lactamase variant in a clinical isolate of *Enterobacter cloacae*. *J Biomed Sci* **22**, 27 (2015). <https://doi.org:10.1186/s12929-015-0131-5>
- 38 Page, A. J. *et al.* Roary: rapid large-scale prokaryote pan genome analysis. *Bioinformatics* **31**, 3691-3693 (2015). <https://doi.org:10.1093/bioinformatics/btv421>
- 39 Alikhan, N. F., Zhou, Z., Sergeant, M. J. & Achtman, M. A genomic overview of the population structure of *Salmonella*. *PLoS Genet* **14**, e1007261 (2018).
<https://doi.org:10.1371/journal.pgen.1007261>
- 40 Wiśniewski, J. R., Zougman, A., Nagaraj, N. & Mann, M. Universal sample preparation method for proteome analysis. *Nat Methods* **6**, 359-362 (2009).
<https://doi.org:10.1038/nmeth.1322>

REVIEWERS' COMMENTS

Reviewer #1 (Remarks to the Author):

I am happy that the authors have addressed everything that I commented on. No further review from me is necessary.

Reviewer #2 (Remarks to the Author):

The authors have addressed the remarks and comments provided during the previous revisions and only one small correction remains to be done for this reviewer:

- Line 143 : "increased prevalence in recent years" should be reconsidered, as the new Figure 1B doesn't show such an increase.

Reviewer #3 (Remarks to the Author):

The authors have now address my concerns and the paper is at a publishable level.

Responses to Reviewers' Comments

We extend our sincere gratitude to the reviewers for their comprehensive evaluation of our revised manuscript and their positive feedback. We appreciate their recognition of our efforts in addressing their previous comments and concerns. Below, we provide a point-by-point response to each reviewer's feedback. The reviewers' comments are presented in bold, followed by our responses in standard font.

Responses to Reviewer #1

I am happy that the authors have addressed everything that I commented on. No further review from me is necessary.

Response: We appreciate the reviewer's satisfaction with our revisions and acknowledgment that we have addressed all previous comments. We are pleased to hear that no further review is necessary from his/her perspective.

Responses to Reviewer #2

The authors have addressed the remarks and comments provided during the previous revisions and only one small correction remains to be done for this reviewer:

- Line 143: "increased prevalence in recent years" should be reconsidered, as the new Figure 1B doesn't show such an increase.

Response: We thank the reviewer for the careful examination of our revised manuscript. We agree that the statement "increased prevalence in recent years" is not fully supported by the data presented in Figure 1B. We have revised the sentence to state that this variant has demonstrated "a persistent prevalence in recent years" to more accurately reflect the trend observed in our analysis.

Responses to Reviewer #3

The authors have now address my concerns and the paper is at a publishable level.

Response: We thank the reviewer for recognizing that the manuscript is now ready for publication.